# Localization of adaptive variants in human genomes using averaged one-dependence estimation

Lauren Alpert Sugden [1,2], Elizabeth G. Atkinson[3], Annie P. Fischer[4], Stephen Rong[1,5], Brenna M. Henn[3] & Sohini Ramachandran[1,2]

Statistical methods for identifying adaptive mutations from population genetic data face several obstacles: assessing the significance of genomic outliers, integrating correlated measures of selection into one analytic framework, and distinguishing adaptive variants from hitchhiking neutral variants. Here, we introduce SWIF(r), a probabilistic method that detects selective sweeps by learning the distributions of multiple selection statistics under different evolutionary scenarios and calculating the posterior probability of a sweep at each genomic site. SWIF(r) is trained using simulations from a user-specified demographic model and explicitly models the joint distributions of selection statistics, thereby increasing its power to both identify regions undergoing sweeps and localize adaptive mutations. Using array and exome data from 45 ǂKhomani San hunter-gatherers of southern Africa, we identify an enrichment of adaptive signals in genes associated with metabolism and obesity. SWIF(r) provides a transparent probabilistic framework for localizing beneficial mutations that is extensible to a variety of evolutionary scenarios.

[1] Center for Computational Molecular Biology, Brown University, Providence, RI 02912, USA. [2] Department of Ecology and Evolutionary Biology, Brown University, Providence, RI 02912, USA. [3] Department of Ecology and Evolution, Stony Brook University, Stony Brook, NY 11794, USA. [4] Division of Applied Mathematics, Brown University, Providence, RI 02912, USA. [5] Department of Molecular Biology, Cell Biology, and Biochemistry, Brown University, Providence, RI 02912, USA. Correspondence and requests for materials should be addressed to L.A.S. (email: lauren_alpert@brown.edu) or to S.R. (email: sramachandran@brown.edu)

Adaptive mutations that spread rapidly through a population, via processes known as selective sweeps, leave distinctive signatures on genomes. These genomic signatures fall into three categories: differentiation among populations, long shared haplotype blocks, and changes in the site frequency spectrum (SFS). Statistics that are commonly used to detect genomic signatures of selective sweeps include fixation index ($F_{ST}$)[1] for measuring population differentiation, integrated haplotype score (iHS)[2] for identifying shared haplotypes, and Tajima's $D$[3] for detecting deviations from the neutral SFS. Some approaches, like SweepFinder[4] and SweeD[5] integrate information across sites by modeling changes to the SFS. Often, statistical scans for adaptive mutations proceed by choosing a particular genomic signature and a corresponding statistic, obtaining the statistic's empirical distribution across loci in a genome-wide dataset, and focusing on loci that fall past an arbitrary, but conservative threshold[2,6,7].

Recently, there has been increased focus on developing composite methods for identifying selective sweeps, which combine multiple statistics into a single framework;[8–15] we refer to the statistics that are aggregated in composite methods such as these as "component statistics." Most composite methods draw upon machine learning approaches like support vector machines[8,11], deep learning[15], boosting[10,13], or random forest classification[14] in order to identify genomic windows containing selective sweeps. These windows vary in size from 20 to 200 kb, often identifying candidate sweep regions containing many genes[14,15]. One method, the Composite of Multiple Signals or "CMS"[9,12], uses component statistics that can be computed site by site in pursuit of localizing adaptive variants within genomic windows, but the output from this method cannot be interpreted without comparison to a genome-wide distribution. In addition, CMS must rely on imputation or other methods of compensation when component statistics are undefined, a complication that typically does not arise when using window-based component statistics. In a subset of populations from the 1000 Genomes Project, we found that more than half of variant sites had at least one undefined component statistic (Supplementary Table 1); iHS was frequently undefined because it requires a minor allele frequency of 5% to be computed[2], and along with cross-population extended haplotype homozygosity (XP-EHH)[16], cannot be calculated near the ends of chromosomes or sequenced regions. This poses a particular problem when scanning for complete sweeps, defined here as sweeps in which the beneficial allele has fixed in the population of interest.

Here we introduce a probabilistic classification framework for detecting and localizing adaptive mutations in population-genomic data called SWIF(r) (SWeep Inference Framework (controlling for correlation)). SWIF(r) has three major features that enable genome-wide characterization of adaptive mutations: first, SWIF(r) computes the per-site calibrated probability of selective sweep, which is immediately interpretable and does not require comparison with a genome-wide distribution; second, SWIF(r) can be run without imputing undefined statistics; and third, we explicitly learn pairwise joint distributions of selection statistics, which gives substantial gains in power to both identify regions containing selective sweeps and localize adaptive variants. Existing composite methods for selection scans have subsets of these features (e.g., CMS[9,12] returns site-based scores, and evolBoosting[10], evoNet[15], and other machine learning approaches[13,14] leverage correlations among component statistics), but SWIF(r) combines all three in a unified statistical framework. Our approach incorporates the demographic history of populations of interest, while being robust to misspecification of that history, and is also agnostic to the frequency of the adaptive allele, identifying both complete and incomplete selective sweeps in a

population of interest. We assess SWIF(r)'s performance in simulations against state-of-the-art univariate and composite methods for identifying genomic targets of selective sweeps, and we confirm that we can localize known adaptive mutations in the human genome using data from the 1000 Genomes Project. We then apply SWIF(r) to identify previously unidentified adaptive variants in genomic data from the ‡Khomani San, an understudied hunter-gatherer KhoeSan population in southern Africa representing the most basal human population divergence. Open-source software for training and running SWIF(r) is freely available at https://github.com/ramachandran-lab/SWIFr.

## Results

**Overview.** Here we describe the theoretical framework of SWIF(r), and compare SWIF(r) to existing methods that scan genome-wide datasets and identify genomic sites or regions containing adaptive mutations. We then apply SWIF(r) to data from the 1000 Genomes Project and from the ‡Khomani San population of southern Africa. Note that we train SWIF(r) on simulations of hard sweeps (see Methods); our focus here is not on the relative roles of various modes of selection in shaping observed human genomic variation (for recent treatments on this question see refs. [17–19]), although we note that SWIF(r) is extensible to multi-class classification, and could be used in future applications to explore multiple modes of selection. In this study, our focus is on localizing genomic sites of adaptive mutations that have spread through populations of interest via hard sweeps.

**Implementation of SWIF(r).** SWIF(r) draws on statistical inference and machine learning to localize the genomic site of a selective sweep based on probabilities that incorporate dependencies among component statistics. Unlike genomic outlier approaches, the output of SWIF(r) can be interpreted directly for each genomic site: given a set of $n$ component statistics for a site, SWIF(r) calculates the probability that the site is neutrally evolving or, alternatively, is the site of a selective sweep. We will refer to these two classes as "neutral" and "adaptive," respectively, and these posterior probabilities can be computed as follows:

$$P(\text{adaptive}|S_1 = s_1, ..., S_n = s_n) =$$
$$\frac{\pi P(s_1,...,s_n|\text{adaptive})}{\pi P(s_1,...,s_n|\text{adaptive}) + (1-\pi)P(s_1,...,s_n|\text{neutral})},$$
$$P(\text{neutral}|S_1 = s_1, ..., S_n = s_n) =$$
$$1 - P(\text{adaptive}|S_1 = s_1, ..., S_n = s_n), \quad (1)$$

where $s_1,...,s_n$ represent observed values for $n$ component statistics such as iHS and $F_{ST}$, and $\pi$ is the prior probability of a sweep, which may be altered to reflect different genomic contexts. If a component statistic is undefined at a site, it is simply left out of Eq. 1, and does not need to be imputed. The data for learning the likelihood terms, $P(s_1,...,s_n|\text{adaptive})$ and $P(s_1,...,s_n|\text{neutral})$, come from calculating component statistics on simulated haplotypes from a demographic model with and without simulated selective sweeps comprising a range of selection coefficients and present-day allele frequencies (see Methods). We note that this general framework is similar to that used by Grossman et al.[9] for CMS, which assumes that the component statistics are independent, and computes the product of posterior probabilities $P(\text{adaptive}|s_i)$ for each statistic $s_i$. SWIF(r) strikes a balance between computational tractability and model accuracy by learning joint distributions of pairs of component statistics, thereby relaxing this strict independence assumption.

We base SWIF(r) on a machine learning classification framework called an averaged one-dependence estimator (AODE)[20], which is built from multiple one-dependence estimators (ODEs), each of which conditions on a different component statistic in

order to compute a posterior sweep probability at a given site. An ODE conditioning on $S_j$ assumes that all other component statistics are conditionally independent of one another, given the class (neutral or adaptive) and the value of $S_j$. As shown in Eq. 2, this assumption effectively reduces the dimensionality of the likelihood terms $P(s_1,...,s_n|\text{class})$ in Eq. 1. The AODE then reduces variance by averaging all possible ODEs to produce a posterior probability that incorporates all pairwise joint probability distributions (Eq. 3; see Methods).

Assumption made by $\text{ODE}_j$ (ODE conditioning on $S_j$):

$$P(s_1, ..., s_n|\text{class}) = P(S_j = s_j|\text{class}) \prod_{i \neq j} P(S_i = s_i|S_j = s_j, \text{class}).$$

(2)

SWIF(r):

$$P(\text{adaptive}|S_1 = s_1, \ldots, S_n = s_n) =$$

$$\frac{\pi \sum_{j=1}^{n} \left[ P(S_j=s_j|\text{adaptive}) \prod_{i \neq j} P(S_i=s_i|S_j=s_j,\text{adaptive}) \right]}{\left[ \pi \sum_{j=1}^{n} \left[ P(S_j = s_j|\text{adaptive}) \prod_{i \neq j} P(S_i = s_i|S_j = s_j, \text{adaptive}) \right] + (1 - \pi) \sum_{j=1}^{n} \left[ P(S_j = s_j|\text{neutral}) \prod_{i \neq j} P(S_i = s_i|S_j = s_j, \text{neutral}) \right] \right]}.$$

(3)

**Calibration of posterior probabilities calculated by SWIF(r)**. A desirable property of probabilities, like those calculated by SWIF(r), is that they be well calibrated: in this context, for the variant positions where the posterior probability reported by SWIF(r) is around 60%, approximately 60% of those sites should contain an adaptive mutation, and approximately 40% should be neutral. We implemented a smoothed isotonic regression (IR) scheme to calibrate the probabilities calculated by SWIF(r) (see Methods). Briefly, when applying SWIF(r) to a given dataset, we calculate the empirical frequencies of neutral and sweep variants that are assigned posterior probabilities between 0 and 1 in simulation, and use IR[21] to map the posterior probabilities to their corresponding empirical sweep frequencies (Supplementary Figs. 1, 2). We then impose a smoothing function that prevents multiple posterior probabilities from being mapped to the same calibrated value (Supplementary Figs. 1e, 2e, 3). This calibration procedure relies on the relative makeup of the training set; a classifier that is calibrated for a training set made up of neutral and sweep variants in equal parts would not be well calibrated for a training set in which sweep variants only make up 1% of the whole. For each application of SWIF(r) in this study, we calibrated SWIF(r) for a specific training set makeup (see Methods; also see Supplementary Figs. 1, 2).

The calibrated probabilities reported by SWIF(r) can be interpreted directly as the probability that a site contains an adaptive mutation, or fed into a straightforward classification scheme by way of a probability threshold; in this study, we classify sites with a posterior probability above 50% as adaptive SWIF(r) signals. The classifier may be tuned by altering either this threshold or the prior sweep probability $\pi$ (Supplementary Fig. 4).

**Performance of SWIF(r) using simulated data**. We implemented SWIF(r) using the following component statistics: $F_{ST}$[1], XP-EHH[16] (altered as in Wagh et al.[22]; Supplementary Note 1), iHS[2], and difference in derived allele frequency (ΔDAF). These statistics can each be calculated site by site in a genomic dataset,

and all but iHS leverage cross-population comparisons. Training simulations used the demographic model of Europeans, West Africans, and East Asians inferred by Schaffner et al.[23], and simulated selective sweeps within each of those populations (see Methods). We compared SWIF(r)'s performance against each component statistic, SweepFinder[4], composite method CMS[9] (altered by excluding ΔiHH because of non-normality; Supplementary Note 2 and Supplementary Fig. 5), and window-based sweep-detection methods evoNet[15], and evolBoosting[10]. We also evaluated the robustness of SWIF(r) to both demographic model misspecification and background selection.

In Fig. 1a and Supplementary Fig. 6, we evaluate the ability of SWIF(r) to localize the site of an adaptive mutation against that of its component statistics, the composite method CMS, and SweepFinder. The performance of each component statistic varies with different sweep parameters: for example, iHS is most powerful for identifying adaptive mutations that have not yet risen to high frequency within the population of interest, while XP-EHH and ΔDAF are more effective for those that have (Supplementary Fig. 7). This underscores the advantage of composite methods for detecting selective sweeps when the parameters of the sweep are unknown[8–15]. Aggregating over many different sweep parameters, SWIF(r) outperforms each component statistic, as well as CMS and SweepFinder, improving the tradeoff between the false-positive rate (fraction of neutral variants incorrectly classified as adaptive) and true-positive rate (fraction of adaptive mutations that are correctly classified as such) (Fig. 1 and Supplementary Fig. 6). SWIF(r) also outperforms CMS in distinguishing adaptive mutations from linked neutral variation (Supplementary Fig. 8). The performance of SWIF(r) is particularly striking for incomplete sweeps: for example, in Fig. 1b, SWIF(r) achieves up to a 50% reduction in the false-positive rate relative to CMS for adaptive mutations that have only swept through 20% of the population at the time of sampling (see also Supplementary Fig. 6a–c, noting that SweepFinder was designed to identify complete sweeps in a population of interest). For the same incomplete sweep simulations summarized in Fig. 1b, Fig. 1c shows the performance of each of the individual ODEs (Eq. 2); in this particular evolutionary scenario, conditioning on $F_{ST}$ or ΔDAF results in the best performance. However, the best-performing ODE changes based on the parameters of the selective sweep (Fig. 1d). By averaging across all ODEs, SWIF(r) is robust to variable performance of ODEs in the absence of prior knowledge of the true sweep parameters (Fig. 1a–c).

While few composite methods for sweep detection operate site by site, there are a handful of machine learning composite approaches that identify genomic windows containing adaptive mutations[10,13–15]. In order to compare SWIF(r) against such methods, we had to alter SWIF(r) to calculate window-based sweep probabilities; there are many potential ways to do this that may be differentially powerful, and here we chose simply to use the highest probability assigned to any variant within a given genomic window as the probability for that window. We compared window-based SWIF(r) to two state-of-the-art composite window-based methods: evolBoosting[10], which combines 120 statistics using boosted logistic regression, and evoNet[15], which was developed to jointly infer demography and selection using 345 component statistics within a deep learning framework. When comparing SWIF(r) with evolBoosting, we use 40 kb windows following Lin et al.[10], and for comparison with evoNet, we use 100 kb windows following Sheehan and Song[15]. We show that SWIF(r) outperforms both methods across a range of sweep parameter values (Supplementary Figs. 9, 10). We note that while the alterations we make to SWIF(r) in order to produce window-based probabilities likely downplay the strengths of SWIF(r),

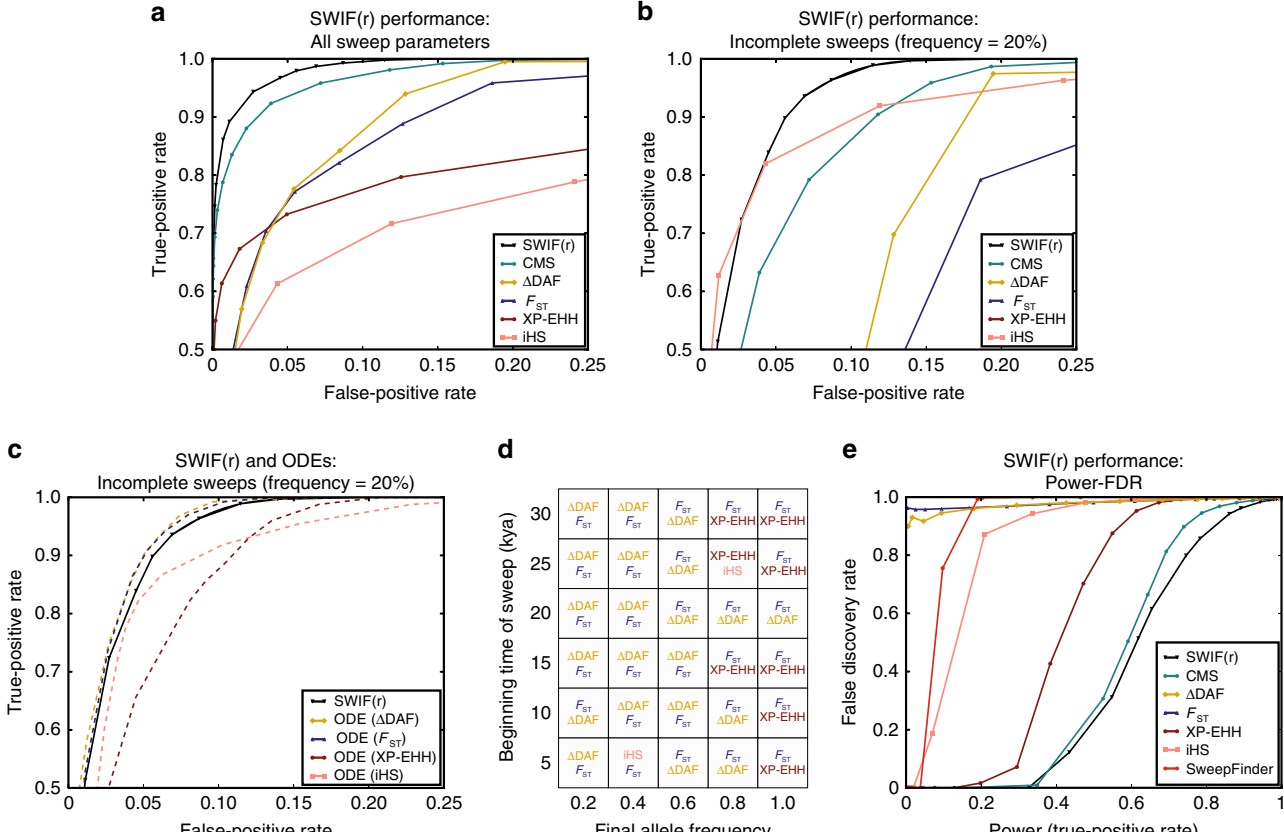

**Fig. 1** SWIF(r) outcompetes existing site-based sweep-detection methods for a range of sweep model parameters. For comparison against window-based sweep-detection methods, see Supplementary Figs. 9 and 10. **a** Receiver operating characteristic (ROC) curves comparing SWIF(r) with CMS[9], $F_{ST}$, iHS, XP-EHH, and ΔDAF across all simulated neutral and sweep scenarios (see Methods). False-positive rate is the fraction of simulated neutral sites that are incorrectly classified as adaptive, and the true-positive rate is the fraction of simulated sites of adaptive mutations that are correctly classified. SWIF(r) constitutes an improvement in the tradeoff between true and false positives. SweepFinder[4] is not visible here or in **b** (Supplementary Fig. 6). **b** ROC curves for incomplete sweeps in which the beneficial allele has a population frequency of 20%. SWIF(r) reduces the false-positive rate by up to 50% relative to CMS. **c** ROC curves for SWIF(r) and the four component ODEs for incomplete sweeps in which the beneficial allele has a frequency of 20%. Since the AODE is an average of the ODEs, there will always be individual ODEs that match or outperform SWIF(r); in this case, the ODEs conditioned on $F_{ST}$ and ΔDAF both achieve this. **d** The two highest-performing ODEs for different sweep parameters. Performance is defined as area under the ROC curve. The statistic that leads to the highest-performing ODE is listed first in bold, followed by the second best. Colors correspond to ROC curves in panels **a**–**c**. While ODEs conditioned on ΔDAF tend to perform extremely well for sweeps that are at lower frequency, ODEs conditioned on $F_{ST}$ and XP-EHH tend to perform better for sweeps that are near-complete or complete in the population of interest. By averaging the ODEs, SWIF(r) is robust to uncertainty about the true parameters of the sweep. **e** Power-FDR curves for SWIF(r), CMS, $F_{ST}$, iHS, XP-EHH, ΔDAF, and SweepFinder. Power is equivalent to true-positive rate as defined in **a**–**c**, and false discovery rate is defined as the fraction of sites classified as adaptive that are actually neutral. These curves assume a training set composed of 99.95% neutral variants and 0.05% adaptive variants. See Supplementary Fig. 11 for curves based on other training set compositions

SWIF(r) does benefit from the inclusion of cross-population statistics, while both evolBoosting and evoNet rely on component statistics defined for single populations.

We also evaluated the performance of SWIF(r) and other methods in the context of the tradeoff between power and false discovery rate (FDR). Power-FDR curves for site-based methods in Fig. 1e and Supplementary Fig. 6f assume a training set composition of 99.95% neutral variants and 0.05% adaptive variants. Curves for other training set compositions can be found in Supplementary Fig. 11. To generate a training set for window-based methods (Supplementary Figs. 9f, 10f), we assume that 1% of windows contain a sweep. Finally, we evaluated the sensitivity of SWIF(r) to demographic misspecification and background selection (see Methods). We find that SWIF(r) is robust to misspecification of multiple parameters including divergence times, bottleneck strength, population sizes, and population size changes (Supplementary Figs. 12, 13), and that the presence of background selection does not induce SWIF(r) to identify false-positive signals (Supplementary Fig. 14).

**SWIF(r) localizes canonical adaptive mutations in humans.** For application to data from phase 1 of the 1000 Genomes Project, we used training simulations from the Schaffner demographic model[23], calibrated SWIF(r) for a training set composed of 0.01% sweep variants and 99.99% neutral variants (Supplementary Fig. 1), and applied it to SNP (single-nucleotide polymorphism) array data from West African (YRI), East Asian (CHB and JPT), and European (CEU) populations (Supplementary Note 3). SWIF(r) reports high sweep probabilities at multiple SNPs within known and suspected selective sweep loci in each of these populations (Supplementary Datas 1, 2). Figure 2 illustrates the ability of SWIF(r) to localize sites of adaptive mutations within genomic regions containing canonical sweeps. Adaptive SNPs have been determined via functional experiments in SLC24A5[24], DARC[25], and HERC2[26]; we find that modeling the dependency structure among component statistics within SWIF(r) enables statistical localization of these experimentally identified adaptive mutations (Fig. 2a, c, d). Methods that treat component statistics as independent, as CMS does, cannot localize these mutations (Supplementary Figs. 15, 16). In CHB and JPT,

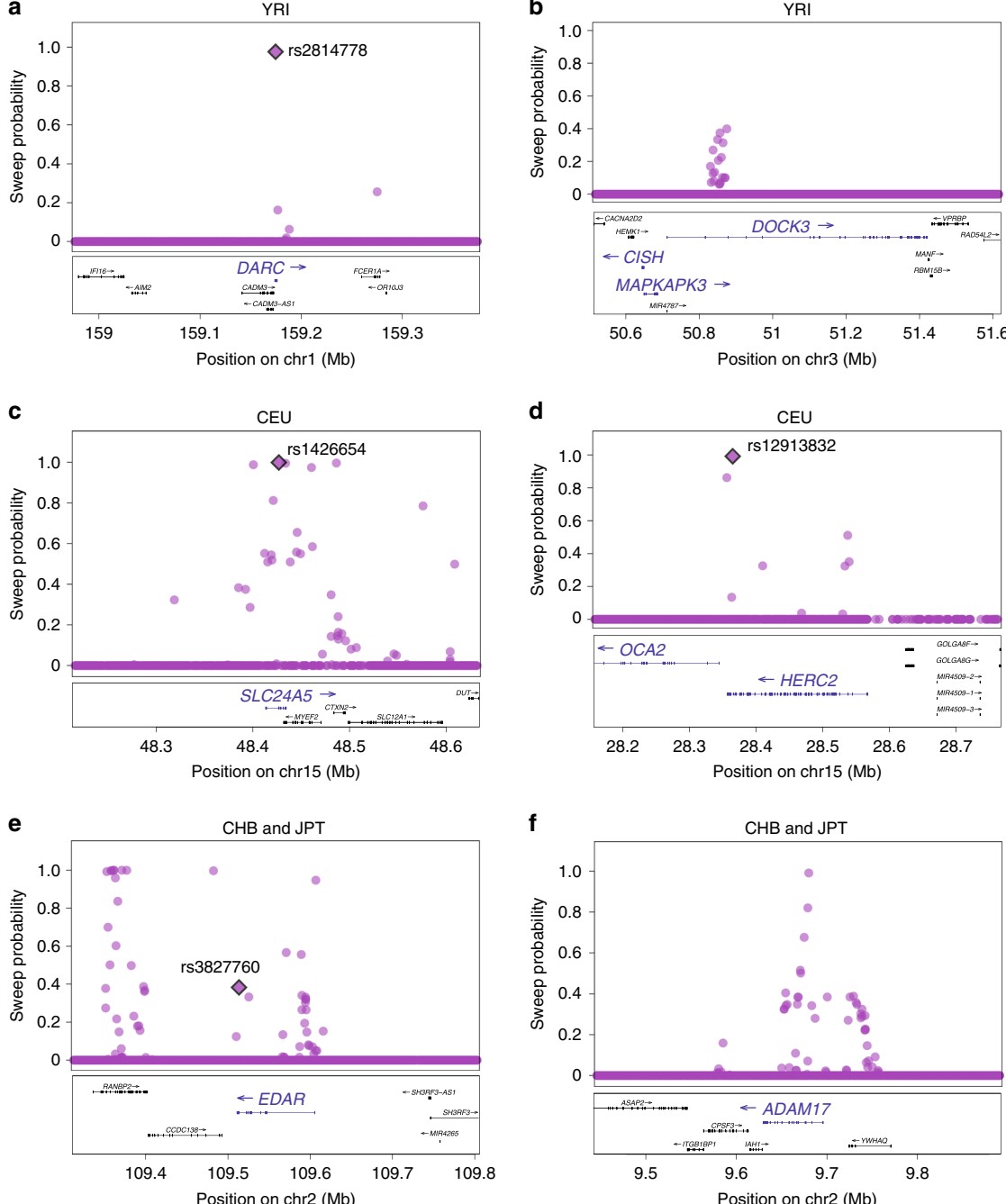

**Fig. 2** SWIF(r) localizes canonical adaptive mutations and provides additional evidence for suspected sweeps in YRI, CHB and JPT, and CEU[63]. Plotted points indicate the calibrated posterior sweep probability calculated by SWIF(r) at each site, using prior sweep probability $\pi = 10^{-5}$. Plots were made with LocusZoom[64]. Where available, functionally verified adaptive SNPs are depicted as filled diamonds and labeled with rsids. **a**, **b** In YRI, two loci where SWIF(r) reports high sweep probabilities are *DARC* and *DOCK3*. *DARC* encodes the Duffy antigen, located on the surface of red blood cells, and is the receptor for malaria parasites. The derived allele of the causal SNP shown has been determined to be protective against *Plasmodium vivax* malaria infection[25]. *DOCK3*, along with neighboring genes *MAPKAPK3* and *CISH*, are all associated with variation in height, and have previously been shown to harbor signals of selection in Pygmy populations[65]. *CISH* may also play a role in susceptibility to infectious diseases, including malaria[66]. **c**, **d** In CEU, we uncover multiple loci in genes involved in pigmentation, including rs1426654 in *SLC24A5*, which is involved in light skin color[24], and rs12913832 in the promoter region of *OCA2*, which is functionally linked with eye color and correlates with skin and hair pigmentation[26]. rs1426654 has the highest sweep probability reported by SWIF(r) in *SLC24A5* (0.9992 after smoothed calibration; see Supplementary Data 1); note each panel depicts genomic windows containing multiple genes. **e**, **f** In CHB and JPT, SWIF(r) recovers a strong adaptive signal in the vicinity of *EDAR*; multiple GWA studies have shown rs3827760 to be associated with hair and tooth morphology[67,68]. SWIF(r) also identifies variants with high sweep probability in *ADAM17*, which is involved in pigmentation[69], and has been identified in other positive selection scans in East Asian individuals[70,71]

SWIF(r) recovers a strong adaptive signal in the vicinity of *EDAR*, offering new hypotheses for targets of selection in this genomic region. Whole-genome results with gene annotations can be found in Supplementary Figs. 7, 8, 9 and Supplementary Data 1 (see also Supplementary Fig. 20). False discovery estimates can be found in Supplementary Table 2. Of the 126 genes across these populations with SWIF(r) signals (i.e., at least one variant within the gene has posterior hard sweep probability >50%), 63% were identified in at least one positive selection scan conducted in humans (Supplementary Data 2).

**KhoeSan sweeps are enriched for metabolism-related genes.** We applied SWIF(r) to samples from the ‡Khomani San, a formerly hunter-gatherer KhoeSan population of the Kalahari desert in southern Africa, using Illumina SNP array data for 670,987 phased autosomal sites genotyped in 45 individuals[27] (Supplementary Note 4). The KhoeSan have likely occupied southern Africa for ~100,000 years, and maintain the largest long-term $N_e$ of any human population[28], a feature that facilitates adaptive evolution. We trained SWIF(r) on simulations from the Gronau demographic model[29] (see Methods, Supplementary Fig. 21, and Supplementary Table 3), and implemented an ascertainment modeling scheme to produce a training dataset with population-level site frequency spectra similar to the observed array data. Briefly, for each simulated haplotype, SNPs were subsampled to match the empirical three-dimensional unfolded SFS for YRI, CEU, and CHB and JPT individuals in the 1000 Genomes Project on the chips used to genotype the ‡Khomani San (see Methods and Supplementary Figs. 22, 23). We calibrated SWIF(r) for this dataset based on a training set composed of 0.05% sweep variants and 99.95% neutral variants (Supplementary Fig. 2). After applying SWIF(r) to SNP data, we then examined whether genomic regions identified by SWIF(r) contain annotated functional mutations identified in high-coverage exome data from the same 45 individuals[30] (Supplementary Note 5).

SWIF(r) identifies a number of genomic regions bearing signatures of selective sweeps in the ‡Khomani San, driven by extreme values in multiple component statistics that together produce a posterior sweep probability >50% (Fig. 3a, b; see also Supplementary Figs. 24, 25 and Supplementary Table 4). These signals comprise 108 SNPs, of which 94 are distributed across 80 genes, and the remaining 14 are intergenic, defined as genomic variants that do not land within 50 kb of an annotated gene (Supplementary Data 3). We observe an abundance of SWIF(r) signals within the major histocompatibility complex (MHC), a region of immunity genes for which studies have indicated ongoing selection in many populations[31,32], including an iHS outlier scan in the ‡Khomani San[33]. We show in Supplementary Fig. 26 that the SWIF(r) signals in this region are not qualitatively different from the SWIF(r) signals we see throughout the genome, despite the fact that balancing selection is typically thought to be the primary mode of selection in the MHC[34].

We tested for a common functional or phenotypic basis among the 80 genes bearing SWIF(r) signals by conducting a gene ontology enrichment analysis across public databases with Enrichr[35]. We found that these genes are significantly enriched for dbGaP categories related to adiponectin, body mass index (BMI), and triglyceride phenotypes (Fig. 3c and Table 1). Specifically, SNPs in genes related to adiponectin (*ADIPOQ*, *PEPD*, *DUT*, and *ASTN2*) have among the highest posterior sweep probabilities (all ≥75%). SWIF(r) also identified SNPs within three other genes (*PDGFRA*, *SIDT2*, and *PHACTR3*) that have previously been associated with obesity and metabolism phenotypes (Fig. 3b and Table 1). Some of the genes highlighted in Fig. 3b are also involved in muscle-based phenotypes

(Supplementary Note 6), but here we focus on the substantial evidence supporting the association of the highlighted genes with obesity and metabolism phenotypes in prior genome-wide association (GWA) and functional studies (Table 1).

One variant that SWIF(r) identifies, rs6444174, has a calibrated sweep probability of 90%, driven by extreme values at this SNP in $F_{ST}$, XP-EHH, and ΔDAF (Fig. 3a; empirical p values $4.4 \times 10^{-4}$, $4.0 \times 10^{-4}$, $5.5 \times 10^{-4}$, respectively). This variant lies in *ADIPOQ*, which is expressed predominantly in adipose tissue[36], and codes for adiponectin, a regulator of glucose and fatty acid metabolism. In a study of associations between *ADIPOQ* variants and adiponectin levels and obesity phenotypes in 2,968 African-American participants, rs6444174 was found to be associated with serum adiponectin levels in female participants ($p = 6.15 \times 10^{-5}$), and with BMI in all normal-weight participants ($p = 3.66 \times 10^{-4}$). The allele at high frequency in the ‡Khomani individuals studied here corresponds to decreased adiponectin levels and increased BMI, respectively[37].

**Exome support for ‡Khomani San sweep loci.** This SWIF(r) scan was performed using SNP array data ascertained from primarily Eurasian polymorphisms, a common feature of commercial SNP array platforms. Thus, the observed SWIF(r) signals are likely tagging haplotypes common in the ‡Khomani San, and may not themselves be causal polymorphisms. We examined high-coverage exome data[30] within each gene to identify putatively functional mutations near the sites identified by SWIF(r) (see Supplementary Data 4 for full results). This allows us to identify variants not captured on SNP array platforms, including variants that are unique to the ‡Khomani San. We note that we did not include the MHC genes in this exome analysis, because of potential issues with mapping and phasing of exome sequence data in the MHC region. In *ADIPOQ*, we identify a missense mutation, rs113716447, for which the nearest SNP that is present on the SNP array is rs6444174 (<1 kb away); rs6444174 has a calibrated SWIF(r) sweep probability of 90%, the highest in *ADIPOQ* (Fig. 4). The missense T allele at rs113716447 is at high frequency in the ‡Khomani San relative to all other populations sequenced in the 1000 Genomes Project (27% vs. <0.5%; Fig. 4). Furthermore, in the Simons Genome Diversity Project, whose samples are drawn from 130 diverse and globally distributed human populations, only four copies of the missense allele at rs113716447 are found: two copies in a ‡Khomani San individual, and one copy each in a Namibian San individual and a Jul'hoansi San individual. This SNP defines the two major haplogroups within the *ADIPOQ* gene in a median-joining haplotype network for the gene region (Supplementary Fig. 27), providing some support for selection at this SNP.

Two other genes highlighted in Fig. 3b harbor promising polymorphisms that may be related to the underlying causal haplotypes. In *PEPD*, we identify a polymorphism at 10% frequency in the ‡Khomani (chr19:33,882,361) which is a missense mutation approximately 42 kb from the SNP identified by SWIF(r). We also identify a missense mutation in the first exon of *PHACTR3* at 38% frequency in this sample, which is at <2% frequency in other global populations including other Africans sequenced as part of the 1000 Genomes Project. Because the SNP array density is low, we expect that SWIF(r) signals in this population may in many cases be somewhat removed from the causal variants that these signals tag. We note that intronic variants in both *PEPD* and *PHACTR3* have been identified as *cis*-expression quantitative trait loci that affect RNAseq expression in adipose tissue, in two independent northern European cohorts[38].

For some of the genes identified by SWIF(r), exome data either were not generated or did not reveal nearby functional

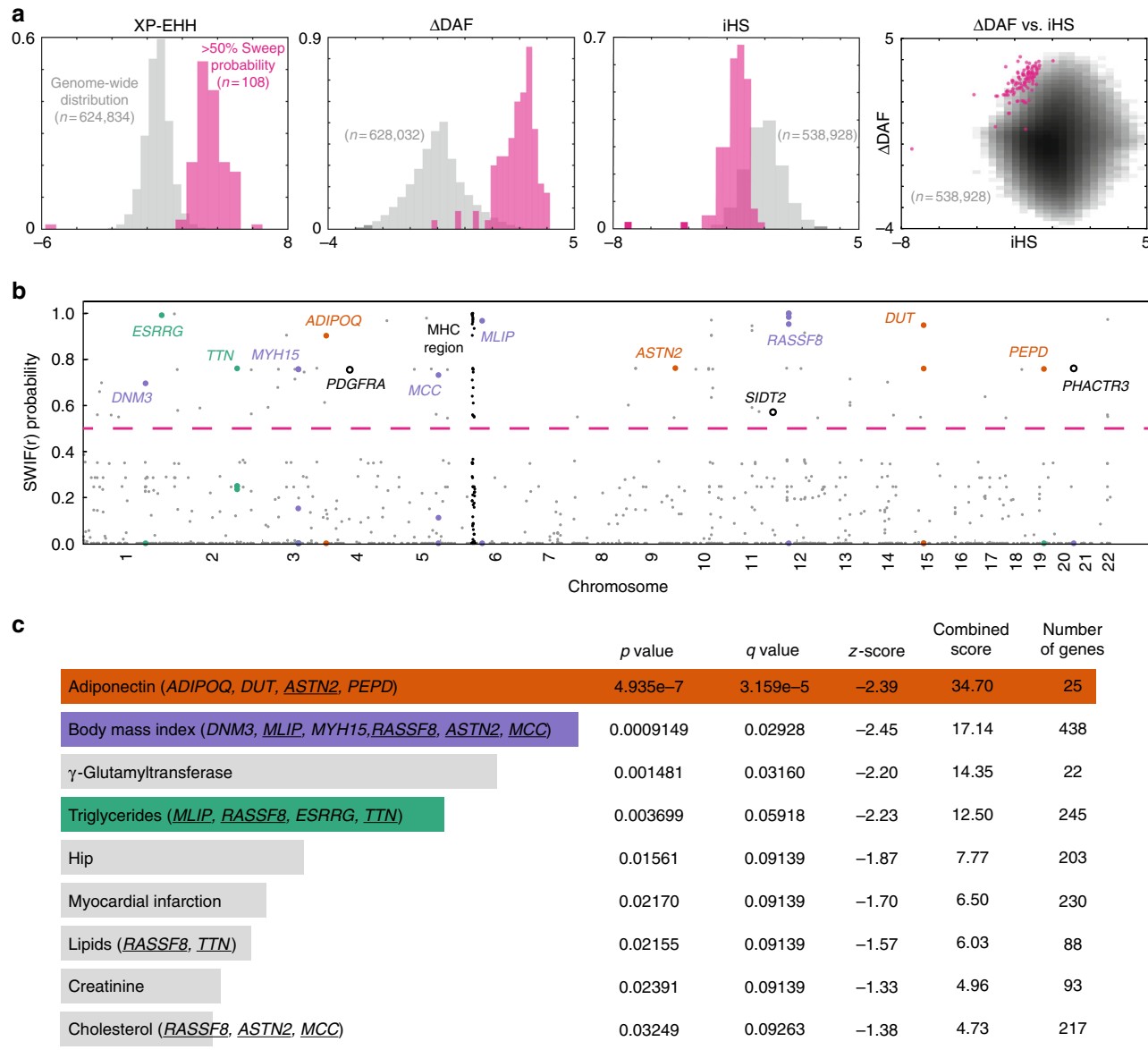

**Fig. 3** Genome-wide SWIF(r) scan for adaptation in ‡Khomani San SNP array data. **a** Empirical genome-wide univariate distributions of three of the component statistics, XP-EHH, ΔDAF, and iHS are shown in gray, as is the empirical joint distribution of ΔDAF and iHS (darker bins have more observations than lighter bins). The number of sites in each genome-wide univariate distribution differs due to some component statistics being undefined more often than others. In pink are the corresponding distributions for the 108 variants that SWIF(r) identifies as having posterior sweep probabilities of >50% (variants above the dashed pink line in **b**). The full set of distributions can be found in Supplementary Fig. 25. **b** The value plotted for each position along the genome is the calibrated posterior probability of adaptation computed by SWIF(r) (per-site prior for a selective sweep is $\pi = 10^{-4}$ to detect signals of relatively old sweeps given the high long-term $N_e$ of the ‡Khomani San); only SNPs with a calibrated posterior sweep probability >1% are plotted and the horizontal line indicates a probability cutoff of 50%. A strong signal of adaptation over the major histocompatibility complex on chromosome 6 is shown in black. Gene names are listed for genes previously associated with metabolism-related and obesity-related traits (colors match categories in **c**; open circles denote genes of interest that are not in any category in **c**). **c** We used gene set enrichment analysis tool Enrichr[35] to identify categories that had an overrepresentation of genes containing SWIF(r) signals (Supplementary Data 3). We found multiple enriched dbGaP categories related to metabolism and obesity, including adiponectin (a protein hormone that influences multiple metabolic processes, including glucose regulation and fatty acid oxidation), body mass index, and triglycerides. Genes in these categories containing SWIF(r) signals are listed next to category names. p values, q values, and the total number of genes are shown for each category, and categories are ranked by a combined score computed by Enrichr[35]. Adiponectin, body mass index, and γ-glutamyltransferase all have q values below 5%

polymorphisms with differential allele frequencies between the ‡Khomani San and other worldwide populations. One such gene is *RASSF8*, previously annotated as under positive selection in the Namibian and ‡Khomani San populations relative to western Africans using XP-EHH;[39] in our SNP array analysis, we detect a cluster of four SNPs in *RASSF8* within 70 kb of each other, each with SWIF(r) sweep probability >98%. *RASSF8* is present in the BMI, triglycerides, lipids, and cholesterol dbGaP categories (Fig. 3c), yet functional mutations underlying this SWIF(r) signal remain elusive.

**Table 1 Multiple published functional and association studies link genes identified by SWIF(r) to metabolism-related and obesity-related phenotypes**

| Gene | SNP(s) | DAF | P (sweep) | Studies relating gene to metabolism-obesity phenotype | Refs. |
|---|---|---|---|---|---|
| DNM3 | rs12121064 | 54% | 70% | Associated with waist–hip ratio in Europeans (rs1011731, GWA $p = 7.5 \times 10^{-9}$) | 73 |
| | | | | Associated with waist circumference in Hispanic women (GWA replication $p = 1.5 \times 10^{-3}$) | 74 |
| | | | | Associated with weight loss after gastric bypass surgery | 75 |
| ESRRG | rs11808388 | 62% | 99.2% | Regulates adipocyte differentiation by modulating the expression of adipogenesis-related genes | 76 |
| | | | | Candidate obesity-susceptibility gene based on epigenetic profile and association with BMI | 77 |
| | | | | May be involved in increasing the potential for energy expenditure in brown adipocytes | 78 |
| | | | | Mediates hepatic gluconeogenesis | 79 |
| | | | | Contributes toward maintenance of hepatic glucose homeostasis | 80 |
| | | | | Necessary for metabolic maturation of pancreatic $\beta$-cells | 81 |
| | | | | Significantly up-regulated under treatment with cholesterol drug fenofibrate | 82 |
| TTN | rs16866534 | 44% | 76% | Isoform composition in cardiac tissue is regulated by insulin signaling, possibly contributing to altered diastolic function in diabetic cardiomyopathy | 83 |
| MYH15 | rs3957559 | 49% | 76% | Variant rs3900940, along with four other variants, contributes to elevated risk for coronary heart disease | 84 |
| ADIPOQ | rs6444174 | 56% | 90% | **rs6444174** associated with adiponectin levels in African-American women | 37 |
| | | | | Associated with plasma adiponectin levels in Europeans (rs17366568, GWA $p = 4.3 \times 10^{-24}$) | 85,86 |
| | | | | Associated with plasma adiponectin levels in African Americans (rs4686807, GWA $p = 1.6 \times 10^{-11}$) | 87 |
| | | | | Associated with plasma adiponectin levels in East Asians (rs822391, GWA $p = 1.6 \times 10^{-10}$) | 87 |
| | | | | Associated with coronary heart disease, BMI, childhood obesity, metabolic syndrome, and type II diabetes | 37,88–92 |
| | | | | Serum adiponectin levels are associated with metabolic health and cardiovascular risk | 93,94 |
| PDGFRA | rs4530695 | 64% | 75% | Used in molecular biology as a marker for white adipocytes | 95 |
| | | | | Controls pancreatic $\beta$-cell proliferation | 96 |
| | | | | Plays a role in the link between obesity and inhibited placental development | 97 |
| ASTN2 | rs16934033 | 50% | 76% | Contributes to genetic variation of plasma triglyceride concentrations | 98 |
| | | | | marginally associated with childhood obesity in Hispanic individuals (GWA $p = 2.4 \times 10^{-6}$) | 99 |
| SIDT2 | rs11605217 | 27% | 57% | Important regulator of insulin secretion | 100 |
| | | | | Mice without the gene are glucose intolerant and have decreased serum insulin | 101,102 |
| | | | | Associated with triglyceride levels (rs1242229, GWA $p = 3.1 \times 10^{-20}$) | 103 |
| RASSF8 | rs16929850 | 61% | 95.3% | Expression is significantly altered by fasting in mice | 104 |
| | rs16929965 | 64% | 99.9% | | |
| | rs2729646 | 68% | 98.4% | | |
| | rs956627 | 64% | 99.9% | | |
| DUT | rs11637235 | 33% | 76% | Missense variant causes a syndrome characterized in part by early onset diabetes mellitus | 105 |
| PEPD | rs12975240 | 62% | 76% | Associated with adiponectin levels in multiple populations (rs731839, rs4805885, rs8182584, rs889139, rs889140, GWA $p$ values between $1.1 \times 10^{-9}$ and $2.2 \times 10^{-13}$) | 87,106 |
| | | | | Associated with type II diabetes (rs3786897, GWA $p = 1.3 \times 10^{-9}$) | 107 |
| | | | | Associated with fasting insulin levels (rs731839, GWA $p = 5.1 \times 10^{-12}$) | 108 |
| | | | | Associated with serum lipid levels | 109 |
| | | | | Expression is modulated by n-3 fatty acids | 110 |
| PHACTR3 | rs1182507 | 54% | 76% | Regulation in adipose tissue is BMI-dependent | 38 |
| | | | | Candidate obesity gene based on epigenetic profile | 77 |

The second column contains all variants within the genes listed that have posterior sweep probability ≥50% as calculated by SWIF(r). Column 3 shows the DAF in the ‡Khomani San at the SNP in column 2, and column 4 shows the calibrated posterior sweep probability calculated by SWIF(r) at that site. For GWA studies, GWA $p$ values are given for the strongest SNP associations. Bold rsid indicates a result about the specific SNP identified by SWIF(r) in column 2. All genes highlighted in Fig. 3b are included in this table except MCC and MLIP, for which additional associations to metabolism-related and obesity-related phenotypes could not be found beyond the dbGaP categories in Fig. 3c
BMI, body mass index; DFA, derived allele frequency; GWA, genome-wide association; SNP, single-nucleotide polymorphism; SWIF(r), SWeep Inference Framework (controlling for correlation)

## Discussion

In this paper, we have presented a method for selective sweep detection, SWIF(r), and insight into adaptive evolution in the ‡Khomani San. SWIF(r) outperforms existing summary statistics and composite methods when detecting both complete and incomplete sweeps in simulation (Fig. 1), and localizes experimentally validated adaptive mutations using genomic data alone (Fig. 2). When analyzing genotype and exome data from 45 ‡Khomani San individuals, we found that SWIF(r) signals are enriched in genes associated with metabolism and obesity (Figs. 3, 4 and Table 1).

Composite classification frameworks such as SWIF(r) quantitatively ground a common qualitative approach used in scans for adaptive sweeps based on summary statistics: evidence for selection at a locus is considered stronger when extreme values are observed for more than one statistic (Fig. 3a). Furthermore, machine learning approaches like SWIF(r) that incorporate joint distributions of selection statistics can detect sweep events that individual univariate statistics cannot (Supplementary Fig. 28). SWIF(r) additionally reports calibrated probabilities site by site, resulting in a transparent probabilistic framework for localizing adaptive mutations rather than adaptive regions. While approaches such as approximate Bayesian computation can exploit higher-dimensional correlations in order to distinguish between selective sweep modes at candidate loci[40], this comes at the cost of genome-scale tractability, and can be vulnerable to the curse of dimensionality[15]. The AODE framework allows us to transparently calculate probabilities without the need for imputation of undefined statistics, and our priors are made explicit, allowing for clearer interpretation. Future applications of SWIF(r) and other

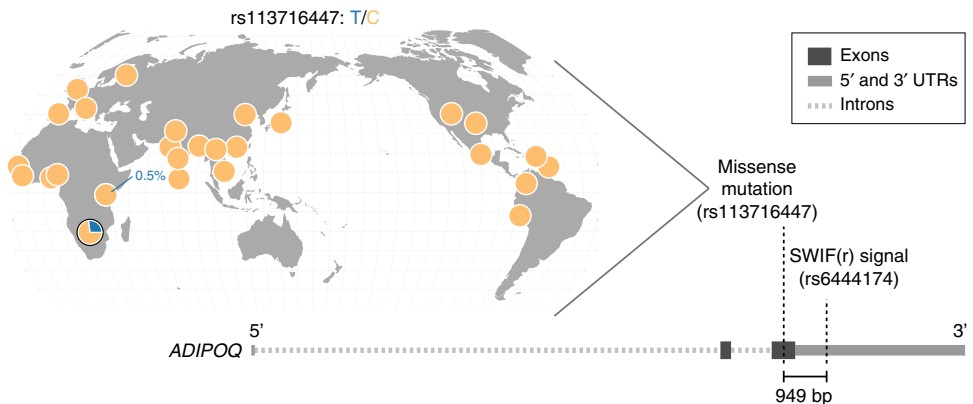

**Fig. 4** Missense mutation rs11316447 is a potential causal mutation in *ADIPOQ*. Worldwide distribution of rs11316447 generated by the Geography of Genetic Variants Browser[72] (http://popgen.uchicago.edu/ggv/) shows that the T allele carried by 27% of ‡Khomani San individuals (pie chart outlined in black) is extremely rare throughout phase 3 of the 1000 Genomes[63], at a maximum of 0.5% in the Luhya population of Kenya. The diagram of *ADIPOQ* highlights the positions of the variant identified by SWIF(r) (rs6444174) and the nearby missense variant (rs11316447). These two variants are within 1 kb of each other, suggesting that the SWIF(r) signal at rs6444174 is tagging this missense variant

composite sweep-detection frameworks can easily incorporate new summary statistics such as iSAFE[41], which ranks candidate adaptive mutations in a predefined region under selection (Supplementary Fig. 29). Future applications of SWIF(r) can additionally assign variable site-specific prior sweep probabilities according to genomic annotations: for example, one could assign a smaller prior for synonymous variants relative to non-synonymous variants, or a higher prior in regulatory regions relative to intergenic regions. Our algorithm for modeling SNP ascertainment in training simulations (see Methods) can also enable the application of SWIF(r) to genotype array data from diverse understudied populations, as well as the use of SFS statistics, which are more vulnerable to ascertainment bias, within the SWIF(r) framework.

In order for the class probabilities reported by SWIF(r) to be practically interpretable, we calibrated SWIF(r), such that $k\%$ of variants with a posterior sweep probability of $k\%$ are indeed sweep variants. We have implemented a calibration scheme based on isotonic regression (IR) for SWIF(r) that maps the posterior sweep probabilities to their empirical sweep proportions in simulated data (Supplementary Figs. 1–3), but importantly, this calibration relies on the composition of the training set used. While for some classifiers, the proportions of classes are known, or can be reliably estimated (e.g., see Scheet and Stephens[42]), the proportion of sites throughout the human genome that are adaptive is unknown. For calibrating SWIF(r), we chose training sets made up overwhelmingly of neutral variants; while our calibration of SWIF(r) always preserves the rank order of posterior probabilities (Supplementary Fig. 3), the specific choice of training set makeup can have a dramatic effect on the calibration, and future applications of SWIF(r) can use different criteria for calibration. For example, one could calibrate SWIF(r) based on specific strengths of selection, or perform different calibrations for scenarios in which certain component statistics are undefined. In each case, a direct interpretation of the posterior probabilities reported by SWIF(r), or any other classifier that calculates probabilities, must incorporate knowledge of the scenarios used for training and calibration.

One caveat for interpretation of the SWIF(r) results presented here is that we train SWIF(r) on hard selective sweeps. In simulation, we found that SWIF(r) is also sensitive to sweeps

from standing variation with a low initial frequency (Supplementary Fig. 30); indeed, the sweep in West Africans in the gene *DARC*, for which SWIF(r) calculates a high sweep probability (Fig. 2a) has recently been shown to have originated from standing variation in the ancestral population[43]. Given the multiple metabolism-related and obesity-related sweep targets identified by SWIF(r) in the ‡Khomani San (Fig. 3), we also suspect that some putative adaptive mutations identified by SWIF(r) may be components of polygenic adaptation.

Genes with SWIF(r) signals in our high-throughput genomic scan for selective sweeps in the ‡Khomani San have been independently identified in multiple GWA studies and functional experiments as associated with metabolism-related and obesity-related phenotypes (Table 1). One way to interpret this signal is through the lens of the "thrifty gene" hypothesis, which posits that ready fat storage was positively selected for in hunter-gatherer populations due to the survival advantage it conferred in unreliable food cycles[44]. The hypothesis further states that modern disease phenotypes such as type 2 diabetes and obesity are the consequence of a radical shift from ancestral environments and forager subsistence strategies to a contemporary environment with abundant food in the form of simple sugars, starches, and high fat, though this is a subject of much debate (e.g., see ref. [45]). Although most indigenous Khoe and San groups of the Kalahari are classically considered small and thin, populations such as the Khoekhoe cow/goat pastoralists are characterized by steatopygia (i.e., extensive fat accumulation along the buttocks and thighs in women), as notoriously described by early European explorers and anthropologists[46,47]. While the thrifty gene hypothesis would predict an increase in metabolic pathology for these individuals, studies have shown that accumulated subcutaneous gluteofemoral fat, found in patients exhibiting steatopygia[48], is protective against diabetes and other metabolic disorders[49]. The mutations and genes identified by our SWIF(r) scan, such as *ADIPOQ*, are natural targets for functional assays to determine the origins and consequences of subcutaneous vs. visceral fat; future studies could merge such assays with phenotypic data on diabetes and metabolic syndromes in KhoeSan groups to gain new insight into the "obesity-mortality paradox"[50].

In this selection scan, we also see an abundance of SWIF(r) signals in the MHC region involved in immunity. It is possible that this signal reflects balancing selection, which is the mode of

selection canonically thought to be occurring within this region[34]. Indeed, it has been shown that the signatures of balancing selection and incomplete or recurrent sweeps may be similar to signatures of positive selection[51,52]. We note, however, that other studies using different methodologies have detected signatures of directional selection in the MHC in human populations[32,53], and others have noted that fluctuating directional selection is a possible mechanism for pathogen-mediated selection in this region[54].

The probabilistic framework of SWIF(r) suggests two natural extensions for future applications. First, while the use of SNP-based component statistics enables us to localize adaptive mutations, SWIF(r) could easily incorporate region-based component statistics, including composite likelihood approaches like XP-CLR[55], and SFS-based measures in order to help detect older selection events, for which haplotype-based statistics are less powerful. Second, future studies can exploit the flexibility and interpretability of SWIF(r) to conduct multi-class classification. Supplementary Figure 31 illustrates a preliminary extension of SWIF(r) that classifies sweeps based on the start time of positive selection. Recent methods have attempted multi-class sweep classification using hierarchical binary classification or other machine learning approaches[10,14,18], but without the benefit of a transparent probabilistic framework in which priors are made explicit. Using the probabilistic framework of SWIF(r), future studies could determine the mode of adaptive evolution at genomic sites, including background selection or sweeps from standing variation or recurrent mutation[17,18], or infer the timing or selective strength of an adaptive event[40]. Thus, SWIF(r) offers a technical advance in genome-wide sweep detection that can yield new insight into the modes and roles of selection in shaping population-genomic diversity.

## Methods

**Simulation of haplotypes for 1000 Genomes analysis.** Simulations based on the demographic model of African, Asian, and European populations outlined in Schaffner et al.[23] were carried out with the following alterations necessary for allowing the simulation of recent selective sweeps (within the past 30 kya): no modern population growth (within the past 30 generations), and migration ending 500 generations following the Asian/European split instead of continuing to the present. We carried out 100 simulations of 1 Mb regions from this neutral demographic model using cosi. We generated a new recombination map for each simulation with the recosim package within cosi, using a hierarchical recombination model that assumes a regional rate drawn from the observed distribution of rates in the deCODE genetic map[56], and then randomly generates recombination hotspots with randomly drawn local rates[23]. For each simulation, we generated 120 1-Mb-long haplotypes from each of the three populations (Supplementary Note 7).

Selective sweeps continued until the time of sampling, and were simulated for a range of sweep parameters: start time ranging over (5, 10, 15, 20, 25, 30 kya), final allele frequency ranging over (0.2, 0.4, 0.6, 0.8, 1.0), and population of origin ranging over (African, Asian, European). Note that these sweeps cover a range of incomplete as well as complete sweeps in a population of interest. The selection coefficients for each parameter set are fully determined by the effective population size, sweep start time, and final allele frequency, and are displayed in Supplementary Table 3. We calculated these selection coefficients using Eq. 4 for complete sweeps (by which we mean sweeps where the beneficial mutation has reached fixation in the population of interest), and Eq. 5 for incomplete sweeps, where $t_1$ is the sweep start time and $t_2$ is the sweep end time, both measured in generations from the present, $N_e$ is the effective population size, $t_s = (t_1 - t_2)/2N_e$, and $\phi$ is the present-day frequency of the beneficial allele, $0 < \phi \leq 1$[57]. The range of selection coefficients corresponds to a range of $\alpha = 2N_s$ of ~100–4,500. For each set of sweep parameters, we generated 100 simulations with the adaptive allele located halfway along the 1 Mb region, for a total of 9000 sweep training points.

$$s = \frac{\log(2N_e)}{t_s N_e},$$  (4)

$$s = \frac{1}{2N_e t_s}\log(2N_e - 1) + \log\left(\frac{1-\phi}{\phi}\right).$$  (5)

**Simulation of haplotypes for ‡Khomani San analysis.** To train our classifier to identify selective sweeps in the ‡Khomani San, we used the demographic model inferred by Gronau et al.[29] based on six diploid whole-genome sequences, one from each of six populations: European, Yoruban, Han Chinese, Korean, Bantu, and San. We used the inferred population sizes, coalescent times, and migration rates reported by Gronau et al.[29], which are calibrated based on a 6.5 million year human–chimpanzee divergence, the presence of migration between the Yoruban and San populations, and 25 years per generation to construct a demographic model for the Han Chinese, European, Yoruban, and San, shown in Supplementary Fig. 21.

Because Uren et al.[27] showed that the ‡Khomani San have experienced recent gene flow from both Western Africa and Europe, we replaced migration in the Gronau model with two pulses of recent migration: one pulse from the Yoruban population with migration rate 0.179 at 7 generations ago, and one pulse from the European population with migration rate 0.227 at 14 generations ago. We found that these rates resulted in present-day admixture levels that matched those found in Uren et al.[27] (Supplementary Note 4).

Using cosi[23], we simulated 1 Mb genomic regions, comprising both neutral and sweep scenarios as described earlier, with sample sizes matching the number of individuals in the filtered 1000 Genomes dataset, and the number of San individuals in our study.

Because the ‡Khomani have been isolated for so long, we included additional sweep scenarios, with sweeps beginning and ending between 30 and 60 kya. We called these "old sweeps" and trained SWIF(r) on three classes: neutral, "old sweeps," and "recent sweeps" (those occurring within the past 30 kya). Since we found that SWIF(r) did not have enough power to reliably distinguish between old and recent sweeps (Supplementary Fig. 31), in applications to data, we only considered the total probability of a sweep, given by the sum of the posterior probabilities for both sweep classes.

**Analysis of demographic robustness.** We first assessed the sensitivity of SWIF(r) to the demographic model used in training simulations using two demographic models of West Africans, Europeans, and East Asians, from Schaffner et al.[23] ("Schaffner model") and Gronau et al.[29] ("Gronau model"). We trained SWIF(r) using simulations from the Gronau model (including simulation of ascertainment bias), and tested SWIF(r) on simulated haplotypes drawn from the Schaffner model (with and without selective sweeps). To test SWIF(r)'s robustness to misspecification of recent population growth, we generated a set of simulations using a demographic model from Gravel et al.[58] that estimates recent exponential population expansions with rates of 0.38% for Europe and 0.48% for East Asia over the last 23,000 years. We also implemented a version of the same demographic model with doubled rates of expansion (0.75% for Europe and 0.96% for East Asia). Since the simulation software cosi[23] cannot simulate sweeps and population-level changes simultaneously, we allowed the expansion to last from 23,0000 years ago to 5000 years ago, to allow for sweeps beginning 5000 years ago. We also included the migration rates inferred by Gravel et al.[58] between Europe, East Asia, and Africa, and between Africa and the ancestral population of Europe and East Asia. As in other analyses, We simulated selective sweeps spanning a range of present-day allele frequencies from 20 to 100%.

**Simulation of background selection.** For evaluating SWIF(r)'s robustness to background selection, we generated three sets of simulations of 1 Mb each using forward simulator slim[59]: neutral regions, regions with a hard sweep, and genic regions. For genic regions, we followed Messer and Petrov[17] to simulate gene structure: each simulation had one gene with 8 exons of 150 bp each, separated by introns of 1.5 kb, and flanked by a 550 bp 5'-untranslated region (UTR) and a 250 bp 3'-UTR. Within exons and UTRs, 75% of sites were assumed to be functional. Mutations were assumed to be codominant, and fitness effects across different sites were assumed to be additive. Functional sites were divided into 40% "strongly deleterious" sites with selection coefficient −0.1, and 60% "weakly deleterious" sites with selection coefficients between −0.01 and −0.0001. The mutation rate was set at $2.5 \times 10^{-8}$ per-site per generation, and the recombination rate at $10^{-8}$.

For all three sets of simulations, we simulated two populations with $N_e = 5,000$, which split from each other 40,000 years ago. For sweep simulations, we drew selection coefficients for the beneficial allele from an exponential distribution with mean 0.03, and set sweeps to begin 10,000 years ago. For testing the robustness of SWIF(r) to background selection, we trained SWIF(r) to distinguish between neutral mutations (from simulated neutral regions), and adaptive mutations (from simulated sweep regions), and then applied this classifier to variants in simulated genic regions undergoing background selection (mutations in UTR, exonic, and intronic sites).

**Implementation of SWIF(r).** For ease of comparison, we built SWIF(r) using the same statistics that comprise CMS[9,12] (Supplementary Note 8): $F_{ST}$, XP-EHH (adapted for improved performance on incomplete sweeps; Supplementary Note 1), iHS, and change in derived allele frequency (ΔDAF). ΔiHH was excluded in applications to real data because of non-normality (Supplementary Note 2 and Supplementary Fig. 5). To avoid overfitting the joint distributions modeled in the AODE framework (Eq. 3), we fit Gaussian mixture models with full covariance matrices (i.e., containing nonzero off-diagonal entries) to the joint probability

distributions of each pair of statistics $S_i$ and $S_j$ within each scenario $C$ (neutral or sweep), $P(S_i = s_i, S_j = s_j | C)$, with the number of components ranging between three and five based on Bayesian information criterion (BIC) curves. Joint probabilities were learned using sites for which both component statistics were defined. We used the python package scikit-learn[60] to compute the BIC curves and fit the mixture models. These mixture models capture the salient features of each pairwise joint distribution, as illustrated in the example in Supplementary Fig. 32. Given the smoothed joint distribution learned for a pair of statistics $(S_i, S_j)$, we calculate the conditional probability distributions $S_j | s_i$ as one-dimensional gaussian mixtures:

$$S_j | s_i \sim \sum_k w^{(k)} N\left( \mu_j^{(k)} + \frac{\sigma_j^{(k)}}{\sigma_i^{(k)}} \rho(s_i - \mu_i^{(k)}), (1 - (\rho^{(k)})^2)(\sigma_j^{(k)})^2 \right)$$

where $N(\mu, \sigma^2)$ denotes the normal distribution with mean $\mu$ and variance $\sigma^2$, $k$ indexes the components in the joint Gaussian mixture, $w^{(k)}$ is the weight assigned to each component, $\mu_i$ and $\mu_j$ are the components of the joint mean, and $\sigma_i$, $\sigma_j$, and $\rho$ are taken from the joint covariance matrix.

We find that SWIF(r) loses power to identify sites with adaptive mutations when iHS is undefined, likely because iHS is far more powerful for detecting incomplete sweeps than the other four statistics. SWIF(r) loses very little power when other component statistics are undefined (Supplementary Fig. 33).

**Calibration of SWIF(r) probabilities.** There are a few techniques for calibrating probabilities returned by a binary classifier so that of all the data points that are given a $k\%$ probability of belonging to class A by the classifier, $k\%$ of those are indeed drawn from class A, and $(100-k)\%$ are drawn from class B[21,61]. IR is a popular method because it makes no assumptions about the mapping function beyond requiring that it be monotonically increasing[62]. In the case of SWIF(r) probabilities, IR calibration works by grouping sweep and neutral variants from a training dataset into posterior probability bins, and mapping each bin to the empirical proportion of variants in the bin that are sweep variants. We used 10 bins for calibration, because we found that using more bins increased the risk of overfitting. This can be mitigated by performing more simulations, but in our case, even with 1000 neutral simulations of 1 Mb each, mid to high posterior probabilities were extremely rare at neutral variants. For localizing sweep sites in data from the 1000 Genomes Project, we calibrated SWIF(r) based on a training dataset composed of 99.99% simulated neutral variants and 0.01% simulated sweep variants, and for application to the ‡Khomani San SNP array data, we calibrated SWIF(r) based on a dataset composed of 99.95% simulated neutral variants and 0.05% simulated sweep variants (Supplementary Figs. 1–3). The slightly larger fraction of sweep simulations in the ‡Khomani San training set relative to the 1000 Genomes training set allowed for more sensitivity to older sweeps, and accounted for the sparser SNP density of this dataset. In both datasets, we restricted the simulated sweep variants to those with present-day allele frequencies over 50%, since we have the most power in this realm (Supplementary Fig. 6), and wanted to avoid overcorrection of strong signals.

A downside to IR is that by its nature, it maps a range of input values to the same output value, which removes some information about which probabilities are larger than others. We implemented a “smoothed” IR for calibration that interpolates the piecewise constant mapping function learned by IR (Supplementary Fig. 3). In practice, we find that both methods of calibration produce equally well-calibrated classifiers; that is, after either method, the data points in our simulated dataset that have a calibrated posterior sweep probability of $k\%$ are made up of approximately $k\%$ sweep simulations and $(100-k)\%$ neutral simulations (Supplementary Figs. 1, 2). Unlike IR alone, however, smoothed IR has the advantage of preserving strict monotonicity of posterior probabilities.

**Implementation of CMS.** We implemented CMS following the algorithm described in Grossman et al.[9] and personal communication with the authors. Based on the simulations and component statistics described above, CMS is computed as the product of individual posterior distributions:

$$\text{CMS} = \prod_{i=1}^{n} P(\text{sweep} | S_i = s_i) = \prod_{i=1}^{n} \frac{\pi P(S_i = s_i | \text{sweep})}{\pi P(S_i = s_i | \text{sweep}) + (1 - \pi) P(S_i = s_i | \text{neutral})}, \quad (6)$$

where $\pi$, the prior probability of a sweep, is $10^{-6}$.

When one or more component statistics are undefined at a locus, CMS is not well defined. If statistics are simply left out of the product, this artificially inflates the reported score. Some compensation is thus required to avoid such a bias, which is not discussed by Grossman et al. 2010[9]. We implemented a conservative compensation scheme: if statistic $S_i$ is undefined at a locus, we set its value to the mean of the distribution of that statistic learned from neutral simulations.

For the purpose of evaluating CMS using YRI, CEU, and CHB and JPT samples from the 1000 Genomes, we use CMS Viewer (https://pubs.broadinstitute.org/mpg/cmsviewer/; use date: 26 April 2016), an interactive tool designed by Grossman et al.[12] for visualizing genome-wide CMS scores.

**Implementation of window-based methods.** We implemented evolBoosting using the R package released by Lin et al.[10] (http://www.picb.ac.cn/evolgen/softwares/) using default settings. We trained and tested evolBoosting on the simulations of YRI, CEU, and CHB and JPT described above, including all sweep durations and present-day allele frequencies, splitting the simulations in two equally sized groups for training and testing. We used the middle 40 kb of each 1 Mb simulation, and generated window-based SWIF(r) probabilities by taking the maximum posterior sweep probability for all SNPs within the 40 kb window.

We implemented evoNet using the software package released by Sheehan and Song[15] (https://sourceforge.net/projects/evonet/) using default settings and 345 component statistics described by the authors (personal communication with S. Mathieson). Training and testing was done as described above for evolBoosting[10], except that we used the central 100 kb windows of each 1 Mb simulation (following Sheehan and Song[15]). For comparison with SWIF(r), we generated window-based SWIF(r) probabilities by taking the maximum posterior sweep probability for all SNPs within the 100 kb window.

**ROC analysis.** To generate the receiver operating characteristic (ROC) curves for CMS, SweepFinder, and the component statistics (Fig. 1a–c), we varied the threshold for classifying a mutation as adaptive in order to cover the range from ~0% false-positive rate to ~100% true-positive rate. For SWIF(r) and the ODEs, we varied the prior $\pi$, and sites with scores >0.5 were classified as adaptive (Supplementary Fig. 4). To generate Fig. 1d, we partitioned all simulations by present-day frequency of the adaptive mutation and sweep start time. For each pair of these parameters, we approximated the area under the ROC curves (AUROC) by summing the areas of the trapezoids defined by each pair of neighboring points in the ROC plane, then identified the summary statistics with the highest and second-highest AUROC. ROC curves for window-based SWIF(r), evoNet, and evolBoosting were generated in much the same way, except that we varied the threshold for classifying a window as containing an adaptive variant.

**Ascertainment modeling.** For our selection scan in the ‡Khomani San population, we use genotype data from two SNP arrays;[27] the ascertainment bias of these arrays means that the simulated haplotypes we generate from the four populations (‡Khomani San, YRI, CEU, and CHB and JPT) for training SWIF(r) differ dramatically from the observed data for these populations at the sites genotyped on the arrays. To account for this, we implemented an ascertainment modeling algorithm that prunes sites from simulated haplotypes in order to provide SWIF(r) with simulations for training that match the SFS of the observed data as closely as possible. The key to this algorithm is to define regions of joint SFS space that are similar in terms of representation on the SNP arrays (e.g., SNPs with low derived allele frequency in all populations are fairly common, while SNPs that are highly differentiated across multiple populations are relatively rare). Defining these “equivalence classes” (hereafter referred to as “SFS regions”) in joint SFS space allows us to learn the density of SNPs from each SFS region along the SNP arrays, and then to thin simulations in order to re-create those densities. This first requires smoothing of the joint SFS to account for sparsity. The full algorithm is as follows:

Step 1: Learn the empirical three-dimensional (3D) SFS for YRI, CEU, and CHB and JPT individuals in the 1000 Genomes Project, restricted to SNPs present in the overlap between the Illumina OmniExpress and OmniExpressPlus platforms (Supplementary Note 5). This results in a 3D array of SNP counts for each triplet of derived allele frequencies (DAF$_{\text{YRI}}$, DAF$_{\text{CEU}}$, and DAF$_{\text{CHB and JPT}}$). For this dataset, given 87 YRI individuals, 81 CEU individuals, and 186 CHB and JPT individuals, the dimensions of this three-dimensional array are $175 \times 165 \times 373$ ($2n+1$ in each dimension for $n$ individuals).

Step 2: To account for sparseness in the empirical 3D SFS, subdivide each axis into 40 evenly spaced bins to create a new $40 \times 40 \times 40$ array where each entry is the average SNP count within that 3D bin; this array approximates the original empirical 3D SFS. Use the one-dimensional histogram of average SNP counts across all $40^3$ bins to define five intervals that span the range of counts, then assign each bin to its interval (Supplementary Fig. 34). Groups of bins belonging to the same interval will be hereafter referred to as “SFS regions.” We note that we choose a $40 \times 40 \times 40$ array for smoothing because it resulted in SFS regions with well-defined boundaries in 3D space; these dimensions may need to be altered for other datasets to achieve well-defined boundaries as in Supplementary Fig. 34.

Step 2a: In most SFS regions, the SNP counts in the 3D SFS are relatively invariant; however, in the SFS region with the highest SNP counts (the region in red in Supplementary Fig. 34, corresponding predominantly to SNPs with low derived allele frequency in all populations), there is a wide range of SNP counts (this is analogous to the higher variability in counts of low-frequency variants in the one-dimensional site frequency spectrum relative to that of medium-frequency and high-frequency variants). To account for this increased variability, apply a similar procedure as above: subdivide each bin in the highest SFS region by 2 in each dimension (resulting in 8 sub-bins), re-learn the average SNP count within that sub-bin, use a histogram of average SNP counts across sub-bins to again define five intervals, and assign each sub-bin to its interval, thereby defining an additional set of SFS regions that gives better resolution in higher-density areas.

Step 3: For each 1 Mb block along the SNP array, count the number of SNPs that fall in each SFS region, based on the observed derived allele frequencies at each SNP for YRI, CEU, and CHB and JPT. This provides a measure of SNP density (counts per Mb) for each SFS region. Applying this over a sliding window of 1 Mb across the entire SNP array results in a distribution of densities for each SFS region.

Step 4: Within each 1 Mb block of simulated sequence data, assign each simulated SNP to its SFS region. For each SFS region, draw a value from the distribution of SNP densities learned in step 3, then randomly downsample the number of simulated SNPs that fall in that region to match this value. In the rare case in which downsampling is not possible for a given SFS region (i.e., there are fewer simulated SNPs in that region than the value drawn from the distribution of densities), retain all simulated SNPs that belong to the SFS region.

Step 5: For training the classifiers, restrict the simulated ‡Khomani San genotype data (as well as the simulated data from the 1000 Genomes populations) to the downsampled set of SNPs.

**Software availability**. SWIF(r) repository: https://github.com/ramachandran-lab/SWIFr
selscan repository: https://github.com/szpiech/selscan

**Data availability**. 1000 Genomes phase 1 data: ftp://ftp.1000genomes.ebi.ac.uk/vol1/ftp/phase1/analysis_results/integrated_call_sets/.
‡Khomani San genotype data were first described by Uren et al.[27], and ‡Khomani San exome data were first described by Martin et al[30]. Queries regarding access to ‡Khomani San data analyzed here should be sent to the South African San Council for research and ethics review by contacting both Leana Snyders (leanacloete@ymail.com) and admin@sasi.org.za.

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

## Acknowledgements

We thank Dean Bobo, Barbara Engelhardt, Chris Gignoux, David Guertin, Erik Sudderth, Zachary Szpiech, Jeremy Mumford, Lorin Crawford, Paul Norman, and the Ramachandran Lab for helpful discussions; we also thank Shari Grossman, Ilya

Shlyakhter, and Pardis Sabeti for discussing details regarding the implementation of CMS, and Sara Mathieson for help implementing evoNet. We are grateful to Ryan Hernandez for multiple discussions regarding analysis of exome sequences and false discovery estimates. This research was supported by the Pew Charitable Trusts (S. Ramachandran is a Pew Scholar in the Biomedical Sciences), and US National Institutes of Health (NIH) grant R01GM118652 (to S. Ramachandran) and COBRE award P20GM109035. S. Ramachandran is an Alfred P. Sloan Research Fellow and also acknowledges support from National Science Foundation (NSF) CAREER Award DBI-1452622. S. Rong is supported by an NSF Graduate Research Fellowship. A.P.F. was supported by an REU supplement to NSF CAREER Award DBI-1452622. E.G.A. was supported by NIH grant K12-GM-102778.

## Author contributions

L.A.S., and S. Ramachandran conceived the study and L.A.S. implemented the methods. Sequence data from the ‡Khomani San were generated and processed by E.G.A., and B.M.H., and L.A.S., E.G.A., B.M.H., and S. Ramachandran contributed to analysis of SWIF(r) results. A.P.F. annotated SWIF(r) results and contributed to comparison of SWIF(r) with existing methods. S. Rong contributed simulations of different modes of selection. L.A.S, E.G.A., B.M.H., and S. Ramachandran wrote the manuscript.

## Additional information

**Competing interests:** The authors declare no competing financial interests.

