## [Peer Review File · Nature Communications]

Reviewer #1 (Remarks to the Author):

The paper describes a composite approach to scan for selective sweeps. I think the paper provides an innovative framework to scan for natural selection in genomes. Genome scans usually relies on a one-dimensional summary statistic and lack of power might result from one-dimensional genome scan. The authors convincingly show that their approach to combine summary statistics is more effective than one-dimensional genome scans and that the existing CMS score partly because they manage to locate more precisely adaptive SNPs. I have only two comments that concern statistical methods.

1. The authors write in different places of the ms that their method is probabilistic which is an advantage because of the natural scale of probabilities. However, to be meaningful probabilities, their returned probabilities should be properly calibrated. For instance, a well-calibrated classifier should classify SNPs such that among the SNPs to which it gave a probability close to 0.8 to be adaptive, approximately 80% of these should be adaptive. I think that the current approach might not be properly calibrated and recalibration techniques exist and should be considered (see eg. <http://scikit-learn.org/stable/modules/calibration.html>). For instance, the paper entitled "Ancestry Composition: A Novel, Efficient Pipeline for Ancestry Deconvolution" (<http://www.biorxiv.org/content/biorxiv/early/2014/10/18/010512.full.pdf>) uses isotonic regression to recalibrate local ancestry probabilities.

2. The authors use ROC curves for method comparisons. I do not think it is a good idea because for selective scans the number of neutral SNPs is several orders of magnitude larger than the number of adaptive SNPs. Let say you have 100 adaptive SNPs and 100,000 neutral ones, a type I error of 5% and a power of 90% (SWIF performance in Figure 1A) would correspond to 90 true positives and 5,000 false positives! That is clearly not a relevant point. The authors should instead consider a Power-FDR curve, which is equivalent to the precision-recall curve used in machine learning.

Sincerely,
Michael Blum

Reviewer #2 (Remarks to the Author):

Review of Sugden et al

In this manuscript the authors join the current cottage industry of using supervised machine learning methods for detecting selection in population genomic data. Briefly the authors slightly tweak vanilla AODE classifiers to use a feature vector of 5 population genetic summary statistics (or 4 for the empirical analysis) to differentiate incomplete (more on this in a moment) hard sweeps from neutral regions of the genome. They show that their method, SWIFr, has better power and accuracy than univariate summaries and CMS. They then apply the method to unpublished data from an KhoeSan population and tell stories about metabolic adaptation.

Frankly the best part of this paper is the name of the method (well done). The method itself represents an incremental advance for the field but one that is presented with such bluster that it is well suited for the age of Trump. For instance the authors claim in lines 10-11 that less attention has been given to "exploiting multiple genomic signatures [...] in a rigorous analytic framework." This statement basically ignores the recent advances in supervised machine learning in population genetics including methods such as evoNet (Sheehan and Song 2016), SHIC (Schridder and Kern 2016), EvoBoosting (Lin et al. 2011), SFSelect (Ronen et al. 2013), the hierarchical boosting method of Pybus et al (2015), and PreCIOSS (2016) to name a few. Perhaps the authors are referring to the thin distinction between window vs. SNP centered methods? If so this is barely

enough of a difference to downplay all of the powerful, related methods that have been developed thus far and may very well outperform SWIFr (more on this later).

Another example of such incredible hype in this paper is one of the supposed major motivations for this new method, repeated again and again but first mentioned in lines 25-26 is that the other ML methods “such as random forest classification, regression, and support vector machines [...] cannot operate with missing data.” This is patently false. For instance for RF, Breiman laid out specifically how to deal with missing data from the very start of the method (i.e. https://www.stat.berkeley.edu/~breiman/RandomForests/cc_home.htm) . Similarly evoNet, SFSelect, SHIC, and EvoBoosting can all handle missing data through training with appropriate matched masked simulations—and the authors perform this in the papers. The authors here also have a way of handling missing data, but this is not unique to their method despite their claims.

Major issues:

- 1) The authors have not compared the performance of SWIFr to appropriate competing methods. Despite their claim from line 38 that they assess SWIFr in comparison to “state-of-the-art univariate and composite methods” the authors have ignored all of the modern ML methods listed above and their claim here is laughable. A proper exploration of their method would compare at the very least to evoNet, SFSelect, SHIC, EvoBoosting, PreCIOSS, and also the new algorithm from Bafna, iSAFE (<http://www.biorxiv.org/content/early/2017/06/02/139055>). Moreover the authors have instead chosen to focus on a Naïve Bayes Classifier that they whipped up for comparison—who cares about this method? What the authors present here is simply not adequate.
- 2) The manuscript is very confused in its use of the term “complete sweeps.” I believe what the authors mean is complete in the sense that the allele frequency has gone to frequency one in a single deme, not the whole population. This is not the canonical use of the term and thus the authors must fix this throughout the manuscript.
- 3) The demographic models used for training, testing, and benchmarking of SWIFr are extremely dated and no longer appropriate. Neither the Shaffner et al model nor the Gronau et al. model incorporate the strengths of population growth now thought to characterize human demographic history. These should be updated to more modern estimates of population size history (see for instance Auton et al). A new demographic model will need to be used for all simulations for training, performance characterization, and eventual application.
- 4) The eventual application is on an unpublished dataset—this is not appropriate unless the authors wish to fold all of the details of that dataset and release it herewith. The authors should apply the method to an already published human data set instead.
- 5) The simulations done in Figure 1 and associated supplemental figures are way too limited with respect to the parameter ranges considered. The authors are focusing on very recent sweeps only, and for undisclosed values of $\alpha = 2N_s$. The authors should simulate under a range of α , preferably between 10^2 - 10^4 so that the results can be put in their proper context. Additionally the authors should vary recombination and mutation rates. Also see above for comments on demographic model.
- 6) The authors need to do simulations with background selection to see how that affects the method. It's crazy that in 2017 this hasn't been done.
- 7) What the heck is going on with the empirical false positive rates in Supplemental tables 3 and 6? It appears that using the 50% threshold (as the authors report they are doing on lines 129) the false positive rates are way too high to believe any of the results. Are the authors really trying to publish a paper with FDR ~56%?

More specific issues:

- 1) In the abstract the authors claim SWIFr “explicitly models demography.” It doesn't. It trains on simulations that have a specified demographic model.

- 2) Lines 1 -8. This introductory paragraph is incredibly thin on scholarship. Further “adaptive sweeps” are not common vernacular—the authors perhaps mean “selective sweeps.” This needs to be fixed considerably.
- 3) Line 15 – the authors say other ML methods combine statistics into “one score.” Actually this usually isn’t the case. Instead most of these methods yield a vector of probabilities indicating class membership.
- 4) Line 32—I believe only CMS “requires comparison with a genome-wide distribution.” None of the modern ML methods needs this.
- 5) Lines 61-62 – if missing data is handled via removing a component of the feature vector the authors need to provide evidence that power/specificity isn’t affected. No such evidence is provided. This should be provided for each summary statistic.
- 6) Line 85 – unclear why the authors are presenting a Naïve bayes classifier here. The authors should focus on competing, published methods (See above).
- 7) Lines 100-101 – we have known for a long time that combining stats using machine learning is better than individual summaries of the data. PLEASE ACKNOWLEDGE THE LITERATURE.
- 8) Figure 1 legend—the authors say they are using an AODE? The whole paper is premised around this Bayesian tweak on AODE—what’s happening here?
- 9) The soft sweep results presented in Supp Fig 17 are unconvincing as the beneficial allele is starting at a very low frequency in these simulations ($f=0.02$) and thus should be equivalent to a hard sweep (See Jensen 2014 for a discussion of this issue). A much more serious treatment of soft sweeps is needed here, including simulating over a range of f and a range of final frequencies. Also the authors should provide further simulations of incomplete sweeps where the final frequency is something different that $p=0.2$ as presented in figure 1.
- 10) Lines 120-122 – the authors claim “we find that modeling the [...] SWIFr is essential for correctly localizing these known adaptive mutations.” That’s a seriously specious claim given that every other method found these regions too and that in figure 2C and 2E other SNPs have higher sweep probabilities than SWIFr. Again, this review is taken aback by the hype the authors are spinning here.
- 11) Line 127 – the authors identify 126 genes in the genome. This is a very low number given recent efforts to localize selection in the human genome (e.g. Schrider and Kern 2017). How do the authors square this? Moreover 63% overlap with previous sweep scans is pretty bad.
- 12) Robustness to misspecification section—the authors should present a case of really bad misspecification – no population growth vs the Auton model or similar. Other methods like SHIC and Evolboosting have been shown to be robust to misspecification – how does SWIFr compare to those methods in this regard?
- 13) Figure 2 legend – why are the authors highlighting rs3827760 in fig 2E? Other neighboring SNPs have much higher sweep probs.
- 14) Lines 161-163 – the authors state “genomic regions identified by SWIFr contain annotated functional mutations identified in high coverage...” what does this mean? That in this unpublished manuscript they found a nonsynonymous SNP? So what? Also as stated above the data from Martin et al must be released for this paper to be publishable as is. Alternatively the authors could use a different data set.
- 15) Lines 169-170 – finding a lot of SWIFr signals in the MHC region is not a good thing—that region is notoriously hard to align, assemble, etc. The onus is on the authors to demonstrate that the data quality in that region is half way decent.
- 16) Line 175 – the authors state “genes related to adiponectin...” they mean SNPs right? Not genes. The methods is SNP-wise.
- 17) Table 1 indicates that the only functional category that survives FDR correction is adiponectin. The authors should remove all of the discussion of non-significant functional categories from the paper.
- 18) Line 183—the authors have belabored the point that combining signals from multiple statistics aids power and that they don’t rely on empirical distributions—why are they now giving empirical p-values from individual features???
- 19) Lines 196-197—this makes no sense with respect to the paper. The authors have gone on about how accurate there method is, but now are pointing to a random nonsynonymous SNP 1kb

away from a SWIFr identified SNP. Sorry this is bait and switch.

20) Line 233 – the claim that SWIFr “localizes experimentally validated adaptive mutations using genomic data along” belies what the authors are actually doing in much of the results. See comment above. This sentence should be softened considerably.

21) Line 227—the algorithm presented is not serious or noteworthy. This should be removed from the discussion.

22) Lines 229-231—only one function survives the correction for multiple tests. This statement is overblown.

23) Line 235 – the statement that SWIFr advances the methodologies used in other ML approaches by learning the joint distributions of features is untrue—this is a feature of the other methods too. Again, compare your method to those before blowing your own horn.

24) Line 238—the authors need to demonstrate that other methods are biased in their handling of missing data before making this claim. They have not done so. Further the authors have not demonstrated the robustness of their own method to missing data and need to do so via simulations.

25) This brings me to another point—the authors have engaged in a major bait and switch in this paper—they benchmark the method using a feature vector of 5 stats, but only analyse the data using 4. The entire simulation section of the paper should be repeated to reflect what they have done in the empirical analysis.

26) Line 239 – how is SIDT2 reflective of “finer localization?” Because their ad hoc, unpublished naïve bayes classifier missed it?!?

27) Line 242—ABC does not have likelihoods. That’s the whole point.

28) Line 244—where is the evidence that the Peter et al. method is prone to overfitting? This is just a baseless claim.

29) Line 244—again the authors say they are using AODEs...

30) Line 248-249 – this bit about “cryptic assumptions about class sizes” belies the authors misunderstanding of supervised ML. None of this is true, and this section should be removed.

31) Line 288—again this baseless claim of other methods not being able to deal with missing data.

32) Line 303—100 simulations is way too small for accurate training. The authors should increase this by \sim and order of magnitude.

33) Lines 327-329—how were these migration rates arrived at? What does a pulse migration with a rate mean? How was the “match” evaluated quantitatively? This seems very ad hoc as presented.

34) What do the “full covariance matrices” mean in the context of the Gaussian mixture model? This section is quite unclear.

35) ROC analysis—the authors should vary the sweep call threshold, not the prior on a sweep here.

36) Supplemental Figure 18 – the authors say that the method can be extended but the results presented look terrible. This isn’t convincing.

Reviewer #3 (Remarks to the Author):

Alpert Sugden et al. develop a novel statistical learning approach for localizing selective sweeps, which they term SWIF(r). In contrast to other recent statistical learning approaches (such as that of S/HIC of Schrider and Kern) that also employ multiple summary statistics as evidence for sweeps, the SWIF(r) approach developed here can provide a score per site rather than per region, enabling it to localize sweeps at the polymorphism level. In addition, this new method is robust to missing data, whereas prior statistical learning approaches required extensive and careful data filtering so as to not bias their results. The authors also apply their new statistic to data from the Khomani San, and characterize a number of regions.

I would like to start out by stating that the manuscript is very well-written, and that I appreciate that the authors included line numbers to make refereeing easier. In addition, I believe that developing statistical learning approaches for localizing adaptive alleles is a hot topic, and the proposed method will likely be of interest to a number of researchers. I am less able to assess the novelty of the empirical findings, but the methodology presented is certainly both innovative and rigorous. Therefore, based solely on the proposed statistical approach, I believe this manuscript is high priority.

Some of the points I find particularly interesting are that each polymorphic site is assigned a posterior probability of being a sweep, that the authors investigated how ascertainment bias could be accounted for (often ignored these days, but important as SNP chips are still common), and that their method can detect both partial hard sweeps to low frequency (e.g., 20%) and soft sweeps from standing variation (though only hard sweeps were used in training). Below are my additional comments and/or concerns of the current manuscript.

Major comments:

(1) The scan in the Khomani San shows a strong peak in the MHC region. This region is classically thought to be undergoing balancing selection, and a number of approaches for detecting long-lived balancing selection are able to detect the HLA genes within this region. Long-lived balancing selection, which tends to yield increased genetic diversity and decreased population differentiation (F_{st}), appears to be incompatible with signatures of selective sweeps. It would be helpful if the authors could discuss why this region would be identified by methods for detecting selective sweeps. Could it be that the SNPs that are identified by SWIF(r) have low scores for methods that detect balancing selection (i.e., these high probability SNPs would not be detected by methods to identify balancing selection)? Or is there something specific happening in the Khomani San population, which would indicate that this locus should be evolving under recent positive selection rather than long-lived balancing selection?

(2) If single nucleotide statistics (such as F_{st} and Delta DAF) were not used, and instead haplotype-based statistics (such as iHS and Delta EHH) were only used, would correcting for ascertainment matter much? Conrad et al. (2006; Nat Genet 38:1251-1260) indicated (Figure 3 of their article) that the effect of ascertainment bias is removed when using haplotype-based statistics.

(3) The authors indicate that they expect that SWIF(r) is robust to background selection, because Enard et al. demonstrated that haplotype-based statistics are robust to background selection. However, two of the statistics, and indeed those with the most power (F_{st} and Delta DAF from what I can tell from Figure 1D and Supplementary Figure 17) are not haplotype-based, and therefore likely influenced by background selection. In addition, based on comment 2 above, the argument that haplotype-based statistics are robust to background selection can also then be used to argue that SWIF(r) is robust to ascertainment bias. Because background selection is a ubiquitous force (e.g., McVicker et al. 2009, Comeron 2014), it would be helpful if the authors tested their method's robustness to this scenario through some illustrative simulations.

Minor comments:

- Citations of figure panels within the text use uppercase letters (e.g., Figure 1A or Figure 1B), whereas the panels are labeled using lowercase letters (e.g, a and b) within each figure.

- On page 11, line 232 the authors write "The composite classification framework of SWIF(r) quantitatively grounds a common qualitative approach used in scans for adaptive sweeps". Though I understand what the authors are getting at, likelihood ratio approaches essentially do to the same thing, as the likelihood ratio quantitatively assess the probability of observing the data under the alternative hypothesis of selection relative to the probability of observing the data under the null hypothesis of neutrality. I think the authors mean that the qualitative approach is common when using summary statistics, such as the component statistics used by SWIF(r).

Reviewer comments and responses:

Reviewer #1 (Remarks to the Author):

The paper describes a composite approach to scan for selective sweeps. I think the paper provides an innovative framework to scan for natural selection in genomes. Genome scans usually relies on a one-dimensional summary statistic and lack of power might result from one-dimensional genome scan. The authors convincingly show that their approach to combine summary statistics is more effective than one-dimensional genome scans and that the existing CMS score partly because they manage to locate more precisely adaptive SNPs. I have only two comments that concern statistical methods.

We thank the reviewer for taking the time to read and critique our manuscript, and for contextualizing the contribution of our framework for genomic scans for selection. We detail our response to the specific comments below.

1. The authors write in different places of the ms that their method is probabilistic which is an advantage because of the natural scale of probabilities. However, to be meaningful probabilities, their returned probabilities should be properly calibrated. For instance, a well-calibrated classifier should classify SNPs such that among the SNPs to which it gave a probability close to 0.8 to be adaptive, approximately 80% of these should be adaptive. I think that the current approach might not be properly calibrated and recalibration techniques exist and should be considered (see eg. <http://scikit-learn.org/stable/modules/calibration.html>). For instance, the paper entitled “Ancestry Composition: A Novel, Efficient Pipeline for Ancestry Deconvolution” (<http://www.biorxiv.org/content/biorxiv/early/2014/10/18/010512.full.pdf>) uses isotonic regression to recalibrate local ancestry probabilities.

*We thank the reviewer for this comment, and agree that the interpretation of SWIF(r)'s reported probabilities requires more detailed explanation. Following the reviewer's suggestion, we have implemented an isotonic regression scheme for applying SWIF(r) to data, and have incorporated that scheme into our analyses of data from the 1000 Genomes Project and from the ‡Khomani San. Text detailing our implementation is excerpted below, (**Results > Implementation of SWIF(r) > lines 88-102**); **Supplementary Figures 1-3** show results from calibration and illustrate the smoothed isotonic regression scheme we implemented to maintain the strict monotonicity of the posterior probabilities reported by SWIF(r) post-calibration. **Figures 2 and 3 and Table 1** in the main text, manhattan plots in **Supplementary Figures 17-19 and 24**, and full results in **Supplementary Tables 2, 4, 6, and 7** have all been updated to reflect the calibrated sweep probabilities. We have added text to the **Discussion (lines 332-344)**, excerpted below, and have added code to the SWIF(r) repository (<https://github.com/ramachandran-lab/SWIFr>) to implement our calibration scheme for any dataset.*

Results > Implementation of SWIF(r) > lines 88-102:

Calibration of posterior probabilities calculated by SWIF(r). A desirable property of probabilities, like those calculated by SWIF(r), is that they be well calibrated: in this context, for the variant positions where the posterior probability reported by SWIF(r) is around 60%, approximately 60% of those sites should contain an adaptive mutation, and approximately 40% should be neutral. We implemented a smoothed isotonic regression scheme to calibrate the probabilities calculated by SWIF(r) (Online Methods). Briefly, when applying SWIF(r) to a given dataset, we calculate the empirical frequencies of neutral and sweep variants that are assigned posterior probabilities between 0 and 1 in simulation, and use isotonic regression (Zadrozny and Elkan, ACM 2002) to map the posterior probabilities to their corresponding empirical sweep frequencies (Supplementary Figure 1, Supplementary Figure 2). We then impose a smoothing function that prevents multiple posterior probabilities from being mapped to the same calibrated value (Supplementary Figure 3, Supplementary Figure 1E, Supplementary Figure 2E). This calibration procedure relies on the relative makeup of the training set; a classifier that is calibrated for a training set made up of neutral and sweep variants in equal parts would not be well-calibrated for a training set in which sweep variants only make up 1% of the whole. For each application of SWIF(r) in this study, we calibrated SWIF(r) for a specific training set makeup (Online Methods; see also Supplementary Figure 1 and Supplementary Figure 2).

Discussion, lines 332-344:

In order for the class probabilities reported by SWIF(r) to be practically interpretable, we calibrated SWIF(r), such that $k\%$ of variants with a posterior sweep probability of $k\%$ are indeed sweep variants. We have implemented a calibration scheme based on isotonic regression for SWIF(r) that maps the posterior sweep probabilities to their empirical sweep proportions in simulated data (Supplementary Figure 1, Supplementary Figure 2, Supplementary Figure 3), but importantly, this calibration relies on the composition of the training set used. While for some classifiers, the proportions of classes are known, or can be reliably estimated (e.g. see Durand et al. biorxiv 2014 and Scheet and Stephens, American Journal of Human Genetics 2006), the proportion of sites throughout the human genome that are adaptive is unknown. For calibrating SWIF(r), we chose training sets made up overwhelmingly of neutral variants; while our calibration of SWIF(r) always preserves the rank order of posterior probabilities (Supplementary Figure 3), the specific choice of training set makeup can have a dramatic effect on the calibration. Therefore, a direct interpretation of the posterior probabilities reported by SWIF(r), or any other classifier that calculates probabilities, must incorporate knowledge of the scenarios used for training and calibration.

2. The authors use ROC curves for method comparisons. I do not think it is a good idea because for selective scans the number of neutral SNPs is several orders of magnitude larger than the number of adaptive SNPs. Let say you have 100 adaptive SNPs and 100,000 neutral ones, a type I error of 5% and a power of 90% (SWIF performance in Figure 1A) would correspond to 90 true positives and 5,000 false positives! That is clearly not a relevant point.

The authors should instead consider a Power-FDR curve, which is equivalent to the precision-recall curve used in machine learning.

*We thank the reviewer for this comment, and agree that the Power-FDR curve offers important information for the performance comparison of SWIF(r) to other sweep detection methods. We note that the shape of these curves depends on the composition of the training set, so we have added **panel E to Figure 1** in the main text that plots the Power-FDR curves for the same training set composition that we use for calibrating SWIF(r) for application to the ‡Khomani San dataset. We have also added **Supplementary Figure 11**, which plots these curves for a range of training set compositions. We further include a panel with Power-FDR curves for each of our benchmarking comparisons, added based on reviewer 2's comments (**Supplementary Figures 6F** for SweepFinder, **9F** for evolBoosting, and **10F** for evoNet). Depending on the composition of the training set, SWIF(r) performs comparably or better than CMS, evolBoosting, and evoNet, and dramatically outperforms the component statistics and SweepFinder, and achieves an FDR near zero for a true positive rate of ~35%. The curves do, however, illustrate the difficulty in detecting evidence genome-wide for adaptation at the SNP level, as very small false positive rates can translate into substantial false discovery rates. This is an important point to make, in light of the number of selection scans that are performed genome-wide. We also note that this will be an issue for window-based methods as well, although with a different manifestation: the false discovery rate for window classification may be smaller (as we show for evolBoosting, evoNet, and a window-based version of SWIF(r) in **Supplementary Figures 9F and 10F**), but a classified window contains a large number of variants. Therefore, these methods will likely not narrow down the list of putative SNP targets. We have added the following text at the end of **Results > Performance of SWIF(r) using simulated data (lines 154-171)**:*

*While ROC curves are informative for illustrating the performance of different sweep detection methods, it is important to note that the genome has far more neutral variants than adaptive mutations. Therefore, more relevant performance comparisons can be made by illustrating the predicted false discovery rate (FDR) for a given true positive rate using Power-FDR curves. These curves depend on the composition of the training set, since the false discovery rate rises as the proportion of adaptive variants in the training set decreases. In **Figure 1E** and **Supplementary Figure 6F**, we plot Power-FDR curves for all methods using the same training set composition we use for calibration of SWIF(r) for application to the ‡Khomani San dataset (99.95% neutral variants and 0.05% adaptive variants). Curves for other training set compositions can be found in **Supplementary Figure 11**. Power-FDR comparisons between SWIF(r) and evolBoosting and evoNet can be found in **Supplementary Figure 9F** and **Supplementary Figure 10F** assuming that 1% of windows contain a sweep. SWIF(r) performs well relative to its component statistics, CMS, and SweepFinder, however, these analyses illustrate the inherent difficulty of site-by-site detection of adaptive mutations. Because there are so many more neutral variants than adaptive variants in the genome, even a small false positive rate can result in a substantial false discovery rate. Window-based methods, including window-based SWIF(r), may appear to have lower false discovery rates (since*

there are many fewer windows than variants, and thus fewer opportunities for false positives to arise; see Supplementary Figure 9 and Supplementary Figure 10), but this comes at the cost of a longer list of putative SNP targets, since each classified window contains a large number of individual variants.

Sincerely,
Michael Blum

We thank the reviewer for signing his review and acknowledge him by name in our Acknowledgments section; his comments improved the manuscript and method.

Reviewer #2 (Remarks to the Author):

Review of Sugden et al

In this manuscript the authors join the current cottage industry of using supervised machine learning methods for detecting selection in population genomic data. Briefly the authors slightly tweak vanilla AODE classifiers to use a feature vector of 5 population genetic summary statistics (or 4 for the empirical analysis) to differentiate incomplete (more on this in a moment) hard sweeps from neutral regions of the genome. They show that their method, SWIFr, has better power and accuracy than univariate summaries and CMS. They then apply the method to unpublished data from an KhoeSan population and tell stories about metabolic adaptation.

Frankly the best part of this paper is the name of the method (well done). The method itself represents an incremental advance for the field but one that is presented with such bluster that it is well suited for the age of Trump. For instance the authors claim in lines 10-11 that less attention has been given to “exploiting multiple genomic signatures [...] in a rigorous analytic framework.” This statement basically ignores the recent advances in supervised machine learning in population genetics including methods such as evoNet (Sheehan and Song 2016), SHIC (Schrider and Kern 2016), EvolBoosting (Lin et al. 2011), SFSelect (Ronen et al. 2013), the hierarchical boosting method of Pybus et al (2015), and PreCISS (2016) to name a few. Perhaps the authors are referring to the thin distinction between window vs. SNP centered methods? If so this is barely enough of a difference to downplay all of the powerful, related methods that have been developed thus far and may very well outperform SWIFr (more on this later).

Another example of such incredible hype in this paper is one of the supposed major motivations for this new method, repeated again and again but first mentioned in lines 25-26 is that the other ML methods “such as random forest classification, regression, and support vector machines [...] cannot operate with missing data.” This is patently false. For instance for RF, Breiman laid out specifically how to deal with missing data from the very start of the method (i.e. https://www.stat.berkeley.edu/~breiman/RandomForests/cc_home.htm). Similarly evoNet, SFSelect, SHIC, and EvolBoosting can all handle missing data through training with appropriate

matched masked simulations—and the authors perform this in the papers. The authors here also have a way of handling missing data, but this is not unique to their method despite their claims.

We thank the reviewer for the close reading of our manuscript and for drawing attention to multiple issues that required clarification on our part, as well as additional necessary benchmarking experiments. We have made serious efforts to address all of this reviewer's concerns in the revised manuscript and present our revisions in detail following each point the reviewer outlines below.

As an overview of this revised manuscript and our new analyses and text in response to this reviewer, we now additionally benchmark SWIF(r) against site-based statistic SweepFinder and two supervised machine-learning sweep detection methods, evolBoosting and evoNet, and show that SWIF(r) outperforms these approaches (Supplementary Figures 6, 9 and 10). We have also added text in multiple places to acknowledge existing machine-learning methods for sweep-detection, both SNP- and window-based, and the contribution SWIF(r) makes in the context of these existing methods. We would like to emphasize, however, that the distinction between localizing putative adaptive SNPs versus identifying regions containing adaptive SNPs is not a “thin” one: the window sizes used by these methods range from 20kb (evoNet) to 200kb (S/HIC), and SWIF(r)'s ability to be applied site-by-site generates hypotheses for localizing functionally adaptive variants (as illustrated in Figures 2 and 4). We believe that SWIF(r) and other SNP-based sweep detection methods can enable new insight into selective sweeps by localizing adaptive variants and therefore we emphasize this distinction. That said, in the revised manuscript we demonstrate how SWIF(r) can be altered to assess evidence of sweeps at various window sizes (Supplementary Figures 9 and 10; Online Methods section “Implementation of window-based methods”). A summary of our new benchmarking experiments can be found in Results > Performance of SWIF(r) using simulated data (lines 138-153).

Before proceeding to directly address the reviewer's comments individually, we also clarified in the revised manuscript that our definition of “missing data” for SWIF(r) in our original submission referred to sites at which one or more selection statistics is undefined, and not sites for which genomic data are unavailable. We have now clarified this in the manuscript by referring to “undefined component statistics” instead of “missing data”. In the revised manuscript, we examine how undefined component statistics affect SWIF(r)'s performance (Supplementary Figure 33), an analysis we did not find in the machine-learning sweep-detection methods the reviewer highlighted (evoNet, SFSelect, S/HIC, and EvoBoosting); we would appreciate being pointed towards such analyses. We do note that the issue of undefined statistics may not arise when using window-based component statistics, as the most common undefined statistics are site-based iHS and XP-EHH. We have removed all claims about how other methods, save for CMS, deal with this problem, but we maintain that SWIF(r)'s ability to compute the probability of a sweep at sites with undefined statistics is a noteworthy feature worth highlighting in this manuscript.

Major issues:

1) The authors have not compared the performance of SWIFr to appropriate competing methods. Despite their claim from line 38 that they assess SWIFr in comparison to “state-of-the-art univariate and composite methods” the authors have ignored all of the modern ML methods listed above and their claim here is laughable. A proper exploration of their method would compare at the very least to evoNet, SFSelect, SHIC, EvoBoosting, PreCI OSS, and also the new algorithm from Bafna, iSAFE (<http://www.biorxiv.org/content/early/2017/06/02/139055>). Moreover the authors have instead chosen to focus on a Naïve Bayes Classifier that they whipped up for comparison—who cares about this method? What the authors present here is simply not adequate.

We thank the reviewer for this comment. As described in our overview above, we have now benchmarked SWIF(r) against SweepFinder (Figure 1 and Supplementary Figure 6), CMS (Figure 1), evolBoosting (Supplementary Figure 9), and evoNet (Supplementary Figure 10), and find that SWIF(r) outperforms all of these methods over a range of sweep parameters. The results of these benchmarking experiments can be found in Results > Performance of SWIF(r) using simulated data. The paragraph in which we summarize the results of benchmarking against the other composite methods is excerpted below (lines 138-153):

While few composite methods for sweep detection operate site-by-site, there are a handful of machine-learning composite approaches that identify genomic windows containing adaptive mutations (Lin et al. Genetics 2011; Pybus et al. Bioinformatics 2015; Schrider and Kern, PLoS Genetics 2016; Sheehan and Song, PLoS Computational Biology 2016). In order to compare SWIF(r) against such methods, we had to alter SWIF(r) to calculate window-based sweep probabilities; there are many potential ways to do this that may be differentially powerful, and here we chose simply to use the highest probability assigned to any variant within a given genomic window as the probability for that window. We compared window-based SWIF(r) to two state-of-the-art composite window-based methods: evolBoosting (Lin et al. Genetics 2011), which combines 120 statistics using boosted logistic regression, and evoNet (Sheehan and Song, PLoS Computational Biology 2016), which was developed to jointly infer demography and selection using a deep learning framework (Online Methods). Since both SWIF(r) and evoNet are frameworks that are designed to incorporate any set of statistics, we implemented evoNet to use the same statistics as SWIF(r). When comparing SWIF(r) with evolBoosting, we used 40kb windows following Lin et al. (Genetics, 2011), and show that SWIF(r) outperforms evolBoosting across a range of sweep parameter values (Supplementary Figure 9). For comparison with evoNet, we used 100kb windows following Sheehan et al. (PLoS Computational Biology, 2016) (Supplementary Figure 10). We find that SWIF(r) performs similarly to or better than evoNet in this implementation, although we note that this analysis likely downplays the strengths of both methods, as each method has been altered from its original design to enable a direct comparison.

We have removed all comparisons to the Naive Bayes Classifier, which we originally implemented to illustrate the gains made over CMS when computing a true posterior sweep probability. We agree with the reviewer that, given the new benchmarking we have done, this comparison is rendered unnecessary.

SWIF(r) is intended to be a flexible framework that can incorporate any number of statistics, and our released software (<https://github.com/ramachandran-lab/SWIFr>) is implemented with this flexibility in mind. The four summary statistics we use in the implementation of SWIF(r) in this paper were done for direct comparison against CMS and because they can all be calculated site-by-site (as the reviewer suggested, all applications of SWIF(r) now consistently exclude ΔiHH throughout the manuscript). As new statistics are developed, leveraging joint likelihoods between those statistics and others will improve classification (as we show when incorporating iSAFE, delineated in a biorxiv preprint <https://www.biorxiv.org/content/early/2017/10/01/139055>, into SWIF(r) in **Supplementary Figure 29, Discussion lines 327-329**). Methods such as evolBoosting and CMS do not have this flexibility; we maintain that the flexible ability to incorporate new selection statistics will make SWIF(r) more valuable as the field moves forward.

The set of methods we benchmark against in this revised manuscript represents a range of state-of-the-art methods for sweep detection. SWIF(r) in the form implemented in this study was not intended to be applied to soft sweeps (in comparison to S/HIC), and existing sweep detection methods vary in their usability and documentation. We hope these new experiments address the reviewer's concerns regarding the performance of SWIF(r) relative to appropriate competing methods.

2) The manuscript is very confused in its use of the term "complete sweeps." I believe what the authors mean is complete in the sense that the allele frequency has gone to frequency one in a single deme, not the whole population. This is not the canonical use of the term and thus the authors must fix this throughout the manuscript.

We thank the reviewer for pointing out this needed clarification; in both the main text (**lines 25-26, 37-38, and 132**) and in all relevant figure captions (**Figure 1, Supplementary Figures 6, 8, 9, 10**), we now state that we use "complete sweeps" to refer to sweeps where the beneficial mutation has reached fixation in the population of interest.

3) The demographic models used for training, testing, and benchmarking of SWIFr are extremely dated and no longer appropriate. Neither the Shaffner et al model nor the Gronau et al. model incorporate the strengths of population growth now thought to characterize human demographic history. These should be updated to more modern estimates of population size history (see for instance Auton et al). A new demographic model will need to be used for all simulations for training, performance characterization, and eventual application.

We thank the reviewer for this comment. We agree that it is important to demonstrate that SWIF(r) is robust to demographic misspecification involving exponential population growth. To demonstrate this, we simulated a new dataset using a demographic model from Gravel et al.

(PNAS, 2011) that incorporates recent exponential population expansion with rates of 0.38% and 0.48% for Europe and East Asia respectively for the last 23,000 years. We allowed this expansion to last until 5,000 years ago to allow for simulating very recent sweeps, which the simulation software *cosi* cannot simulate simultaneously with demographic change (*discussion and details can be found in lines 186-191 and lines 439-447, excerpted below*). We have added a panel to *Supplementary Figure 13* that illustrates that, when *SWIF(r)* is trained on the Schaffner demographic model and tested on these simulations from the Gravel model, *SWIF(r)* maintains roughly the same power as when *SWIF(r)* is both trained and tested on simulations from the Schaffner model. Because of this, and the fact that we intend *SWIF(r)* to be a generalizable framework that can learn from any set of simulations and scenarios, we have retained our original results, now calibrated (see our response to reviewer 1), from application to the 1000 Genomes dataset and the \ddagger Khomani San dataset.

Results > “Performance of SWIF(r) using simulated data” > “Robustness of SWIF(r) to demographic model misspecification” lines 186-191:

*Both the Schaffner model and the Gronau model, as implemented, do not include very recent population expansion, so we implemented a third demographic model from Gravel et al. (PNAS 2011) that includes exponential population growth within the last 23,000 years. Since *cosi* cannot simulate selective sweeps overlapping with demographic changes, we only simulated sweeps beginning 5kya, allowing for 18,000 years of exponential expansion (Online Methods). In *Supplementary Figure 13*, we show that *SWIF(r)* is also robust to this recent population expansion.*

Online Methods > “Simulation of haplotypes including recent population growth” lines 440-447:

*To test *SWIF(r)*’s robustness to misspecification of recent population growth, we implemented a set of simulations using a demographic model from Gravel et al. (PNAS 2011) that estimates recent exponential population expansions with rates of 0.38% for Europe and 0.48% for East Asia over the last 23,000 years. Since the simulation software *cosi* (Schaffner et al. *Genome Research* 2005) cannot simulate sweeps and population-level changes simultaneously, we allowed the expansion to last from 23,000 years ago to 5,000 years ago, to allow for sweeps beginning 5,000 years ago. We also included the migration rates inferred by Gravel et al. (PNAS 2011) between Europe, East Asia, and Africa, and between Africa and the ancestral population of Europe and East Asia. As in other analyses, We simulated selective sweeps spanning a range of present-day allele frequencies from 20% to 100%.*

4) The eventual application is on an unpublished dataset—this is not appropriate unless the authors wish to fold all of the details of that dataset and release it herewith. The authors should apply the method to an already published human data set instead.

SWIF(r) is applied to (1) publicly available data from the 1000 Genomes (URLs listed in the revised manuscript under “Software and Data availability”, preceding Acknowledgments) and (2) published data (Uren et al. *Genetics* 2016, and Martin et al. in press) from the \ddagger Khomani San.

Below we detail how investigators can access the \ddot{K} homani San data and revisions in the manuscript text that provide these details to readers of this study.

The \ddot{K} homani San exome dataset is described by Martin et al. (in press at Cell, and available as a preprint on bioRxiv <https://www.biorxiv.org/content/early/2017/10/13/200139>). Details regarding mapping and variant calling may be found in Martin et al. (preprint link above) and in Kidd et al. (2014, BMC Genomics) for a partial dataset. Unfortunately, we are unable to make these data freely available due to local community constraints on data sharing by the San Council; please see

<<http://www.sciencemag.org/news/2017/03/san-people-africa-draft-code-ethics-researchers>> and

<<http://trust-project.eu/wp-content/uploads/2017/03/San-Code-of-RESEARCH-Ethics-Booklet-final.pdf>>. Interested parties should contact the South African San Council for research and ethics review (Leana Snyders <leana@yemail.com>; admin@sasi.org.za>).

To make these details clear in the revised manuscript, we changed the URLs section preceding Acknowledgments to be titled “Software and Data availability” and now give details on contact information for the South African San Council in that manuscript section (“**Software and Data availability**”, lines 608-611; text excerpted below).

“ \ddot{K} homani San genotype data were first described by Uren et al. (2016), and \ddot{K} homani San exome data were first described by Martin et al. (in press). Queries regarding access to \ddot{K} homani San data analyzed here should be sent to the South African San Council for research and ethics review by contacting both Leana Snyders (leana@yemail.com) and admin@sasi.org.za.”

5) The simulations done in Figure 1 and associated supplemental figures are way too limited with respect to the parameter ranges considered. The authors are focusing on very recent sweeps only, and for undisclosed values of $\alpha = 2Ns$. The authors should simulate under a range of α , preferably between 10^2 - 10^4 so that the results can be put in their proper context. Additionally the authors should vary recombination and mutation rates. Also see above for comments on demographic model.

We thank the reviewer for pointing out the need to clarify our simulation parameters. The selection strength (s) parameters are provided in **Supplementary Table 5**, and do in fact constitute a range of values for α from ~ 100 to ~ 4500 . In addition, we vary recombination rates; for each simulation, we draw a new recombination map from a hierarchical recombination model that draws realistic regional rates and recombination hotspots, as described in Schaffner et al. (2005). In response to this comment, we have made both of these points more explicit in the text (**Online Methods > “Simulation of haplotypes for 1000 Genomes analysis”, lines 399-401 and lines 405-413**). We believe that varying the mutation rate on top of this large parameter set is beyond the scope of this paper, in which our goal is to present a general framework for hard sweep detection that incorporates training simulations from a given demographic model.

6) The authors need to do simulations with background selection to see how that affects the method. Its crazy that in 2017 this hasn't been done.

We thank the reviewer for this comment, and we agree that it is important to demonstrate that SWIF(r) is robust to background selection; in the revised manuscript we demonstrate this in Supplementary Figure 14, and discuss these results in a new section in the manuscript (Results > "Performance of SWIF(r) using simulated data", lines 192-200 and Online Methods > "Simulation of background selection", lines 448-464). We excerpt these sections below:

Results > "Performance of SWIF(r) using simulated data", lines 192-200:

Robustness of SWIF(r) to background selection. We also assessed the sensitivity of SWIF(r) to background selection by generating a set of training simulations containing neutral regions, selective sweeps, and exonic regions, using forward simulator slim (Messer, Genetics 2013) (Online Methods). We trained SWIF(r) on both neutral and sweep simulations, and tested the ability of SWIF(r) to distinguish between exonic and sweep sites, relative to its ability to distinguish between neutral and sweep sites. We find that SWIF(r) is fully robust in this scenario, meaning that we would not expect background selection in genic regions to result in false positive sweep signals (Supplementary Figure 14). These results are aligned with those of Enard et al. (Genome Research, 2014), who have shown that background selection has little to no effect on haplotype-based statistics iHS and XP-EHH (and in fact makes iHS more conservative).

Online Methods > "Simulation of background selection", lines 448-464

Simulation of background selection

For evaluating SWIF(r)'s robustness to background selection, we generated 3 sets of simulations of 1Mb each using forward simulator slim (Messer, Genetics 2013): neutral regions, regions with a hard sweep, and genic regions. For genic regions, we followed Messer and Petrov (Trends in Ecology and Evolution, 2013) to simulate gene structure: each simulation had one gene with 8 exons of 150bp each, separated by introns of 1.5kb, and flanked by a 550bp 5'UTR and a 250bp 3'UTR. Within exons and UTRs, 75% of sites were assumed to be functional. Mutations were assumed to be codominants, and fitness effects across different sites were assumed to be additive. Functional sites were divided into 40% "strongly deleterious" sites with selection coefficient -0.1, and 60% "weakly deleterious" sites with selection coefficients between -0.01 and -0.0001. The mutation rate was set at 2.5×10^{-8} per site per generation, and the recombination rate at 10^{-8} . Note that for testing the robustness of SWIF(r), we only considered sites from these simulations that landed in exons or UTRs.

For all three sets of simulations, we simulated two populations with $N_e = 5000$, which split from each other 40,000 years ago. For sweep simulations, we drew selection coefficients for the beneficial allele from an exponential distribution with mean 0.03, and sweeps begin 10,000 years ago. Since forward simulations are much more computationally intensive than coalescent simulations, we rescaled our parameters by a

factor of 10 (10 times larger for mutation rate, recombination rate, and selection coefficients, 10 times smaller for population sizes and 10 times shorter for all times) to make the simulations feasible (Messer, Genetics 2013).

7) What the heck is going on with the empirical false positive rates in Supplemental tables 3 and 6? It appears that using the 50% threshold (as the authors report they are doing on lines 129) the false positive rates are way too high to believe any of the results. Are the authors really trying to publish a paper with FDR ~56%?

We thank the reviewer for drawing attention to this point. Following this comment and feedback from reviewer 1 (see our response to point 2 of reviewer 1's comments), we have added FDR analysis to our assessment of SWIF(r)'s performance (Figure 1E, Supplementary Figure 11; see lines 154-170) that shows that SWIF(r) outperforms other site-based methods including CMS in terms of the tradeoff between power and FDR (see also panel F in Supplementary Figures 9 and 10 for comparison of window-based SWIF(r) to evolBoosting and evoNet). These curves also illustrate the difficulty of identifying evidence for selection at the SNP-level genome-wide, as small false positive rates can result in moderate or substantial false discovery rates when the ratio of neutral sites to adaptive sites is extremely high. To our knowledge, no SNP-based sweep-detection methods report FDR rates. We also show that window-based SWIF(r) outperforms evolBoosting and evoNet in FDR analyses (Supplementary Figures 9F, 10F). While FDR rates appear to be lower for window-based methods including window-based SWIF(r), this is a tradeoff that results in less information about localization of the adaptive mutation site.

In the case of Supplementary Tables 4 and 6, we offer two estimates of false discovery using two different definitions of neutrally evolving sequence. The rates provided in the tables define neutral regions as non-conserved and non-coding following Hernandez et al. (Science, 2011), and result in higher false discovery rate estimates; we believe these are very conservative estimates of FDR, as signals at sites with low conservation may well be tagging nearby sweep variants, and intergenic regions contain many regulatory and functional elements. Using another, stricter definition of neutral sequence following Gazave et al. (PNAS, 2014), we see zero false positives in all four populations. We have made these points more explicit in the captions of Supplementary Tables 4 and 6.

More specific issues:

1) In the abstract the authors claim SWIFr “explicitly models demography.” It doesn't. It trains on simulations that have a specified demographic model.

We have changed the abstract to incorporate the reviewer's suggested phrasing: “SWIF(r) is trained using simulations from a user-specified demographic model and explicitly models the joint distributions of selection statistics....”

2) Lines 1 -8. This introductory paragraph is incredibly thin on scholarship. Further “adaptive sweeps” are not common vernacular—the authors perhaps mean “selective sweeps.” This needs to be fixed considerably.

We thank the reviewer for pointing out the need for a more detailed description of the literature. The second paragraph of the introduction now includes a more comprehensive overview of the suite of machine-learning approaches that have been designed for identification of selective sweeps, with citations pointing to evolBoosting, evoNet, S/HIC, SFSelect, and Hierarchical Boosting, and we are more explicit about what we see as the importance and challenges of a sweep-detection method that localizes the site of adaptive mutations (versus identifying windows containing adaptive mutations). In addition, as the reviewer suggested, we have changed “adaptive sweeps” to “selective sweeps” throughout the revised manuscript. The new introductory text is excerpted below (Introduction, lines 11-21 and 27-35).

Introduction, lines 11-21:

Recently, there has been increased focus on developing composite methods for identifying selective sweeps, which combine multiple statistics into a single framework (Pavlidis et al. Genetics 2010; Grossman et al. Science 2010; Lin et al. Genetics 2011; Ronen et al. Genetics 2013; Grossman et al. Cell 2013; Pybus et al. Bioinformatics 2015; Schrider and Kern, PLoS Genetics 2016; Sheehan and Song, PLoS Computational Biology 2016); we refer to the statistics that are aggregated in composite methods such as these as “component statistics”. Most composite methods draw upon machine learning approaches like support vector machines (Pavlidis et al. Genetics 2010; Ronen et al. Genetics 2013) deep learning (Sheehan and Song, PLoS Computational Biology 2016), boosting (Lin et al. Genetics 2011; Pybus et al. Bioinformatics 2015) or random forest classification (Schrider and Kern, PLoS Genetics 2016) in order to identify genomic windows containing selective sweeps. These windows vary in size from 20kb to 200kb, often identifying candidate sweep regions containing many genes (Schrider and Kern, PLoS Genetics 2016; Sheehan and Song, PLoS Computational Biology 2016). One method, the Composite of Multiple Signals or “CMS” (Grossman et al. Science 2010; Grossman et al. Cell 2013), uses component statistics that can be computed site-by-site in pursuit of localizing adaptive variants within genomic windows, but the output from this method cannot be interpreted without comparison to a genome-wide distribution. In addition, CMS must rely on imputation or other methods of compensation when component statistics are undefined, a complication that typically does not arise when using window-based component statistics.

Introduction, lines 27-35:

Here we introduce a Bayesian classification framework for detecting and localizing adaptive mutations in population-genomic data called SWIF(r) (SWEEP Inference Framework (controlling for correlation)). SWIF(r) has three major features that enable genome-wide characterization of adaptive mutations: first, SWIF(r) computes the per-site probability of selective sweep, which is

immediately interpretable and does not require comparison with a genome-wide distribution; second, no imputation or compensation mechanisms are necessary in the case of undefined component statistics; and third, we explicitly learn pairwise joint distributions of selection statistics, which gives substantial gains in power to both identify regions containing selective sweeps and localize adaptive variants. Existing composite methods for selection scans have subsets of these features, but SWIF(r) combines all three in a unified statistical framework.

3) Line 15 – the authors say other ML methods combine statistics into “one score.” Actually this usually isn’t the case. Instead most of these methods yield a vector of probabilities indicating class membership.

We agree with the reviewer that this sentence was unclear. We have changed it to read: “Recently, there has been increased focus on developing composite methods for identifying selective sweeps, which combine multiple statistics into a single framework.” (Introduction, lines 11-12)

4) Line 32—I believe only CMS “requires comparison with a genome-wide distribution.” None of the modern ML methods needs this.

SweepFinder, CMS, and all the univariate selection statistics studied here require comparison with a genome-wide distribution; while we agree that modern ML methods do not require this, their output may not be well calibrated, as SWIF(r) is in this revised manuscript (see our response to point 1 of reviewer 1). In response to this and other comments from this reviewer, we have softened our language as follows:

Introduction, lines 29-35:

SWIF(r) has three major features that enable genome-wide characterization of adaptive mutations: first, SWIF(r) computes the per-site probability of selective sweep, which is immediately interpretable and does not require comparison with a genome-wide distribution; second, no imputation or compensation mechanisms are necessary in the case of undefined component statistics; and third, we explicitly learn pairwise joint distributions of selection statistics, which gives substantial gains in power to both identify regions containing selective sweeps and localize adaptive variants. Existing composite methods for selection scans have subsets of these features, but SWIF(r) combines all three in a unified statistical framework.

5) Lines 61-62 – if missing data is handled via removing a component of the feature vector the authors need to provide evidence that power/specificity isn’t affected. No such evidence is provided. This should be provided for each summary statistic.

We thank the reviewer for pointing out the need to evaluate the robustness of our method in the presence of undefined statistics. We have provided such an analysis in Supplementary Figure

33, where we plot ROC curves for sites with all statistics defined, and the corresponding curves for situations where each of the summary statistics is undefined. We show that for XP-EHH, DDAF, and Fst, the power of the method is not compromised, but that missing values of iHS do result in somewhat lower power. This is unsurprising, since iHS is by far the most sensitive component statistic of the four in the case of incomplete sweeps. We have added text to the manuscript that acknowledges this (*Online Methods > “Implementation of SWIF(r), lines 484-486*).

6) Line 85 – unclear why the authors are presenting a Naïve bayes classifier here. The authors should focus on competing, published methods (See above).

As noted in our response to point 1 under “Major issues” from the reviewer, we have removed mention of the Naive Bayes classifier here and throughout the manuscript.

7) Lines 100-101 – we have known for a long time that combining stats using machine learning is better than individual summaries of the data. PLEASE ACKNOWLEDGE THE LITERATURE.

This sentence was originally intended to provide intuition behind the power that any composite approach gains by combining statistics that are differentially powerful across different scenarios. We have altered the sentence to read: “This underscores the advantage of composite methods for detecting selective sweeps when the parameters of the sweep are unknown (Pavlidis et al. Genetics 2010; Grossman et al. Science 2010; Lin et al. Genetics 2011; Ronen et al. Genetics 2013; Grossman et al. Cell 2013; Pybus et al. Bioinformatics 2015; Schrider and Kern, PLoS Genetics 2016; Sheehan and Song, PLoS Computational Biology 2016).” (Results > “Performance of SWIF(r) using simulated data”, lines 122-124)

8) Figure 1 legend—the authors say they are using an AODE? The whole paper is premised around this Bayesian tweak on AODE—what’s happening here?

We thank the reviewer for pointing out the confusion here. SWIF(r) combines sweep-detection statistics using an AODE, as published by Webb et al. (Machine Learning, 2005). We have replaced “Bayesian” with “averaged” in the title to make this clearer, and do not use the word “Bayesian” anywhere in the manuscript.

9) The soft sweep results presented in Supp Fig 17 are unconvincing as the beneficial allele is starting at a very low frequency in these simulations ($f=0.02$) and thus should be equivalent to a hard sweep (See Jensen 2014 for a discussion of this issue). A much more serious treatment of soft sweeps is needed here, including simulating over a range of f and a range of final frequencies. Also the authors should provide further simulations of incomplete sweeps where the final frequency is something different that $p=0.2$ as presented in figure 1.

We thank the reviewer for bringing attention to this point. We have added a new panel and caption to Supplementary Figure 30 that shows that although the initial frequency of the beneficial mutation in our simulations is low, most of the simulations do maintain multiple origins of the beneficial mutation at the time of sampling, with only 1% of soft sweep simulations “hardening” (i.e. losing all but one mutation origin), indicating that SWIF(r) is indeed detecting a class of soft sweeps. We appreciate the point that sweeps beginning from higher initial frequencies will be more difficult to detect, and we have added a clause to the Discussion that acknowledges the low initial frequency that we used for simulations (lines 346-347) As noted in our overview of responses to the reviewer, a full treatment of soft sweeps is beyond the scope of this manuscript. We now state clearly in the results that our implementation of SWIF(r) was trained on hard sweeps (Results, lines 54-59, excerpted below):

Note that we train SWIF(r) on simulations of hard sweeps (Online Methods); our focus here is not on the relative roles of various modes of selection in shaping observed human genomic variation (for recent treatments on this question see (Schridder and Kern, PLoS Genetics 2016; Hernandez et al. Science 2011; Messer and Petrov, Trends in Ecology and Evolution 2013; Garud et al. PLoS Genetics 2015; Schridder and Kern, Molecular Biology and Evolution 2017), although we note that SWIF(r) is extensible to multi-class classification, and could be used in future applications to explore multiple modes of selection. In this study, our focus is on localizing genomic sites of adaptive mutations that have spread through populations of interest via hard sweeps.

Our simulations do include sweeps with a range of final frequencies (0.2, 0.4, 0.6, 0.8, and 1.0), which we have now made clearer in Online Methods and in lines 405-413. In addition, we now provide ROC curves that assess SWIF(r)’s performance against its component statistics in this implementation, CMS, SweepFinder, evoNet, and evlBoosting for simulated incomplete sweeps across these final allele frequencies (Figure 1, Supplementary Figures 6, 9 and 10).

10) Lines 120-122 – the authors claim “we find that modeling the [...] SWIFr is essential for correctly localizing these known adaptive mutations.” That’s a seriously specious claim given that every other method found these regions too and that in figure 2C and 2E other SNPs have higher sweep probabilities than SWIFr. Again, this review is taken aback by the hype the authors are spinning here.

We agree with the reviewer that the section entitled “SWIF(r) correctly localizes canonical adaptive mutations in humans” required significant rewriting for clarity; here we summarize how this section has been altered in the revised manuscript to address this comment from the reviewer. In our genome-wide scan of YRI, CHB and JPT, and CEU genomes from the 1000 Genomes Project, SWIF(r) reports high sweep probabilities at many SNPs and we report and annotate these in Supplementary Tables 2 and 3. Our focus in this manuscript section and in Figure 2 (see our response to point 13 below) is to highlight SWIF(r)’s ability to statistically identify the specific site of an adaptive mutation. This manuscript section focuses on canonical sweep targets in order to further validate SWIF(r)’s output in genome-wide scans. SWIF(r) does

not simply identify the regions highlighted in **Figure 2** as sweep targets (as the reviewer notes, all methods identify these regions as sweep targets), but reports the highest sweep probability at SNPs that experimental studies (all cited in the text) have identified as functionally adaptive. Such experimental evidence is only available for a few genomic targets of sweeps in the human genome; **Figure 2's legend** now reflects this clearly, and we now explicitly state that rs1426654 in **Figure 2C** has the highest sweep probability reported by SWIF(r) in SLC24A5. As we note in point 13 below, in some regions identified as undergoing sweeps by multiple methods, SWIF(r) offers resolution on specific sites selection may be targeting, offering hypotheses for downstream validation (**Figure 2 B,E,F**). Our specific changes to the main text are excerpted below:

Results > "SWIF(r) correctly localizes canonical adaptive mutations in humans", lines 201-214:
For application to data from phase 1 of the 1000 Genomes Project, we used training simulations from the Schaffner demographic model (Schaffner et al. Genome Research 2005), calibrated SWIF(r) for a training set composed of 0.01% sweep variants and 99.99% neutral variants (Supplementary Figure 1), and applied it to SNP array data from West African (YRI), East Asian (CHB and JPT), and European (CEU) populations. SWIF(r) reports high sweep probabilities at multiple SNPs within known and suspected selective sweep loci in each of these populations (Supplementary Table 2, Supplementary Table 3). Figure 2 illustrates the ability of SWIF(r) to localize sites of adaptive mutations within genomic regions containing canonical sweeps. Adaptive SNPs have been determined via functional experiments in SLC24A5 (Soejima and Koda, International Journal of Legal Medicine 2007), DARC (Reich et al. PLoS Genetics 2005), and HERC2 (Eiberg et al. Human Genetics 2008); we find that modeling the dependency structure among component statistics within SWIF(r) enables statistical localization of these experimentally identified adaptive mutations (Figures 2A,C,D). Methods that treat component statistics as independent, as CMS does, cannot localize these experimentally identified adaptive SNPs (Supplementary Figure 15, Supplementary Figure 16). In CHB and JPT, SWIF(r) recovers a strong adaptive signal in the vicinity of EDAR, offering new hypotheses for targets of selection in this genomic region.

11) Line 127 – the authors identify 126 genes in the genome. This is a very low number given recent efforts to localize selection in the human genome (e.g. Schrider and Kern 2017). How do the authors square this? Moreover 63% overlap with previous sweep scans is pretty bad.

*As stated in our Results, SWIF(r) is trained on simulations of hard sweeps (see **response to point 9 above and Results, lines 54-59**); the SWIF(r) framework is easily extensible to multiple evolutionary scenarios (**Supplementary Figure 30**), but our focus is on localizing adaptive variants in hard sweeps. Therefore, we do not think the number 126 is low, given our focus on hard sweeps.*

In the particular line the reviewer cites in this comment, we have revised the text to read as follows (Results > “SWIF(r) correctly localizes canonical adaptive mutations in humans”, lines 217-219 in the revised manuscript):

Of the 126 genes across these populations with SWIF(r) signals (i.e. at least one variant within the gene has posterior hard sweep probability greater than 50%), 63% were identified in at least one positive selection scan conducted in humans (Supplementary Table 3).

Regarding the overlap of our results with previous selection scans, the literature search in Supplementary Table 3 includes methods applied to different data sets from different population samples over the last decade, that were designed for different purposes (i.e., detecting incomplete sweeps alone versus complete sweeps). We find it difficult to form an expectation of percentage overlap for this broad range of comparisons. Given the new benchmarking we have done, and that our goal in presenting SWIF(r) is to present a method that identifies genomic targets of selection missed by existing methods, we feel we have addressed this comment.

12) Robustness to misspecification section—the authors should present a case of really bad misspecification – no population growth vs the Auton model or similar. Other methods like SHIC and Evolboosting have been shown to be robust to misspecification – how does SWIFr compare to those methods in this regard?

In response to Major Issues #3 above, we simulated a new dataset using a demographic model from Gravel et al. (2011) that includes exponential population growth, where our simulations using the Schaffner mode have no population growth. As mentioned above, a new panel in Supplementary Figure 13, and text in lines 186-191 shows that SWIF(r) is robust to this kind of misspecification.

13) Figure 2 legend – why are the authors highlighting rs3827760 in fig 2E? Other neighboring SNPs have much higher sweep probs.

We thank the reviewer for this question. To clarify, we now note in Figure 2’s legend that rsids and diamonds are used to highlight functionally verified adaptive SNPs; the figure legend details why rs3827760 is classified in the literature as putatively functional. We also state in the following in the revised main text (Results > “SWIF(r) correctly localizes canonical adaptive mutations in humans”, line numbers 213-214):

“In CHB and JPT, SWIF(r) recovers a strong adaptive signal in the vicinity of EDAR, offering new hypotheses for targets of selection in this genomic region.”

In conjunction with our response to point 10 above, we hope this sufficiently addresses the reviewer’s comment.

14) Lines 161-163 – the authors state “genomic regions identified by SWIFr contain annotated functional mutations identified in high coverage...” what does this mean? That in this unpublished manuscript they found a nonsynonymous SNP? So what? Also as stated above the data from Martin et al must be released for this paper to be publishable as is. Alternatively the authors could use a different data set.

We thank the reviewer for pointing out the need to clarify our results; as stated earlier in response to Major Issues #4, for our application of SWIF(r) to the ‡Khomani San, we used SNP array data described in Uren et al. (2016). We then validate that SWIF(r) signals tag functional SNPs using exome sequencing data for the same individuals in the SNP array dataset; these data are described by Martin et al. (in press at Cell; see bioRxiv preprint: <https://www.biorxiv.org/content/early/2017/10/13/200139XXX>). Please see our earlier response to Major Issues #4 regarding data release.

*The results in **Figure 3** were generated by applying SWIF(r) to SNP array data from the ‡Khomani San. The Illumina OmniExpress SNP array used here is relatively sparse (only containing ~700,000 autosomal SNPs); in order to validate whether the high-probability sweep sites SWIF(r) identified were tagging functional variants, we turned to the exome data from the same individuals to look for novel or highly differentiated mutations not on the SNP array that the SWIF(r) signals might be tagging. In the case of the gene ADIPOQ, the SWIF(r) signal at rs6444174 is the closest possible SNP on the array to the nonsynonymous mutation rs113716447 identified in the exome data (see **lines 274-276 of the revised manuscript and our response to Specific Issues #19 below**).*

15) Lines 169-170 – finding a lot of SWIFr signals in the MHC region is not a good thing—that region is notoriously hard to align, assemble, etc. The onus is on the authors to demonstrate that the data quality in that region is half way decent.

*We thank the reviewer for this pointing out the need for clarity here. We only apply SWIF(r) to SNP array data which contains validated MHC variants and on SNPs which passed normal quality control, ie. no more than 5% missing data at a given locus; we do not carry out any followup in this region using the exome dataset precisely because of potential issues with mapping and phasing. We have made this more explicit in the manuscript (**lines 272-274**). We have also added a paragraph to the **Discussion** that further contextualizes this MHC signal, in response to Reviewer #3’s comments; please see that response as well.*

16) Line 175 – the authors state “genes related to adiponectin...” they mean SNPs right? Not genes. The methods is SNP-wise.

*We thank the reviewer for pointing this out. We have edited the sentence to read “**SNPs in genes related to adiponectin (ADIPOQ, PEPD, DUT, ASTN2) have among the highest posterior sweep probabilities (all ≥ 75%)**” (**lines 250-251 in revised manuscript**).*

17) Table 1 indicates that the only functional category that survives FDR correction is adiponectin. The authors should remove all of the discussion of non-significant functional categories from the paper.

After calibrating the probabilities reported by SWIF(r), two more functional categories survived FDR correction: Body Mass Index and gamma-Glutamyltransferase (Figure 3 and its caption now reflect this). Body Mass Index is also related to metabolism and obesity. Most highlighted genes in Table 1 are found in either the Adiponectin or Body Mass Index gene sets, and all are associated with metabolic and obesity-related phenotypes. For these reasons, and because we present FDR results for this analysis, we have kept our discussion of functional categories as is.

18) Line 183—the authors have belabored the point that combining signals from multiple statistics aids power and that they don't rely on empirical distributions—why are they now giving empirical p-values from individual features???

We appreciate the reviewer's question. In the section of the manuscript the reviewer is referring to, our goal is to determine the biological function of variants with high sweep probabilities in our application of SWIF(r) to \ddagger Khomani San SNP array data. rs6444174 has a high sweep probability as reported by SWIF(r) (92%), and has been identified as associated with adiponectin levels and body mass index in a genome-wide association study of African Americans. We note that this same rsid is an empirical genomic outlier for three univariate selection statistics to both underscore why its SWIF(r) sweep probability is so high, and to illustrate which statistics are driving this signal.

19) Lines 196-197—this makes no sense with respect to the paper. The authors have gone on about how accurate their method is, but now are pointing to a random nonsynonymous SNP 1kb away from a SWIFr identified SNP. Sorry this is bait and switch.

We thank the reviewer for bringing attention to this point. As we mentioned above (see our earlier response to Specific Issue #14), the nonsynonymous SNP is only present in the exome data, and not on the sparse SNP array that was used for identifying SWIF(r) signals in Figure 3. In the case of ADIPOQ, the SWIF(r) signal was the closest possible SNP on the SNP array to the nonsynonymous mutation identified in the exome data. We therefore offer this as evidence for a putative driving mutation that is being tagged by the SWIF(r) signal <1kb away. We have made this point more explicit in the manuscript (Results > "Adaptive loci in the \ddagger Khomani San are enriched for metabolism- and obesity-related genes" > "Exome-based support for targets of selection identified by SWIF(r)", lines 274-276):

In ADIPOQ, we identify a missense mutation, rs113716447, for which the nearest SNP that is present on the SNP array is rs6444174 (less than 1kb away); rs6444174 has a calibrated SWIF(r) sweep probability of 90%, the highest in ADIPOQ (Figure 4).

20) Line 233 – the claim that SWIFr “localizes experimentally validated adaptive mutations using genomic data along” belies what the authors are actually doing in much of the results. See comment above. This sentence should be softened considerably.

*Based on our response to Specific Issues #10, we stand by this claim, which is stated in reference to **Figure 2**. For those canonical sweep targets in the 1000 Genomes where molecular experiments have identified a site containing a functional beneficial mutation, SWIF(r) assigns that site the highest sweep probability in that region. So we respectfully leave this sentence as it was in our original submission, in hopes that the new analyses and text revisions we have made in response to other comments by this reviewer (in particular about the analysis of SNP array and exome data from the †Khomani San) and the other reviewers lend support to our claim.*

21) Line 227—the algorithm presented is not serious or noteworthy. This should be removed from the discussion.

We respectfully maintain that this algorithm is of value, and point to Reviewer 3’s comments that addressing ascertainment bias is important because SNP arrays are still widely used for assaying genetic variation in diverse human populations.

22) Lines 229-231—only one function survives the correction for multiple tests. This statement is overblown.

Given our response to point 17 above, we feel this statement is now accurate and respectfully leave it as is.

23) Line 235 – the statement that SWIFr advances the methodologies used in other ML approaches by learning the joint distributions of features is untrue—this is a feature of the other methods too. Again, compare your method to those before blowing your own horn.

*We feel this has been addressed with new benchmarking in our revised manuscript, and with the new language that we added in response to Specific Issues #4 above. As described in response to Major Issues #1 above, we now compare SWIF(r) to a broader array of state-of-the-art methodologies including evolBoosting and evoNet, and demonstrate that in most scenarios, SWIF(r) outperforms current competitors. We have softened the language at the start of the **Discussion** to read as follows:*

Discussion, lines 315-321:

Composite classification frameworks such as SWIF(r) quantitatively ground a common qualitative approach used in scans for adaptive sweeps based on summary statistics: evidence for selection at a locus is considered stronger when extreme values are observed for more than one statistic (Figure 3A). Furthermore, machine-learning approaches like SWIF(r) that learn joint distributions of selection statistics can detect sweep events that individual univariate statistics cannot (Supplementary Figure 16).

SWIF(r) additionally reports calibrated probabilities assessing evidence for selective sweeps site-by-site, resulting in a transparent probabilistic framework for localizing adaptive mutations.

24) Line 238—the authors need to demonstrate that other methods are biased in their handling of missing data before making this claim. They have not done so. Further the authors have not demonstrated the robustness of their own method to missing data and need to do so via simulations.

*We appreciate the reviewer's comment. Please see our overview response to Reviewer #2's comments concerning our clarification of undefined statistics versus missing data. We have removed all mentions of how window-based methods handle undefined statistics, except in lines 19-21 of the revised manuscript in which we acknowledge that this issue does not typically arise for window-based component statistics. Please see our response to Specific Issues #5, as well as **Supplementary Figure 33**, with regards to our analysis demonstrating robustness to undefined statistics.*

25) This brings me to another point—the authors have engaged in a major bait and switch in this paper—they benchmark the method using a feature vector of 5 stats, but only analyse the data using 4. The entire simulation section of the paper should be repeated to reflect what they have done in the empirical analysis.

We appreciate the reviewer's suggestion to remove ΔiHH from our implementation of SWIF(r) throughout the manuscript and have done so in the revised manuscript (see manuscript section entitled "Performance of SWIF(r) using simulated data" and Figures cited in that section).

26) Line 239 – how is SIDT2 reflective of “finer localization?” Because their ad hoc, unpublished naïve bayes classifier missed it?!?

*We have softened the language based on this reviewer's comments, to acknowledge that incorporating correlation of summary statistics is a feature of other methods as well; **Discussion, lines 317-319, now reads:***

Furthermore, machine-learning approaches like SWIF(r) that incorporate joint distributions of selection statistics can detect sweep events that univariate statistics cannot (Supplementary Figure 28).

27) Line 242—ABC does not have likelihoods. That's the whole point.

*We thank the reviewer for pointing out this typo. We have altered the sentence to read: "**While approaches such as Approximate Bayesian Computation can exploit higher-dimensional correlations...**" (Discussion, lines 323-325)*

28) Line 244—where is the evidence that the Peter et al. method is prone to overfitting? This is just a baseless claim.

We thank the reviewer for this comment; we have altered the sentence to read: "...this comes at the cost of genome-scale tractability, and can be vulnerable to the curse of dimensionality" (Discussion, lines 323-325) and have added a citation to Sheehan and Song (PLoS Computational Biology, 2016) where this claim is made.

29) Line 244—again the authors say they are using AODEs...

We thank the reviewer for pointing out the confusion in our terminology; as noted above in our response to Specific Issues #8, we have removed "Bayesian" from the title and abstract, and it does not appear in the manuscript.

30) Line 248-249 – this bit about "cryptic assumptions about class sizes" belies the authors misunderstanding of supervised ML. None of this is true, and this section should be removed.

We thank the reviewer for this comment, and we have removed this sentence.

31) Line 288—again this baseless claim of other methods not being able to deal with missing data.

We thank the reviewer for this comment, and we have removed all discussion about methods other than SWIF(r) and CMS dealing with missing data. We refer to our overview above, where we address the apparent confusion over missing genotype data versus undefined component statistics; throughout the manuscript, we are referring to the latter problem, one that only arises for site-based composite methods. We have made this explicit in lines 19-21 in the revised manuscript.

32) Line 303—100 simulations is way too small for accurate training. The authors should increase this by ~ and order of magnitude.

We thank the reviewer for pointing out the need to clarify the details of our training set. Since SWIF(r) operates site-by-site, 100 neutral simulations results in ~400,000 training points for learning neutral distributions. We also carried out 9,000 simulations containing a sweep, which translates to 9,000 training points for learning sweep distributions. We have made both of these points more explicit in the text (see Online Methods > "Simulation of haplotypes for 1000 Genomes analysis", lines 398-415).

33) Lines 327-329—how were these migration rates arrived at? What does a pulse migration with a rate mean? How was the “match” evaluated quantitatively? This seems very ad hoc as presented.

We thank the reviewer for this comment. The term “pulse migration” refers to migration in one generation; the migration rate is the fraction of the target population replaced by haplotypes from the source population at the given generation. We have added a section to the supplementary text, entitled “Calculation of migration rates from YRI and CEU to ‡Khomani San”, that describes exactly how we arrived at the migration rate estimates. Briefly, we use tract length distribution analysis from Uren et al. (2016) to calculate a matrix of ancestry proportions from YRI, CEU, and San ancestry in the ‡Khomani San at every generation from 14 generations ago to present. We then infer one-generation pulse migration rates (again defined as the replacement fraction in the ‡Khomani San in one generation) that result in the same present-day ancestry proportions, with the YRI migration pulse taking place at 14 generations ago and the CEU pulse at 7 generations ago (following Uren et al. 2016). This is now presented in detail in the new supplementary text entitled “Calculation of migration rates from YRI and CEU to ‡Khomani San”.

34) What do the “full covariance matrices” mean in the context of the Gaussian mixture model? This section is quite unclear.

We thank the reviewer for pointing out the need to clarify the meaning of “full covariance matrices” -- what we mean by this is that the covariance matrices have nonzero off-diagonal elements. We have added this clarification to the text (Online Methods > “Implementation of the Classifiers” > “Implementation of SWIF(r)”, lines 472-473).

35) ROC analysis—the authors should vary the sweep call threshold, not the prior on a sweep here.

*We thank the reviewer for this comment. We initially chose to vary the prior instead of the sweep call threshold because of the interpretability of using 50% as a threshold, but varying the threshold would also be a valid way to generate these curves. We have added **Supplementary Figure 4** that shows that these methods result in equivalent ROC curves, and mention this in Results > “Implementation of SWIF(r)” > Calibration of posterior probabilities calculated by SWIF(r) lines 103-107:*

The calibrated probabilities reported by SWIF(r) can be interpreted directly as the probability that a site contains an adaptive mutation, or fed into a straightforward classification scheme by way of a probability threshold; in this study, we classify sites with a posterior probability above 50% as adaptive SWIF(r) signals. The classifier may be tuned by altering either this threshold or the prior sweep probability π (Supplementary Figure 4).

36) Supplemental Figure 18 – the authors say that the method can be extended but the results presented look terrible. This isn't convincing.

*We refer the reviewer to our overview, in which we emphasize that SWIF(r) is designed as an extensible framework, both in terms of the component statistics used and the classification scenarios. In this figure (now **Supplementary Figure 31**), we show an illustrative example of how SWIF(r) can be extended to more than two classification scenarios. We acknowledge in the figure caption that based on the four component statistics we use in this manuscript, SWIF(r) does not have enough power to distinguish between different sweep timings, but that other component statistics could be incorporated that are more sensitive to the time of sweep onset.*

Reviewer #3 (Remarks to the Author):

Alpert Sugden et al. develop a novel statistical learning approach for localizing selective sweeps, which they term SWIF(r). In contrast to other recent statistical learning approaches (such as that of S/HIC of Schrider and Kern) that also employ multiple summary statistics as evidence for sweeps, the SWIF(r) approach developed here can provide a score per site rather than per region, enabling it to localize sweeps at the polymorphism level. In addition, this new method is robust to missing data, whereas prior statistical learning approaches required extensive and careful data filtering so as to not bias their results. The authors also apply their new statistic to data from the †Khomani San, and characterize a number of regions.

I would like to start out by stating that the manuscript is very well-written, and that I appreciate that the authors included line numbers to make refereeing easier. In addition, I believe that developing statistical learning approaches for localizing adaptive alleles is a hot topic, and the proposed method will likely be of interest to a number of researchers. I am less able to assess the novelty of the empirical findings, but the methodology presented is certainly both innovative and rigorous. Therefore, based solely on the proposed statistical approach, I believe this manuscript is high priority.

Some of the points I find particularly interesting are that each polymorphic site is assigned a posterior probability of being a sweep, that the authors investigated how ascertainment bias could be accounted for (often ignored these days, but important as SNP chips are still common), and that their method can detect both partial hard sweeps to low frequency (e.g., 20%) and soft sweeps from standing variation (though only hard sweeps were used in training). Below are my additional comments and/or concerns of the current manuscript.

We thank the reviewer for reading the manuscript closely and contextualizing its contribution; these thoughtful critiques and suggestions have improved the manuscript. We detail our responses to the specific comments below.

Major comments:

(1) The scan in the ǀKhomani San shows a strong peak in the MHC region. This region is classically thought to be undergoing balancing selection, and a number of approaches for detecting long-lived balancing selection are able to detect the HLA genes within this region. Long-lived balancing selection, which tends to yield increased genetic diversity and decreased population differentiation (F_{st}), appears to be incompatible with signatures of selective sweeps. It would be helpful if the authors could discuss why this region would be identified by methods for detecting selective sweeps. Could it be that the SNPs that are identified by SWIF(r) have low scores for methods that detect balancing selection (i.e., these high probability SNPs would not be detected by methods to identify balancing selection)? Or is there something specific happening in the ǀKhomani San population, which would indicate that this locus should be evolving under recent positive selection rather than long-lived balancing selection?

While the MHC region is a classical example of long-lived balancing selection, alternative selection scenarios have also been posed, including recurrent selective sweeps for different haplotypes and fluctuating directional selection whereby selective targets change or alternate in response to rapid fluctuation in pathogen-mediated pressures (Spurgin and Richardson, Proceedings of the Royal Society B 2010). Our MHC component statistics are qualitatively similar to signals elsewhere in the genome (Supplementary Figure 26, new to this revised manuscript) suggesting that long-lived balancing selection (i.e. heterozygote advantage or frequency dependent selection) is not the primary cause of selection in our SWIF(r) results. However, we remain cautious about over-interpreting the MHC signals because SNP array ascertainment bias in this region is likely worse than elsewhere (as the San contain a huge diversity of HLA alleles, Hilton et al. PLoS Genetics 2015) and we are unable to validate them in the exome follow-up due to poor read mapping sensitivity in the standard exome variant calling pipeline. We have added the following text to the Discussion (lines 370-376):

“In this selection scan, we also see an abundance of SWIF(r) signals in the MHC region involved in immunity. It is possible that this signal reflects balancing selection, which is the mode of selection canonically thought to be occurring within this region (Hedrick, Genetica 1998). Indeed, it has been shown that the signatures of balancing selection and incomplete or recurrent sweeps may be similar to signatures of positive selection (Andrés et al. Molecular Biology and Evolution 2009; Lenz et al. Molecular Biology and Evolution 2016). We note, however, that other studies using different methodologies have detected signatures of directional selection in the MHC in human populations (Albrechtsen et al. Genetics 2010; Zhou et al. PLoS Genetics 2016), and others have noted that fluctuating directional selection is a possible mechanism for pathogen-mediated selection in this region (Spurgin and Richardson, Proceedings of the Royal Society B 2010).”

(2) If single nucleotide statistics (such as F_{st} and Delta DAF) were not used, and instead haplotype-based statistics (such as iHS and Delta EHH) were only used, would correcting for ascertainment matter much? Conrad et al. (2006; Nat Genet 38:1251-1260) indicated (Figure 3

of their article) that the effect of ascertainment bias is removed when using haplotype-based statistics.

*We thank the reviewer for raising this question. While we note that iHS is robust to ascertainment bias, we would not necessarily expect $XP-EHH$ to be, since ascertainment bias could lead to inflation of haplotype lengths in populations that are not represented on the SNP array relative to populations that are (since less variation will be represented for the former populations); we note that the Figure 3 of the Conrad et al. article is looking at haplotype statistics within populations. We do appreciate the point, however, that different statistics would be more robust to ascertainment bias than others; in particular, in future applications, we would like $SWIF(r)$ to be able to incorporate any number of statistics, including those based on the SFS, which would certainly be more vulnerable to ascertainment bias than the statistics we use for this implementation. To acknowledge this point, we have added the following sentence to the **Discussion** (lines 310-312 in the revised manuscript):*

While some of the component statistics we use here may be fairly robust to ascertainment bias, this algorithm also enables future use of component statistics that are more vulnerable to ascertainment bias, such as SFS statistics, within $SWIF(r)$'s framework.

(3) The authors indicate that they expect that $SWIF(r)$ is robust to background selection, because Enard et al. demonstrated that haplotype-based statistics are robust to background selection. However, two of the statistics, and indeed those with the most power (F_{st} and Delta DAF from what I can tell from Figure 1D and Supplementary Figure 17) are not haplotype-based, and therefore likely influenced by background selection. In addition, based on comment 2 above, the argument that haplotype-based statistics are robust to background selection can also then be used to argue that $SWIF(r)$ is robust to ascertainment bias. Because background selection is a ubiquitous force (e.g., McVicker et al. 2009, Comeron 2014), it would be helpful if the authors tested their method's robustness to this scenario through some illustrative simulations.

*We thank the reviewer for this comment, and we agree that illustrating robustness to background selection in simulation is important. Please see our response to "Major issues" #6 from reviewer 2 for details on the new simulations that we generated in order to demonstrate this robustness, as well as **Results > "Performance of $SWIF(r)$ using simulated data"** (lines 192-200), **Online Methods > "Simulation of background selection"** (lines 449-464), and **Supplementary Figure 20** in the revised manuscript.*

Minor comments:

- Citations of figure panels within the text use uppercase letters (e.g., Figure 1A or Figure 1B), whereas the panels are labeled using lowercase letters (e.g, a and b) within each figure.

We thank the reviewer for catching this, and have remedied it.

- On page 11, line 232 the authors write "The composite classification framework of SWIF(r) quantitatively grounds a common qualitative approach used in scans for adaptive sweeps". Though I understand what the authors are getting at, likelihood ratio approaches essentially do to the same thing, as the likelihood ratio quantitatively assess the probability of observing the data under the alternative hypothesis of selection relative to the probability of observing the data under the null hypothesis of neutrality. I think the authors mean that the qualitative approach is common when using summary statistics, such as the component statistics used by SWIF(r).

We agree with the reviewer that our original writing oversimplifies available approaches for selection scans. We have altered the start of the paragraph the reviewer cites here (including the sentence cited in this particular reviewer comment) as follows:

(Discussion, lines 315-320):

Composite classification frameworks such as SWIF(r) quantitatively ground a common qualitative approach used in scans for adaptive sweeps based on summary statistics: evidence for selection at a locus is considered stronger when extreme values are observed for more than one statistic (Figure 3A). Furthermore, machine-learning approaches like SWIF(r) that learn joint distributions of selection statistics can detect sweep events that individual univariate statistics cannot (Supplementary Figure 16). SWIF(r) additionally reports calibrated probabilities assessing evidence for selective sweeps site-by-site, resulting in a transparent probabilistic framework for localizing adaptive mutations.

Reviewer comments and responses:

Reviewer #1 (Remarks to the Author):

The paper describes a composite approach to scan for selective sweeps. I think the paper provides an innovative framework to scan for natural selection in genomes. Genome scans usually relies on a one-dimensional summary statistic and lack of power might result from one-dimensional genome scan. The authors convincingly show that their approach to combine summary statistics is more effective than one-dimensional genome scans and that the existing CMS score partly because they manage to locate more precisely adaptive SNPs. I have only two comments that concern statistical methods.

We thank the reviewer for taking the time to read and critique our manuscript, and for contextualizing the contribution of our framework for genomic scans for selection. We detail our response to the specific comments below.

1. The authors write in different places of the ms that their method is probabilistic which is an advantage because of the natural scale of probabilities. However, to be meaningful probabilities, their returned probabilities should be properly calibrated. For instance, a well-calibrated classifier should classify SNPs such that among the SNPs to which it gave a probability close to 0.8 to be adaptive, approximately 80% of these should be adaptive. I think that the current approach might not be properly calibrated and recalibration techniques exist and should be considered (see eg. <http://scikit-learn.org/stable/modules/calibration.html>). For instance, the paper entitled “Ancestry Composition: A Novel, Efficient Pipeline for Ancestry Deconvolution” (<http://www.biorxiv.org/content/biorxiv/early/2014/10/18/010512.full.pdf>) uses isotonic regression to recalibrate local ancestry probabilities.

*We thank the reviewer for this comment, and agree that the interpretation of SWIF(r)'s reported probabilities requires more detailed explanation. Following the reviewer's suggestion, we have implemented an isotonic regression scheme for applying SWIF(r) to data, and have incorporated that scheme into our analyses of data from the 1000 Genomes Project and from the ‡Khomani San. Text detailing our implementation is excerpted below, (**Results > Implementation of SWIF(r) > lines 88-102**); **Supplementary Figures 1-3** show results from calibration and illustrate the smoothed isotonic regression scheme we implemented to maintain the strict monotonicity of the posterior probabilities reported by SWIF(r) post-calibration. **Figures 2 and 3 and Table 1** in the main text, manhattan plots in **Supplementary Figures 17-19 and 24**, and full results in **Supplementary Tables 2, 4, 6, and 7** have all been updated to reflect the calibrated sweep probabilities. We have added text to the **Discussion (lines 332-344)**, excerpted below, and have added code to the SWIF(r) repository (<https://github.com/ramachandran-lab/SWIFr>) to implement our calibration scheme for any dataset.*

Results > Implementation of SWIF(r) > lines 88-102:

Calibration of posterior probabilities calculated by SWIF(r). A desirable property of probabilities, like those calculated by SWIF(r), is that they be well calibrated: in this context, for the variant positions where the posterior probability reported by SWIF(r) is around 60%, approximately 60% of those sites should contain an adaptive mutation, and approximately 40% should be neutral. We implemented a smoothed isotonic regression scheme to calibrate the probabilities calculated by SWIF(r) (Online Methods). Briefly, when applying SWIF(r) to a given dataset, we calculate the empirical frequencies of neutral and sweep variants that are assigned posterior probabilities between 0 and 1 in simulation, and use isotonic regression (Zadrozny and Elkan, ACM 2002) to map the posterior probabilities to their corresponding empirical sweep frequencies (Supplementary Figure 1, Supplementary Figure 2). We then impose a smoothing function that prevents multiple posterior probabilities from being mapped to the same calibrated value (Supplementary Figure 3, Supplementary Figure 1E, Supplementary Figure 2E). This calibration procedure relies on the relative makeup of the training set; a classifier that is calibrated for a training set made up of neutral and sweep variants in equal parts would not be well-calibrated for a training set in which sweep variants only make up 1% of the whole. For each application of SWIF(r) in this study, we calibrated SWIF(r) for a specific training set makeup (Online Methods; see also Supplementary Figure 1 and Supplementary Figure 2).

Discussion, lines 332-344:

In order for the class probabilities reported by SWIF(r) to be practically interpretable, we calibrated SWIF(r), such that $k\%$ of variants with a posterior sweep probability of $k\%$ are indeed sweep variants. We have implemented a calibration scheme based on isotonic regression for SWIF(r) that maps the posterior sweep probabilities to their empirical sweep proportions in simulated data (Supplementary Figure 1, Supplementary Figure 2, Supplementary Figure 3), but importantly, this calibration relies on the composition of the training set used. While for some classifiers, the proportions of classes are known, or can be reliably estimated (e.g. see Durand et al. biorxiv 2014 and Scheet and Stephens, American Journal of Human Genetics 2006), the proportion of sites throughout the human genome that are adaptive is unknown. For calibrating SWIF(r), we chose training sets made up overwhelmingly of neutral variants; while our calibration of SWIF(r) always preserves the rank order of posterior probabilities (Supplementary Figure 3), the specific choice of training set makeup can have a dramatic effect on the calibration. Therefore, a direct interpretation of the posterior probabilities reported by SWIF(r), or any other classifier that calculates probabilities, must incorporate knowledge of the scenarios used for training and calibration.

2. The authors use ROC curves for method comparisons. I do not think it is a good idea because for selective scans the number of neutral SNPs is several orders of magnitude larger than the number of adaptive SNPs. Let say you have 100 adaptive SNPs and 100,000 neutral ones, a type I error of 5% and a power of 90% (SWIF performance in Figure 1A) would correspond to 90 true positives and 5,000 false positives! That is clearly not a relevant point.

The authors should instead consider a Power-FDR curve, which is equivalent to the precision-recall curve used in machine learning.

*We thank the reviewer for this comment, and agree that the Power-FDR curve offers important information for the performance comparison of SWIF(r) to other sweep detection methods. We note that the shape of these curves depends on the composition of the training set, so we have added **panel E to Figure 1** in the main text that plots the Power-FDR curves for the same training set composition that we use for calibrating SWIF(r) for application to the ‡Khomani San dataset. We have also added **Supplementary Figure 11**, which plots these curves for a range of training set compositions. We further include a panel with Power-FDR curves for each of our benchmarking comparisons, added based on reviewer 2's comments (**Supplementary Figures 6F** for SweepFinder, **9F** for evolBoosting, and **10F** for evoNet). Depending on the composition of the training set, SWIF(r) performs comparably or better than CMS, evolBoosting, and evoNet, and dramatically outperforms the component statistics and SweepFinder, and achieves an FDR near zero for a true positive rate of ~35%. The curves do, however, illustrate the difficulty in detecting evidence genome-wide for adaptation at the SNP level, as very small false positive rates can translate into substantial false discovery rates. This is an important point to make, in light of the number of selection scans that are performed genome-wide. We also note that this will be an issue for window-based methods as well, although with a different manifestation: the false discovery rate for window classification may be smaller (as we show for evolBoosting, evoNet, and a window-based version of SWIF(r) in **Supplementary Figures 9F and 10F**), but a classified window contains a large number of variants. Therefore, these methods will likely not narrow down the list of putative SNP targets. We have added the following text at the end of **Results > Performance of SWIF(r) using simulated data (lines 154-171)**:*

*While ROC curves are informative for illustrating the performance of different sweep detection methods, it is important to note that the genome has far more neutral variants than adaptive mutations. Therefore, more relevant performance comparisons can be made by illustrating the predicted false discovery rate (FDR) for a given true positive rate using Power-FDR curves. These curves depend on the composition of the training set, since the false discovery rate rises as the proportion of adaptive variants in the training set decreases. In **Figure 1E** and **Supplementary Figure 6F**, we plot Power-FDR curves for all methods using the same training set composition we use for calibration of SWIF(r) for application to the ‡Khomani San dataset (99.95% neutral variants and 0.05% adaptive variants). Curves for other training set compositions can be found in **Supplementary Figure 11**. Power-FDR comparisons between SWIF(r) and evolBoosting and evoNet can be found in **Supplementary Figure 9F** and **Supplementary Figure 10F** assuming that 1% of windows contain a sweep. SWIF(r) performs well relative to its component statistics, CMS, and SweepFinder, however, these analyses illustrate the inherent difficulty of site-by-site detection of adaptive mutations. Because there are so many more neutral variants than adaptive variants in the genome, even a small false positive rate can result in a substantial false discovery rate. Window-based methods, including window-based SWIF(r), may appear to have lower false discovery rates (since*

there are many fewer windows than variants, and thus fewer opportunities for false positives to arise; see Supplementary Figure 9 and Supplementary Figure 10), but this comes at the cost of a longer list of putative SNP targets, since each classified window contains a large number of individual variants.

Sincerely,
Michael Blum

We thank the reviewer for signing his review and acknowledge him by name in our Acknowledgments section; his comments improved the manuscript and method.

Reviewer #2 (Remarks to the Author):

Review of Sugden et al

In this manuscript the authors join the current cottage industry of using supervised machine learning methods for detecting selection in population genomic data. Briefly the authors slightly tweak vanilla AODE classifiers to use a feature vector of 5 population genetic summary statistics (or 4 for the empirical analysis) to differentiate incomplete (more on this in a moment) hard sweeps from neutral regions of the genome. They show that their method, SWIFr, has better power and accuracy than univariate summaries and CMS. They then apply the method to unpublished data from an KhoeSan population and tell stories about metabolic adaptation.

Frankly the best part of this paper is the name of the method (well done). The method itself represents an incremental advance for the field but one that is presented with such bluster that it is well suited for the age of Trump. For instance the authors claim in lines 10-11 that less attention has been given to “exploiting multiple genomic signatures [...] in a rigorous analytic framework.” This statement basically ignores the recent advances in supervised machine learning in population genetics including methods such as evoNet (Sheehan and Song 2016), SHIC (Schridder and Kern 2016), EvolBoosting (Lin et al. 2011), SFSelect (Ronen et al. 2013), the hierarchical boosting method of Pybus et al (2015), and PreCISS (2016) to name a few. Perhaps the authors are referring to the thin distinction between window vs. SNP centered methods? If so this is barely enough of a difference to downplay all of the powerful, related methods that have been developed thus far and may very well outperform SWIFr (more on this later).

Another example of such incredible hype in this paper is one of the supposed major motivations for this new method, repeated again and again but first mentioned in lines 25-26 is that the other ML methods “such as random forest classification, regression, and support vector machines [...] cannot operate with missing data.” This is patently false. For instance for RF, Breiman laid out specifically how to deal with missing data from the very start of the method (i.e. https://www.stat.berkeley.edu/~breiman/RandomForests/cc_home.htm). Similarly evoNet, SFSelect, SHIC, and EvolBoosting can all handle missing data through training with appropriate

matched masked simulations—and the authors perform this in the papers. The authors here also have a way of handling missing data, but this is not unique to their method despite their claims.

We thank the reviewer for the close reading of our manuscript and for drawing attention to multiple issues that required clarification on our part, as well as additional necessary benchmarking experiments. We have made serious efforts to address all of this reviewer's concerns in the revised manuscript and present our revisions in detail following each point the reviewer outlines below.

As an overview of this revised manuscript and our new analyses and text in response to this reviewer, we now additionally benchmark SWIF(r) against site-based statistic SweepFinder and two supervised machine-learning sweep detection methods, evolBoosting and evoNet, and show that SWIF(r) outperforms these approaches (Supplementary Figures 6, 9 and 10). We have also added text in multiple places to acknowledge existing machine-learning methods for sweep-detection, both SNP- and window-based, and the contribution SWIF(r) makes in the context of these existing methods. We would like to emphasize, however, that the distinction between localizing putative adaptive SNPs versus identifying regions containing adaptive SNPs is not a “thin” one: the window sizes used by these methods range from 20kb (evoNet) to 200kb (S/HIC), and SWIF(r)'s ability to be applied site-by-site generates hypotheses for localizing functionally adaptive variants (as illustrated in Figures 2 and 4). We believe that SWIF(r) and other SNP-based sweep detection methods can enable new insight into selective sweeps by localizing adaptive variants and therefore we emphasize this distinction. That said, in the revised manuscript we demonstrate how SWIF(r) can be altered to assess evidence of sweeps at various window sizes (Supplementary Figures 9 and 10; Online Methods section “Implementation of window-based methods”). A summary of our new benchmarking experiments can be found in Results > Performance of SWIF(r) using simulated data (lines 138-153).

Before proceeding to directly address the reviewer's comments individually, we also clarified in the revised manuscript that our definition of “missing data” for SWIF(r) in our original submission referred to sites at which one or more selection statistics is undefined, and not sites for which genomic data are unavailable. We have now clarified this in the manuscript by referring to “undefined component statistics” instead of “missing data”. In the revised manuscript, we examine how undefined component statistics affect SWIF(r)'s performance (Supplementary Figure 33), an analysis we did not find in the machine-learning sweep-detection methods the reviewer highlighted (evoNet, SFSelect, S/HIC, and EvoBoosting); we would appreciate being pointed towards such analyses. We do note that the issue of undefined statistics may not arise when using window-based component statistics, as the most common undefined statistics are site-based iHS and XP-EHH. We have removed all claims about how other methods, save for CMS, deal with this problem, but we maintain that SWIF(r)'s ability to compute the probability of a sweep at sites with undefined statistics is a noteworthy feature worth highlighting in this manuscript.

Major issues:

1) The authors have not compared the performance of SWIFr to appropriate competing methods. Despite their claim from line 38 that they assess SWIFr in comparison to “state-of-the-art univariate and composite methods” the authors have ignored all of the modern ML methods listed above and their claim here is laughable. A proper exploration of their method would compare at the very least to evoNet, SFSelect, SHIC, EvoBoosting, PreCI OSS, and also the new algorithm from Bafna, iSAFE (<http://www.biorxiv.org/content/early/2017/06/02/139055>). Moreover the authors have instead chosen to focus on a Naïve Bayes Classifier that they whipped up for comparison—who cares about this method? What the authors present here is simply not adequate.

We thank the reviewer for this comment. As described in our overview above, we have now benchmarked SWIF(r) against SweepFinder (Figure 1 and Supplementary Figure 6), CMS (Figure 1), evolBoosting (Supplementary Figure 9), and evoNet (Supplementary Figure 10), and find that SWIF(r) outperforms all of these methods over a range of sweep parameters. The results of these benchmarking experiments can be found in Results > Performance of SWIF(r) using simulated data. The paragraph in which we summarize the results of benchmarking against the other composite methods is excerpted below (lines 138-153):

While few composite methods for sweep detection operate site-by-site, there are a handful of machine-learning composite approaches that identify genomic windows containing adaptive mutations (Lin et al. Genetics 2011; Pybus et al. Bioinformatics 2015; Schrider and Kern, PLoS Genetics 2016; Sheehan and Song, PLoS Computational Biology 2016). In order to compare SWIF(r) against such methods, we had to alter SWIF(r) to calculate window-based sweep probabilities; there are many potential ways to do this that may be differentially powerful, and here we chose simply to use the highest probability assigned to any variant within a given genomic window as the probability for that window. We compared window-based SWIF(r) to two state-of-the-art composite window-based methods: evolBoosting (Lin et al. Genetics 2011), which combines 120 statistics using boosted logistic regression, and evoNet (Sheehan and Song, PLoS Computational Biology 2016), which was developed to jointly infer demography and selection using a deep learning framework (Online Methods). Since both SWIF(r) and evoNet are frameworks that are designed to incorporate any set of statistics, we implemented evoNet to use the same statistics as SWIF(r). When comparing SWIF(r) with evolBoosting, we used 40kb windows following Lin et al. (Genetics, 2011), and show that SWIF(r) outperforms evolBoosting across a range of sweep parameter values (Supplementary Figure 9). For comparison with evoNet, we used 100kb windows following Sheehan et al. (PLoS Computational Biology, 2016) (Supplementary Figure 10). We find that SWIF(r) performs similarly to or better than evoNet in this implementation, although we note that this analysis likely downplays the strengths of both methods, as each method has been altered from its original design to enable a direct comparison.

We have removed all comparisons to the Naive Bayes Classifier, which we originally implemented to illustrate the gains made over CMS when computing a true posterior sweep probability. We agree with the reviewer that, given the new benchmarking we have done, this comparison is rendered unnecessary.

SWIF(r) is intended to be a flexible framework that can incorporate any number of statistics, and our released software (<https://github.com/ramachandran-lab/SWIFr>) is implemented with this flexibility in mind. The four summary statistics we use in the implementation of SWIF(r) in this paper were done for direct comparison against CMS and because they can all be calculated site-by-site (as the reviewer suggested, all applications of SWIF(r) now consistently exclude ΔiHH throughout the manuscript). As new statistics are developed, leveraging joint likelihoods between those statistics and others will improve classification (as we show when incorporating iSAFE, delineated in a biorxiv preprint <https://www.biorxiv.org/content/early/2017/10/01/139055>, into SWIF(r) in **Supplementary Figure 29, Discussion lines 327-329**). Methods such as evolBoosting and CMS do not have this flexibility; we maintain that the flexible ability to incorporate new selection statistics will make SWIF(r) more valuable as the field moves forward.

The set of methods we benchmark against in this revised manuscript represents a range of state-of-the-art methods for sweep detection. SWIF(r) in the form implemented in this study was not intended to be applied to soft sweeps (in comparison to S/HIC), and existing sweep detection methods vary in their usability and documentation. We hope these new experiments address the reviewer's concerns regarding the performance of SWIF(r) relative to appropriate competing methods.

2) The manuscript is very confused in its use of the term "complete sweeps." I believe what the authors mean is complete in the sense that the allele frequency has gone to frequency one in a single deme, not the whole population. This is not the canonical use of the term and thus the authors must fix this throughout the manuscript.

We thank the reviewer for pointing out this needed clarification; in both the main text (**lines 25-26, 37-38, and 132**) and in all relevant figure captions (**Figure 1, Supplementary Figures 6, 8, 9, 10**), we now state that we use "complete sweeps" to refer to sweeps where the beneficial mutation has reached fixation in the population of interest.

3) The demographic models used for training, testing, and benchmarking of SWIFr are extremely dated and no longer appropriate. Neither the Shaffner et al model nor the Gronau et al. model incorporate the strengths of population growth now thought to characterize human demographic history. These should be updated to more modern estimates of population size history (see for instance Auton et al). A new demographic model will need to be used for all simulations for training, performance characterization, and eventual application.

We thank the reviewer for this comment. We agree that it is important to demonstrate that SWIF(r) is robust to demographic misspecification involving exponential population growth. To demonstrate this, we simulated a new dataset using a demographic model from Gravel et al.

(PNAS, 2011) that incorporates recent exponential population expansion with rates of 0.38% and 0.48% for Europe and East Asia respectively for the last 23,000 years. We allowed this expansion to last until 5,000 years ago to allow for simulating very recent sweeps, which the simulation software *cosi* cannot simulate simultaneously with demographic change (*discussion and details can be found in lines 186-191 and lines 439-447, excerpted below*). We have added a panel to *Supplementary Figure 13* that illustrates that, when *SWIF(r)* is trained on the Schaffner demographic model and tested on these simulations from the Gravel model, *SWIF(r)* maintains roughly the same power as when *SWIF(r)* is both trained and tested on simulations from the Schaffner model. Because of this, and the fact that we intend *SWIF(r)* to be a generalizable framework that can learn from any set of simulations and scenarios, we have retained our original results, now calibrated (see our response to reviewer 1), from application to the 1000 Genomes dataset and the \ddagger Khomani San dataset.

Results > “Performance of SWIF(r) using simulated data” > “Robustness of SWIF(r) to demographic model misspecification” lines 186-191:

*Both the Schaffner model and the Gronau model, as implemented, do not include very recent population expansion, so we implemented a third demographic model from Gravel et al. (PNAS 2011) that includes exponential population growth within the last 23,000 years. Since *cosi* cannot simulate selective sweeps overlapping with demographic changes, we only simulated sweeps beginning 5kya, allowing for 18,000 years of exponential expansion (Online Methods). In Supplementary Figure 13, we show that *SWIF(r)* is also robust to this recent population expansion.*

Online Methods > “Simulation of haplotypes including recent population growth” lines 440-447:

*To test *SWIF(r)*’s robustness to misspecification of recent population growth, we implemented a set of simulations using a demographic model from Gravel et al. (PNAS 2011) that estimates recent exponential population expansions with rates of 0.38% for Europe and 0.48% for East Asia over the last 23,000 years. Since the simulation software *cosi* (Schaffner et al. Genome Research 2005) cannot simulate sweeps and population-level changes simultaneously, we allowed the expansion to last from 23,000 years ago to 5,000 years ago, to allow for sweeps beginning 5,000 years ago. We also included the migration rates inferred by Gravel et al. (PNAS 2011) between Europe, East Asia, and Africa, and between Africa and the ancestral population of Europe and East Asia. As in other analyses, We simulated selective sweeps spanning a range of present-day allele frequencies from 20% to 100%.*

4) The eventual application is on an unpublished dataset—this is not appropriate unless the authors wish to fold all of the details of that dataset and release it herewith. The authors should apply the method to an already published human data set instead.

SWIF(r) is applied to (1) publicly available data from the 1000 Genomes (URLs listed in the revised manuscript under “Software and Data availability”, preceding Acknowledgments) and (2) published data (Uren et al. Genetics 2016, and Martin et al. in press) from the \ddagger Khomani San.

Below we detail how investigators can access the \ddot{K} homani San data and revisions in the manuscript text that provide these details to readers of this study.

The \ddot{K} homani San exome dataset is described by Martin et al. (in press at Cell, and available as a preprint on bioRxiv <https://www.biorxiv.org/content/early/2017/10/13/200139>). Details regarding mapping and variant calling may be found in Martin et al. (preprint link above) and in Kidd et al. (2014, BMC Genomics) for a partial dataset. Unfortunately, we are unable to make these data freely available due to local community constraints on data sharing by the San Council; please see

<<http://www.sciencemag.org/news/2017/03/san-people-africa-draft-code-ethics-researchers>> and

<<http://trust-project.eu/wp-content/uploads/2017/03/San-Code-of-RESEARCH-Ethics-Booklet-final.pdf>>. Interested parties should contact the South African San Council for research and ethics review (Leana Snyders <leana@yemail.com>; admin@sasi.org.za>).

To make these details clear in the revised manuscript, we changed the URLs section preceding Acknowledgments to be titled “Software and Data availability” and now give details on contact information for the South African San Council in that manuscript section (“**Software and Data availability**”, lines 608-611; text excerpted below).

“ \ddot{K} homani San genotype data were first described by Uren et al. (2016), and \ddot{K} homani San exome data were first described by Martin et al. (in press). Queries regarding access to \ddot{K} homani San data analyzed here should be sent to the South African San Council for research and ethics review by contacting both Leana Snyders (leana@yemail.com) and admin@sasi.org.za.”

5) The simulations done in Figure 1 and associated supplemental figures are way too limited with respect to the parameter ranges considered. The authors are focusing on very recent sweeps only, and for undisclosed values of $\alpha = 2Ns$. The authors should simulate under a range of α , preferably between 10^2 - 10^4 so that the results can be put in their proper context. Additionally the authors should vary recombination and mutation rates. Also see above for comments on demographic model.

We thank the reviewer for pointing out the need to clarify our simulation parameters. The selection strength (s) parameters are provided in **Supplementary Table 5**, and do in fact constitute a range of values for α from ~ 100 to ~ 4500 . In addition, we vary recombination rates; for each simulation, we draw a new recombination map from a hierarchical recombination model that draws realistic regional rates and recombination hotspots, as described in Schaffner et al. (2005). In response to this comment, we have made both of these points more explicit in the text (**Online Methods > “Simulation of haplotypes for 1000 Genomes analysis”, lines 399-401 and lines 405-413**). We believe that varying the mutation rate on top of this large parameter set is beyond the scope of this paper, in which our goal is to present a general framework for hard sweep detection that incorporates training simulations from a given demographic model.

6) The authors need to do simulations with background selection to see how that affects the method. Its crazy that in 2017 this hasn't been done.

We thank the reviewer for this comment, and we agree that it is important to demonstrate that SWIF(r) is robust to background selection; in the revised manuscript we demonstrate this in Supplementary Figure 14, and discuss these results in a new section in the manuscript (Results > "Performance of SWIF(r) using simulated data", lines 192-200 and Online Methods > "Simulation of background selection", lines 448-464). We excerpt these sections below:

Results > "Performance of SWIF(r) using simulated data", lines 192-200:

Robustness of SWIF(r) to background selection. We also assessed the sensitivity of SWIF(r) to background selection by generating a set of training simulations containing neutral regions, selective sweeps, and exonic regions, using forward simulator slim (Messer, Genetics 2013) (Online Methods). We trained SWIF(r) on both neutral and sweep simulations, and tested the ability of SWIF(r) to distinguish between exonic and sweep sites, relative to its ability to distinguish between neutral and sweep sites. We find that SWIF(r) is fully robust in this scenario, meaning that we would not expect background selection in genic regions to result in false positive sweep signals (Supplementary Figure 14). These results are aligned with those of Enard et al. (Genome Research, 2014), who have shown that background selection has little to no effect on haplotype-based statistics iHS and XP-EHH (and in fact makes iHS more conservative).

Online Methods > "Simulation of background selection", lines 448-464

Simulation of background selection

For evaluating SWIF(r)'s robustness to background selection, we generated 3 sets of simulations of 1Mb each using forward simulator slim (Messer, Genetics 2013): neutral regions, regions with a hard sweep, and genic regions. For genic regions, we followed Messer and Petrov (Trends in Ecology and Evolution, 2013) to simulate gene structure: each simulation had one gene with 8 exons of 150bp each, separated by introns of 1.5kb, and flanked by a 550bp 5'UTR and a 250bp 3'UTR. Within exons and UTRs, 75% of sites were assumed to be functional. Mutations were assumed to be codominants, and fitness effects across different sites were assumed to be additive. Functional sites were divided into 40% "strongly deleterious" sites with selection coefficient -0.1, and 60% "weakly deleterious" sites with selection coefficients between -0.01 and -0.0001. The mutation rate was set at 2.5×10^{-8} per site per generation, and the recombination rate at 10^{-8} . Note that for testing the robustness of SWIF(r), we only considered sites from these simulations that landed in exons or UTRs.

For all three sets of simulations, we simulated two populations with $N_e = 5000$, which split from each other 40,000 years ago. For sweep simulations, we drew selection coefficients for the beneficial allele from an exponential distribution with mean 0.03, and sweeps begin 10,000 years ago. Since forward simulations are much more computationally intensive than coalescent simulations, we rescaled our parameters by a

factor of 10 (10 times larger for mutation rate, recombination rate, and selection coefficients, 10 times smaller for population sizes and 10 times shorter for all times) to make the simulations feasible (Messer, Genetics 2013).

7) What the heck is going on with the empirical false positive rates in Supplemental tables 3 and 6? It appears that using the 50% threshold (as the authors report they are doing on lines 129) the false positive rates are way too high to believe any of the results. Are the authors really trying to publish a paper with FDR ~56%?

We thank the reviewer for drawing attention to this point. Following this comment and feedback from reviewer 1 (see our response to point 2 of reviewer 1's comments), we have added FDR analysis to our assessment of SWIF(r)'s performance (Figure 1E, Supplementary Figure 11; see lines 154-170) that shows that SWIF(r) outperforms other site-based methods including CMS in terms of the tradeoff between power and FDR (see also panel F in Supplementary Figures 9 and 10 for comparison of window-based SWIF(r) to evolBoosting and evoNet). These curves also illustrate the difficulty of identifying evidence for selection at the SNP-level genome-wide, as small false positive rates can result in moderate or substantial false discovery rates when the ratio of neutral sites to adaptive sites is extremely high. To our knowledge, no SNP-based sweep-detection methods report FDR rates. We also show that window-based SWIF(r) outperforms evolBoosting and evoNet in FDR analyses (Supplementary Figures 9F, 10F). While FDR rates appear to be lower for window-based methods including window-based SWIF(r), this is a tradeoff that results in less information about localization of the adaptive mutation site.

In the case of Supplementary Tables 4 and 6, we offer two estimates of false discovery using two different definitions of neutrally evolving sequence. The rates provided in the tables define neutral regions as non-conserved and non-coding following Hernandez et al. (Science, 2011), and result in higher false discovery rate estimates; we believe these are very conservative estimates of FDR, as signals at sites with low conservation may well be tagging nearby sweep variants, and intergenic regions contain many regulatory and functional elements. Using another, stricter definition of neutral sequence following Gazave et al. (PNAS, 2014), we see zero false positives in all four populations. We have made these points more explicit in the captions of Supplementary Tables 4 and 6.

More specific issues:

1) In the abstract the authors claim SWIFr “explicitly models demography.” It doesn't. It trains on simulations that have a specified demographic model.

We have changed the abstract to incorporate the reviewer's suggested phrasing: “SWIF(r) is trained using simulations from a user-specified demographic model and explicitly models the joint distributions of selection statistics....”

2) Lines 1 -8. This introductory paragraph is incredibly thin on scholarship. Further “adaptive sweeps” are not common vernacular—the authors perhaps mean “selective sweeps.” This needs to be fixed considerably.

We thank the reviewer for pointing out the need for a more detailed description of the literature. The second paragraph of the introduction now includes a more comprehensive overview of the suite of machine-learning approaches that have been designed for identification of selective sweeps, with citations pointing to evolBoosting, evoNet, S/HIC, SFSelect, and Hierarchical Boosting, and we are more explicit about what we see as the importance and challenges of a sweep-detection method that localizes the site of adaptive mutations (versus identifying windows containing adaptive mutations). In addition, as the reviewer suggested, we have changed “adaptive sweeps” to “selective sweeps” throughout the revised manuscript. The new introductory text is excerpted below (Introduction, lines 11-21 and 27-35).

Introduction, lines 11-21:

Recently, there has been increased focus on developing composite methods for identifying selective sweeps, which combine multiple statistics into a single framework (Pavlidis et al. Genetics 2010; Grossman et al. Science 2010; Lin et al. Genetics 2011; Ronen et al. Genetics 2013; Grossman et al. Cell 2013; Pybus et al. Bioinformatics 2015; Schrider and Kern, PLoS Genetics 2016; Sheehan and Song, PLoS Computational Biology 2016); we refer to the statistics that are aggregated in composite methods such as these as “component statistics”. Most composite methods draw upon machine learning approaches like support vector machines (Pavlidis et al. Genetics 2010; Ronen et al. Genetics 2013) deep learning (Sheehan and Song, PLoS Computational Biology 2016), boosting (Lin et al. Genetics 2011; Pybus et al. Bioinformatics 2015) or random forest classification (Schrider and Kern, PLoS Genetics 2016) in order to identify genomic windows containing selective sweeps. These windows vary in size from 20kb to 200kb, often identifying candidate sweep regions containing many genes (Schrider and Kern, PLoS Genetics 2016; Sheehan and Song, PLoS Computational Biology 2016). One method, the Composite of Multiple Signals or “CMS” (Grossman et al. Science 2010; Grossman et al. Cell 2013), uses component statistics that can be computed site-by-site in pursuit of localizing adaptive variants within genomic windows, but the output from this method cannot be interpreted without comparison to a genome-wide distribution. In addition, CMS must rely on imputation or other methods of compensation when component statistics are undefined, a complication that typically does not arise when using window-based component statistics.

Introduction, lines 27-35:

Here we introduce a Bayesian classification framework for detecting and localizing adaptive mutations in population-genomic data called SWIF(r) (SWEEP Inference Framework (controlling for correlation)). SWIF(r) has three major features that enable genome-wide characterization of adaptive mutations: first, SWIF(r) computes the per-site probability of selective sweep, which is

immediately interpretable and does not require comparison with a genome-wide distribution; second, no imputation or compensation mechanisms are necessary in the case of undefined component statistics; and third, we explicitly learn pairwise joint distributions of selection statistics, which gives substantial gains in power to both identify regions containing selective sweeps and localize adaptive variants. Existing composite methods for selection scans have subsets of these features, but SWIF(r) combines all three in a unified statistical framework.

3) Line 15 – the authors say other ML methods combine statistics into “one score.” Actually this usually isn’t the case. Instead most of these methods yield a vector of probabilities indicating class membership.

We agree with the reviewer that this sentence was unclear. We have changed it to read: “Recently, there has been increased focus on developing composite methods for identifying selective sweeps, which combine multiple statistics into a single framework.” (Introduction, lines 11-12)

4) Line 32—I believe only CMS “requires comparison with a genome-wide distribution.” None of the modern ML methods needs this.

SweepFinder, CMS, and all the univariate selection statistics studied here require comparison with a genome-wide distribution; while we agree that modern ML methods do not require this, their output may not be well calibrated, as SWIF(r) is in this revised manuscript (see our response to point 1 of reviewer 1). In response to this and other comments from this reviewer, we have softened our language as follows:

Introduction, lines 29-35:

SWIF(r) has three major features that enable genome-wide characterization of adaptive mutations: first, SWIF(r) computes the per-site probability of selective sweep, which is immediately interpretable and does not require comparison with a genome-wide distribution; second, no imputation or compensation mechanisms are necessary in the case of undefined component statistics; and third, we explicitly learn pairwise joint distributions of selection statistics, which gives substantial gains in power to both identify regions containing selective sweeps and localize adaptive variants. Existing composite methods for selection scans have subsets of these features, but SWIF(r) combines all three in a unified statistical framework.

5) Lines 61-62 – if missing data is handled via removing a component of the feature vector the authors need to provide evidence that power/specificity isn’t affected. No such evidence is provided. This should be provided for each summary statistic.

We thank the reviewer for pointing out the need to evaluate the robustness of our method in the presence of undefined statistics. We have provided such an analysis in Supplementary Figure

33, where we plot ROC curves for sites with all statistics defined, and the corresponding curves for situations where each of the summary statistics is undefined. We show that for XP-EHH, DDAF, and Fst, the power of the method is not compromised, but that missing values of iHS do result in somewhat lower power. This is unsurprising, since iHS is by far the most sensitive component statistic of the four in the case of incomplete sweeps. We have added text to the manuscript that acknowledges this (*Online Methods > “Implementation of SWIF(r), lines 484-486*).

6) Line 85 – unclear why the authors are presenting a Naïve bayes classifier here. The authors should focus on competing, published methods (See above).

As noted in our response to point 1 under “Major issues” from the reviewer, we have removed mention of the Naive Bayes classifier here and throughout the manuscript.

7) Lines 100-101 – we have known for a long time that combining stats using machine learning is better than individual summaries of the data. PLEASE ACKNOWLEDGE THE LITERATURE.

This sentence was originally intended to provide intuition behind the power that any composite approach gains by combining statistics that are differentially powerful across different scenarios. We have altered the sentence to read: “This underscores the advantage of composite methods for detecting selective sweeps when the parameters of the sweep are unknown (Pavlidis et al. Genetics 2010; Grossman et al. Science 2010; Lin et al. Genetics 2011; Ronen et al. Genetics 2013; Grossman et al. Cell 2013; Pybus et al. Bioinformatics 2015; Schrider and Kern, PLoS Genetics 2016; Sheehan and Song, PLoS Computational Biology 2016).” (Results > “Performance of SWIF(r) using simulated data”, lines 122-124)

8) Figure 1 legend—the authors say they are using an AODE? The whole paper is premised around this Bayesian tweak on AODE—what’s happening here?

We thank the reviewer for pointing out the confusion here. SWIF(r) combines sweep-detection statistics using an AODE, as published by Webb et al. (Machine Learning, 2005). We have replaced “Bayesian” with “averaged” in the title to make this clearer, and do not use the word “Bayesian” anywhere in the manuscript.

9) The soft sweep results presented in Supp Fig 17 are unconvincing as the beneficial allele is starting at a very low frequency in these simulations ($f=0.02$) and thus should be equivalent to a hard sweep (See Jensen 2014 for a discussion of this issue). A much more serious treatment of soft sweeps is needed here, including simulating over a range of f and a range of final frequencies. Also the authors should provide further simulations of incomplete sweeps where the final frequency is something different that $p=0.2$ as presented in figure 1.

We thank the reviewer for bringing attention to this point. We have added a new panel and caption to Supplementary Figure 30 that shows that although the initial frequency of the beneficial mutation in our simulations is low, most of the simulations do maintain multiple origins of the beneficial mutation at the time of sampling, with only 1% of soft sweep simulations “hardening” (i.e. losing all but one mutation origin), indicating that SWIF(r) is indeed detecting a class of soft sweeps. We appreciate the point that sweeps beginning from higher initial frequencies will be more difficult to detect, and we have added a clause to the Discussion that acknowledges the low initial frequency that we used for simulations (lines 346-347) As noted in our overview of responses to the reviewer, a full treatment of soft sweeps is beyond the scope of this manuscript. We now state clearly in the results that our implementation of SWIF(r) was trained on hard sweeps (Results, lines 54-59, excerpted below):

Note that we train SWIF(r) on simulations of hard sweeps (Online Methods); our focus here is not on the relative roles of various modes of selection in shaping observed human genomic variation (for recent treatments on this question see (Schridder and Kern, PLoS Genetics 2016; Hernandez et al. Science 2011; Messer and Petrov, Trends in Ecology and Evolution 2013; Garud et al. PLoS Genetics 2015; Schridder and Kern, Molecular Biology and Evolution 2017), although we note that SWIF(r) is extensible to multi-class classification, and could be used in future applications to explore multiple modes of selection. In this study, our focus is on localizing genomic sites of adaptive mutations that have spread through populations of interest via hard sweeps.

Our simulations do include sweeps with a range of final frequencies (0.2, 0.4, 0.6, 0.8, and 1.0), which we have now made clearer in Online Methods and in lines 405-413. In addition, we now provide ROC curves that assess SWIF(r)’s performance against its component statistics in this implementation, CMS, SweepFinder, evoNet, and evoBoosting for simulated incomplete sweeps across these final allele frequencies (Figure 1, Supplementary Figures 6, 9 and 10).

10) Lines 120-122 – the authors claim “we find that modeling the [...] SWIFr is essential for correctly localizing these known adaptive mutations.” That’s a seriously specious claim given that every other method found these regions too and that in figure 2C and 2E other SNPs have higher sweep probabilities than SWIFr. Again, this review is taken aback by the hype the authors are spinning here.

We agree with the reviewer that the section entitled “SWIF(r) correctly localizes canonical adaptive mutations in humans” required significant rewriting for clarity; here we summarize how this section has been altered in the revised manuscript to address this comment from the reviewer. In our genome-wide scan of YRI, CHB and JPT, and CEU genomes from the 1000 Genomes Project, SWIF(r) reports high sweep probabilities at many SNPs and we report and annotate these in Supplementary Tables 2 and 3. Our focus in this manuscript section and in Figure 2 (see our response to point 13 below) is to highlight SWIF(r)’s ability to statistically identify the specific site of an adaptive mutation. This manuscript section focuses on canonical sweep targets in order to further validate SWIF(r)’s output in genome-wide scans. SWIF(r) does

not simply identify the regions highlighted in **Figure 2** as sweep targets (as the reviewer notes, all methods identify these regions as sweep targets), but reports the highest sweep probability at SNPs that experimental studies (all cited in the text) have identified as functionally adaptive. Such experimental evidence is only available for a few genomic targets of sweeps in the human genome; **Figure 2's legend** now reflects this clearly, and we now explicitly state that rs1426654 in **Figure 2C** has the highest sweep probability reported by SWIF(r) in SLC24A5. As we note in point 13 below, in some regions identified as undergoing sweeps by multiple methods, SWIF(r) offers resolution on specific sites selection may be targeting, offering hypotheses for downstream validation (**Figure 2 B,E,F**). Our specific changes to the main text are excerpted below:

Results > “SWIF(r) correctly localizes canonical adaptive mutations in humans”, lines 201-214:
For application to data from phase 1 of the 1000 Genomes Project, we used training simulations from the Schaffner demographic model (Schaffner et al. Genome Research 2005), calibrated SWIF(r) for a training set composed of 0.01% sweep variants and 99.99% neutral variants (Supplementary Figure 1), and applied it to SNP array data from West African (YRI), East Asian (CHB and JPT), and European (CEU) populations. SWIF(r) reports high sweep probabilities at multiple SNPs within known and suspected selective sweep loci in each of these populations (Supplementary Table 2, Supplementary Table 3). Figure 2 illustrates the ability of SWIF(r) to localize sites of adaptive mutations within genomic regions containing canonical sweeps. Adaptive SNPs have been determined via functional experiments in SLC24A5 (Soejima and Koda, International Journal of Legal Medicine 2007), DARC (Reich et al. PLoS Genetics 2005), and HERC2 (Eiberg et al. Human Genetics 2008); we find that modeling the dependency structure among component statistics within SWIF(r) enables statistical localization of these experimentally identified adaptive mutations (Figures 2A,C,D). Methods that treat component statistics as independent, as CMS does, cannot localize these experimentally identified adaptive SNPs (Supplementary Figure 15, Supplementary Figure 16). In CHB and JPT, SWIF(r) recovers a strong adaptive signal in the vicinity of EDAR, offering new hypotheses for targets of selection in this genomic region.

11) Line 127 – the authors identify 126 genes in the genome. This is a very low number given recent efforts to localize selection in the human genome (e.g. Schrider and Kern 2017). How do the authors square this? Moreover 63% overlap with previous sweep scans is pretty bad.

*As stated in our Results, SWIF(r) is trained on simulations of hard sweeps (see **response to point 9 above and Results, lines 54-59**); the SWIF(r) framework is easily extensible to multiple evolutionary scenarios (**Supplementary Figure 30**), but our focus is on localizing adaptive variants in hard sweeps. Therefore, we do not think the number 126 is low, given our focus on hard sweeps.*

In the particular line the reviewer cites in this comment, we have revised the text to read as follows (Results > “SWIF(r) correctly localizes canonical adaptive mutations in humans”, lines 217-219 in the revised manuscript):

Of the 126 genes across these populations with SWIF(r) signals (i.e. at least one variant within the gene has posterior hard sweep probability greater than 50%), 63% were identified in at least one positive selection scan conducted in humans (Supplementary Table 3).

Regarding the overlap of our results with previous selection scans, the literature search in Supplementary Table 3 includes methods applied to different data sets from different population samples over the last decade, that were designed for different purposes (i.e., detecting incomplete sweeps alone versus complete sweeps). We find it difficult to form an expectation of percentage overlap for this broad range of comparisons. Given the new benchmarking we have done, and that our goal in presenting SWIF(r) is to present a method that identifies genomic targets of selection missed by existing methods, we feel we have addressed this comment.

12) Robustness to misspecification section—the authors should present a case of really bad misspecification – no population growth vs the Auton model or similar. Other methods like SHIC and Evolboosting have been shown to be robust to misspecification – how does SWIFr compare to those methods in this regard?

In response to Major Issues #3 above, we simulated a new dataset using a demographic model from Gravel et al. (2011) that includes exponential population growth, where our simulations using the Schaffner mode have no population growth. As mentioned above, a new panel in Supplementary Figure 13, and text in lines 186-191 shows that SWIF(r) is robust to this kind of misspecification.

13) Figure 2 legend – why are the authors highlighting rs3827760 in fig 2E? Other neighboring SNPs have much higher sweep probs.

We thank the reviewer for this question. To clarify, we now note in Figure 2’s legend that rsids and diamonds are used to highlight functionally verified adaptive SNPs; the figure legend details why rs3827760 is classified in the literature as putatively functional. We also state in the following in the revised main text (Results > “SWIF(r) correctly localizes canonical adaptive mutations in humans”, line numbers 213-214):

“In CHB and JPT, SWIF(r) recovers a strong adaptive signal in the vicinity of EDAR, offering new hypotheses for targets of selection in this genomic region.”

In conjunction with our response to point 10 above, we hope this sufficiently addresses the reviewer’s comment.

14) Lines 161-163 – the authors state “genomic regions identified by SWIFr contain annotated functional mutations identified in high coverage...” what does this mean? That in this unpublished manuscript they found a nonsynonymous SNP? So what? Also as stated above the data from Martin et al must be released for this paper to be publishable as is. Alternatively the authors could use a different data set.

We thank the reviewer for pointing out the need to clarify our results; as stated earlier in response to Major Issues #4, for our application of SWIF(r) to the ‡Khomani San, we used SNP array data described in Uren et al. (2016). We then validate that SWIF(r) signals tag functional SNPs using exome sequencing data for the same individuals in the SNP array dataset; these data are described by Martin et al. (in press at Cell; see bioRxiv preprint: <https://www.biorxiv.org/content/early/2017/10/13/200139XXX>). Please see our earlier response to Major Issues #4 regarding data release.

*The results in **Figure 3** were generated by applying SWIF(r) to SNP array data from the ‡Khomani San. The Illumina OmniExpress SNP array used here is relatively sparse (only containing ~700,000 autosomal SNPs); in order to validate whether the high-probability sweep sites SWIF(r) identified were tagging functional variants, we turned to the exome data from the same individuals to look for novel or highly differentiated mutations not on the SNP array that the SWIF(r) signals might be tagging. In the case of the gene ADIPOQ, the SWIF(r) signal at rs6444174 is the closest possible SNP on the array to the nonsynonymous mutation rs113716447 identified in the exome data (see **lines 274-276 of the revised manuscript and our response to Specific Issues #19 below**).*

15) Lines 169-170 – finding a lot of SWIFr signals in the MHC region is not a good thing—that region is notoriously hard to align, assemble, etc. The onus is on the authors to demonstrate that the data quality in that region is half way decent.

*We thank the reviewer for this pointing out the need for clarity here. We only apply SWIF(r) to SNP array data which contains validated MHC variants and on SNPs which passed normal quality control, ie. no more than 5% missing data at a given locus; we do not carry out any followup in this region using the exome dataset precisely because of potential issues with mapping and phasing. We have made this more explicit in the manuscript (**lines 272-274**). We have also added a paragraph to the **Discussion** that further contextualizes this MHC signal, in response to Reviewer #3’s comments; please see that response as well.*

16) Line 175 – the authors state “genes related to adiponectin...” they mean SNPs right? Not genes. The methods is SNP-wise.

*We thank the reviewer for pointing this out. We have edited the sentence to read **“SNPs in genes related to adiponectin (ADIPOQ, PEPD, DUT, ASTN2) have among the highest posterior sweep probabilities (all ≥ 75%)”** (lines 250-251 in revised manuscript).*

17) Table 1 indicates that the only functional category that survives FDR correction is adiponectin. The authors should remove all of the discussion of non-significant functional categories from the paper.

After calibrating the probabilities reported by SWIF(r), two more functional categories survived FDR correction: Body Mass Index and gamma-Glutamyltransferase (Figure 3 and its caption now reflect this). Body Mass Index is also related to metabolism and obesity. Most highlighted genes in Table 1 are found in either the Adiponectin or Body Mass Index gene sets, and all are associated with metabolic and obesity-related phenotypes. For these reasons, and because we present FDR results for this analysis, we have kept our discussion of functional categories as is.

18) Line 183—the authors have belabored the point that combining signals from multiple statistics aids power and that they don't rely on empirical distributions—why are they now giving empirical p-values from individual features???

We appreciate the reviewer's question. In the section of the manuscript the reviewer is referring to, our goal is to determine the biological function of variants with high sweep probabilities in our application of SWIF(r) to \ddagger Khomani San SNP array data. rs6444174 has a high sweep probability as reported by SWIF(r) (92%), and has been identified as associated with adiponectin levels and body mass index in a genome-wide association study of African Americans. We note that this same rsid is an empirical genomic outlier for three univariate selection statistics to both underscore why its SWIF(r) sweep probability is so high, and to illustrate which statistics are driving this signal.

19) Lines 196-197—this makes no sense with respect to the paper. The authors have gone on about how accurate their method is, but now are pointing to a random nonsynonymous SNP 1kb away from a SWIFr identified SNP. Sorry this is bait and switch.

We thank the reviewer for bringing attention to this point. As we mentioned above (see our earlier response to Specific Issue #14), the nonsynonymous SNP is only present in the exome data, and not on the sparse SNP array that was used for identifying SWIF(r) signals in Figure 3. In the case of ADIPOQ, the SWIF(r) signal was the closest possible SNP on the SNP array to the nonsynonymous mutation identified in the exome data. We therefore offer this as evidence for a putative driving mutation that is being tagged by the SWIF(r) signal <1kb away. We have made this point more explicit in the manuscript (Results > "Adaptive loci in the \ddagger Khomani San are enriched for metabolism- and obesity-related genes" > "Exome-based support for targets of selection identified by SWIF(r)", lines 274-276):

In ADIPOQ, we identify a missense mutation, rs113716447, for which the nearest SNP that is present on the SNP array is rs6444174 (less than 1kb away); rs6444174 has a calibrated SWIF(r) sweep probability of 90%, the highest in ADIPOQ (Figure 4).

20) Line 233 – the claim that SWIFr “localizes experimentally validated adaptive mutations using genomic data along” belies what the authors are actually doing in much of the results. See comment above. This sentence should be softened considerably.

*Based on our response to Specific Issues #10, we stand by this claim, which is stated in reference to **Figure 2**. For those canonical sweep targets in the 1000 Genomes where molecular experiments have identified a site containing a functional beneficial mutation, SWIF(r) assigns that site the highest sweep probability in that region. So we respectfully leave this sentence as it was in our original submission, in hopes that the new analyses and text revisions we have made in response to other comments by this reviewer (in particular about the analysis of SNP array and exome data from the ‡Khomani San) and the other reviewers lend support to our claim.*

21) Line 227—the algorithm presented is not serious or noteworthy. This should be removed from the discussion.

We respectfully maintain that this algorithm is of value, and point to Reviewer 3’s comments that addressing ascertainment bias is important because SNP arrays are still widely used for assaying genetic variation in diverse human populations.

22) Lines 229-231—only one function survives the correction for multiple tests. This statement is overblown.

Given our response to point 17 above, we feel this statement is now accurate and respectfully leave it as is.

23) Line 235 – the statement that SWIFr advances the methodologies used in other ML approaches by learning the joint distributions of features is untrue—this is a feature of the other methods too. Again, compare your method to those before blowing your own horn.

*We feel this has been addressed with new benchmarking in our revised manuscript, and with the new language that we added in response to Specific Issues #4 above. As described in response to Major Issues #1 above, we now compare SWIF(r) to a broader array of state-of-the-art methodologies including evolBoosting and evoNet, and demonstrate that in most scenarios, SWIF(r) outperforms current competitors. We have softened the language at the start of the **Discussion** to read as follows:*

Discussion, lines 315-321:

Composite classification frameworks such as SWIF(r) quantitatively ground a common qualitative approach used in scans for adaptive sweeps based on summary statistics: evidence for selection at a locus is considered stronger when extreme values are observed for more than one statistic (Figure 3A). Furthermore, machine-learning approaches like SWIF(r) that learn joint distributions of selection statistics can detect sweep events that individual univariate statistics cannot (Supplementary Figure 16).

SWIF(r) additionally reports calibrated probabilities assessing evidence for selective sweeps site-by-site, resulting in a transparent probabilistic framework for localizing adaptive mutations.

24) Line 238—the authors need to demonstrate that other methods are biased in their handling of missing data before making this claim. They have not done so. Further the authors have not demonstrated the robustness of their own method to missing data and need to do so via simulations.

*We appreciate the reviewer's comment. Please see our overview response to Reviewer #2's comments concerning our clarification of undefined statistics versus missing data. We have removed all mentions of how window-based methods handle undefined statistics, except in lines 19-21 of the revised manuscript in which we acknowledge that this issue does not typically arise for window-based component statistics. Please see our response to Specific Issues #5, as well as **Supplementary Figure 33**, with regards to our analysis demonstrating robustness to undefined statistics.*

25) This brings me to another point—the authors have engaged in a major bait and switch in this paper—they benchmark the method using a feature vector of 5 stats, but only analyse the data using 4. The entire simulation section of the paper should be repeated to reflect what they have done in the empirical analysis.

We appreciate the reviewer's suggestion to remove ΔiHH from our implementation of SWIF(r) throughout the manuscript and have done so in the revised manuscript (see manuscript section entitled "Performance of SWIF(r) using simulated data" and Figures cited in that section).

26) Line 239 – how is SIDT2 reflective of “finer localization?” Because their ad hoc, unpublished naïve bayes classifier missed it?!?

*We have softened the language based on this reviewer's comments, to acknowledge that incorporating correlation of summary statistics is a feature of other methods as well; **Discussion, lines 317-319, now reads:***

Furthermore, machine-learning approaches like SWIF(r) that incorporate joint distributions of selection statistics can detect sweep events that univariate statistics cannot (Supplementary Figure 28).

27) Line 242—ABC does not have likelihoods. That's the whole point.

*We thank the reviewer for pointing out this typo. We have altered the sentence to read: "**While approaches such as Approximate Bayesian Computation can exploit higher-dimensional correlations...**" (Discussion, lines 323-325)*

28) Line 244—where is the evidence that the Peter et al. method is prone to overfitting? This is just a baseless claim.

We thank the reviewer for this comment; we have altered the sentence to read: "...this comes at the cost of genome-scale tractability, and can be vulnerable to the curse of dimensionality" (Discussion, lines 323-325) and have added a citation to Sheehan and Song (PLoS Computational Biology, 2016) where this claim is made.

29) Line 244—again the authors say they are using AODEs...

We thank the reviewer for pointing out the confusion in our terminology; as noted above in our response to Specific Issues #8, we have removed "Bayesian" from the title and abstract, and it does not appear in the manuscript.

30) Line 248-249 – this bit about "cryptic assumptions about class sizes" belies the authors misunderstanding of supervised ML. None of this is true, and this section should be removed.

We thank the reviewer for this comment, and we have removed this sentence.

31) Line 288—again this baseless claim of other methods not being able to deal with missing data.

We thank the reviewer for this comment, and we have removed all discussion about methods other than SWIF(r) and CMS dealing with missing data. We refer to our overview above, where we address the apparent confusion over missing genotype data versus undefined component statistics; throughout the manuscript, we are referring to the latter problem, one that only arises for site-based composite methods. We have made this explicit in lines 19-21 in the revised manuscript.

32) Line 303—100 simulations is way too small for accurate training. The authors should increase this by ~ and order of magnitude.

We thank the reviewer for pointing out the need to clarify the details of our training set. Since SWIF(r) operates site-by-site, 100 neutral simulations results in ~400,000 training points for learning neutral distributions. We also carried out 9,000 simulations containing a sweep, which translates to 9,000 training points for learning sweep distributions. We have made both of these points more explicit in the text (see Online Methods > "Simulation of haplotypes for 1000 Genomes analysis", lines 398-415).

33) Lines 327-329—how were these migration rates arrived at? What does a pulse migration with a rate mean? How was the “match” evaluated quantitatively? This seems very ad hoc as presented.

We thank the reviewer for this comment. The term “pulse migration” refers to migration in one generation; the migration rate is the fraction of the target population replaced by haplotypes from the source population at the given generation. We have added a section to the supplementary text, entitled “Calculation of migration rates from YRI and CEU to ‡Khomani San”, that describes exactly how we arrived at the migration rate estimates. Briefly, we use tract length distribution analysis from Uren et al. (2016) to calculate a matrix of ancestry proportions from YRI, CEU, and San ancestry in the ‡Khomani San at every generation from 14 generations ago to present. We then infer one-generation pulse migration rates (again defined as the replacement fraction in the ‡Khomani San in one generation) that result in the same present-day ancestry proportions, with the YRI migration pulse taking place at 14 generations ago and the CEU pulse at 7 generations ago (following Uren et al. 2016). This is now presented in detail in the new supplementary text entitled “Calculation of migration rates from YRI and CEU to ‡Khomani San”.

34) What do the “full covariance matrices” mean in the context of the Gaussian mixture model? This section is quite unclear.

We thank the reviewer for pointing out the need to clarify the meaning of “full covariance matrices” -- what we mean by this is that the covariance matrices have nonzero off-diagonal elements. We have added this clarification to the text (Online Methods > “Implementation of the Classifiers” > “Implementation of SWIF(r)”, lines 472-473).

35) ROC analysis—the authors should vary the sweep call threshold, not the prior on a sweep here.

*We thank the reviewer for this comment. We initially chose to vary the prior instead of the sweep call threshold because of the interpretability of using 50% as a threshold, but varying the threshold would also be a valid way to generate these curves. We have added **Supplementary Figure 4** that shows that these methods result in equivalent ROC curves, and mention this in Results > “Implementation of SWIF(r)” > Calibration of posterior probabilities calculated by SWIF(r) lines 103-107:*

The calibrated probabilities reported by SWIF(r) can be interpreted directly as the probability that a site contains an adaptive mutation, or fed into a straightforward classification scheme by way of a probability threshold; in this study, we classify sites with a posterior probability above 50% as adaptive SWIF(r) signals. The classifier may be tuned by altering either this threshold or the prior sweep probability π (Supplementary Figure 4).

36) Supplemental Figure 18 – the authors say that the method can be extended but the results presented look terrible. This isn't convincing.

*We refer the reviewer to our overview, in which we emphasize that SWIF(r) is designed as an extensible framework, both in terms of the component statistics used and the classification scenarios. In this figure (now **Supplementary Figure 31**), we show an illustrative example of how SWIF(r) can be extended to more than two classification scenarios. We acknowledge in the figure caption that based on the four component statistics we use in this manuscript, SWIF(r) does not have enough power to distinguish between different sweep timings, but that other component statistics could be incorporated that are more sensitive to the time of sweep onset.*

Reviewer #3 (Remarks to the Author):

Alpert Sugden et al. develop a novel statistical learning approach for localizing selective sweeps, which they term SWIF(r). In contrast to other recent statistical learning approaches (such as that of S/HIC of Schrider and Kern) that also employ multiple summary statistics as evidence for sweeps, the SWIF(r) approach developed here can provide a score per site rather than per region, enabling it to localize sweeps at the polymorphism level. In addition, this new method is robust to missing data, whereas prior statistical learning approaches required extensive and careful data filtering so as to not bias their results. The authors also apply their new statistic to data from the ‡Khomani San, and characterize a number of regions.

I would like to start out by stating that the manuscript is very well-written, and that I appreciate that the authors included line numbers to make refereeing easier. In addition, I believe that developing statistical learning approaches for localizing adaptive alleles is a hot topic, and the proposed method will likely be of interest to a number of researchers. I am less able to assess the novelty of the empirical findings, but the methodology presented is certainly both innovative and rigorous. Therefore, based solely on the proposed statistical approach, I believe this manuscript is high priority.

Some of the points I find particularly interesting are that each polymorphic site is assigned a posterior probability of being a sweep, that the authors investigated how ascertainment bias could be accounted for (often ignored these days, but important as SNP chips are still common), and that their method can detect both partial hard sweeps to low frequency (e.g., 20%) and soft sweeps from standing variation (though only hard sweeps were used in training). Below are my additional comments and/or concerns of the current manuscript.

We thank the reviewer for reading the manuscript closely and contextualizing its contribution; these thoughtful critiques and suggestions have improved the manuscript. We detail our responses to the specific comments below.

Major comments:

(1) The scan in the ǀKhomani San shows a strong peak in the MHC region. This region is classically thought to be undergoing balancing selection, and a number of approaches for detecting long-lived balancing selection are able to detect the HLA genes within this region. Long-lived balancing selection, which tends to yield increased genetic diversity and decreased population differentiation (F_{st}), appears to be incompatible with signatures of selective sweeps. It would be helpful if the authors could discuss why this region would be identified by methods for detecting selective sweeps. Could it be that the SNPs that are identified by SWIF(r) have low scores for methods that detect balancing selection (i.e., these high probability SNPs would not be detected by methods to identify balancing selection)? Or is there something specific happening in the ǀKhomani San population, which would indicate that this locus should be evolving under recent positive selection rather than long-lived balancing selection?

While the MHC region is a classical example of long-lived balancing selection, alternative selection scenarios have also been posed, including recurrent selective sweeps for different haplotypes and fluctuating directional selection whereby selective targets change or alternate in response to rapid fluctuation in pathogen-mediated pressures (Spurgin and Richardson, Proceedings of the Royal Society B 2010). Our MHC component statistics are qualitatively similar to signals elsewhere in the genome (Supplementary Figure 26, new to this revised manuscript) suggesting that long-lived balancing selection (i.e. heterozygote advantage or frequency dependent selection) is not the primary cause of selection in our SWIF(r) results. However, we remain cautious about over-interpreting the MHC signals because SNP array ascertainment bias in this region is likely worse than elsewhere (as the San contain a huge diversity of HLA alleles, Hilton et al. PLoS Genetics 2015) and we are unable to validate them in the exome follow-up due to poor read mapping sensitivity in the standard exome variant calling pipeline. We have added the following text to the Discussion (lines 370-376):

“In this selection scan, we also see an abundance of SWIF(r) signals in the MHC region involved in immunity. It is possible that this signal reflects balancing selection, which is the mode of selection canonically thought to be occurring within this region (Hedrick, Genetica 1998). Indeed, it has been shown that the signatures of balancing selection and incomplete or recurrent sweeps may be similar to signatures of positive selection (Andrés et al. Molecular Biology and Evolution 2009; Lenz et al. Molecular Biology and Evolution 2016). We note, however, that other studies using different methodologies have detected signatures of directional selection in the MHC in human populations (Albrechtsen et al. Genetics 2010; Zhou et al. PLoS Genetics 2016), and others have noted that fluctuating directional selection is a possible mechanism for pathogen-mediated selection in this region (Spurgin and Richardson, Proceedings of the Royal Society B 2010).”

(2) If single nucleotide statistics (such as F_{st} and Delta DAF) were not used, and instead haplotype-based statistics (such as iHS and Delta EHH) were only used, would correcting for ascertainment matter much? Conrad et al. (2006; Nat Genet 38:1251-1260) indicated (Figure 3

of their article) that the effect of ascertainment bias is removed when using haplotype-based statistics.

*We thank the reviewer for raising this question. While we note that iHS is robust to ascertainment bias, we would not necessarily expect $XP-EHH$ to be, since ascertainment bias could lead to inflation of haplotype lengths in populations that are not represented on the SNP array relative to populations that are (since less variation will be represented for the former populations); we note that the Figure 3 of the Conrad et al. article is looking at haplotype statistics within populations. We do appreciate the point, however, that different statistics would be more robust to ascertainment bias than others; in particular, in future applications, we would like $SWIF(r)$ to be able to incorporate any number of statistics, including those based on the SFS, which would certainly be more vulnerable to ascertainment bias than the statistics we use for this implementation. To acknowledge this point, we have added the following sentence to the **Discussion** (lines 310-312 in the revised manuscript):*

While some of the component statistics we use here may be fairly robust to ascertainment bias, this algorithm also enables future use of component statistics that are more vulnerable to ascertainment bias, such as SFS statistics, within $SWIF(r)$'s framework.

(3) The authors indicate that they expect that $SWIF(r)$ is robust to background selection, because Enard et al. demonstrated that haplotype-based statistics are robust to background selection. However, two of the statistics, and indeed those with the most power (F_{st} and Delta DAF from what I can tell from Figure 1D and Supplementary Figure 17) are not haplotype-based, and therefore likely influenced by background selection. In addition, based on comment 2 above, the argument that haplotype-based statistics are robust to background selection can also then be used to argue that $SWIF(r)$ is robust to ascertainment bias. Because background selection is a ubiquitous force (e.g., McVicker et al. 2009, Comeron 2014), it would be helpful if the authors tested their method's robustness to this scenario through some illustrative simulations.

*We thank the reviewer for this comment, and we agree that illustrating robustness to background selection in simulation is important. Please see our response to "Major issues" #6 from reviewer 2 for details on the new simulations that we generated in order to demonstrate this robustness, as well as **Results > "Performance of $SWIF(r)$ using simulated data"** (lines 192-200), **Online Methods > "Simulation of background selection"** (lines 449-464), and **Supplementary Figure 20** in the revised manuscript.*

Minor comments:

- Citations of figure panels within the text use uppercase letters (e.g., Figure 1A or Figure 1B), whereas the panels are labeled using lowercase letters (e.g, a and b) within each figure.

We thank the reviewer for catching this, and have remedied it.

- On page 11, line 232 the authors write "The composite classification framework of SWIF(r) quantitatively grounds a common qualitative approach used in scans for adaptive sweeps". Though I understand what the authors are getting at, likelihood ratio approaches essentially do to the same thing, as the likelihood ratio quantitatively assess the probability of observing the data under the alternative hypothesis of selection relative to the probability of observing the data under the null hypothesis of neutrality. I think the authors mean that the qualitative approach is common when using summary statistics, such as the component statistics used by SWIF(r).

We agree with the reviewer that our original writing oversimplifies available approaches for selection scans. We have altered the start of the paragraph the reviewer cites here (including the sentence cited in this particular reviewer comment) as follows:

(Discussion, lines 315-320):

Composite classification frameworks such as SWIF(r) quantitatively ground a common qualitative approach used in scans for adaptive sweeps based on summary statistics: evidence for selection at a locus is considered stronger when extreme values are observed for more than one statistic (Figure 3A). Furthermore, machine-learning approaches like SWIF(r) that learn joint distributions of selection statistics can detect sweep events that individual univariate statistics cannot (Supplementary Figure 16). SWIF(r) additionally reports calibrated probabilities assessing evidence for selective sweeps site-by-site, resulting in a transparent probabilistic framework for localizing adaptive mutations.

Reviewer #1:

Remarks to the Author:

I have read your answers to the reviewers and I am glad that you have carefully addressed my previous comments. You have included isotonic regression to calibrate probabilities and you are now using FDR/power curves for method comparisons. I have no further comments to provide.

Reviewer #2:

Remarks to the Author:

Review of Sugden et al revision

The authors have improved the scholarship and quality of the analyses to a degree in this revision but they still have not adequately addressed many of my major concerns. Moreover they continue to make some misleading claims--why do this guys? My critiques fall along the following lines:

- 1) Proper comparison to other tools for finding sweeps- the authors have added comparisons to SweepFinder, evolBoosting, and a modified evoNet. First the comparison to SweepFinder is inappropriate—that is specifically for finding completed sweeps and is known to lack power in comparison to ML methods. If the authors want to use a univariate summary they should compare to iSAFE (by the way the inclusion of iSAFE as a component as is done in the supplement is misleading—I'll come back to this point). Secondly modifying the summary stats used in evoNet is unhelpful—this does not present a fair comparison to the way the method was designed or known in the community—you need to do the actual comparison. Lastly I realize upon reading the revision that no where through this do the authors acknowledge that they are using data from multiple populations in comparison to methods that were designed for single populations (save for CMS). This should be brought to the forefront of the paper and is not meant as a criticism.
- 2) The authors make the claim in the responses and in lines 327-329 that somehow SWIFr is unique in being able to include new summary stats- this is true for all supervised ML methods including evolBoosting (which is just using boosted regression on a collection of summary stats...). The authors need to delete this false claim.
- 3) The authors have not followed my very important suggestion of redoing their entire analysis using a modern demographic model with strong growth such as Auton et al.. This is still a deal breaker and what they present is inappropriate. What the authors instead do is show that model-misspecification between the Schaffner and (now dated) Gravel models do lead to changes in performance, albeit small. This is problematic and does not satisfy this reviewer.

4) The authors have not provided the misspecification results that I asked for—they were asked to perform gross model misspecification (equilibrium population size training set vs Auton et al testing set) so that a proper comparison could be done among modern ML methods. This needs to be done.

5) The details surrounding the simulation of background selection are unclear and sometimes inappropriate. First lines 457-458 “Note that for testing the robustness of SWIFr we only considered sites from these simulations that landed in exons or UTRs”—what does this mean? Further the rescaling of factors that you are doing (lines 461-464) is inappropriate for models with selection. This is well known (Uricchio and Hernandez 2014) and is described for background selection specifically in the SFS_CODE manual. The authors need to redo these simulations without this parameter rescaling for it to be valid. As an aside Philip Messer incorrectly points to parameter rescaling as acceptable in the SLiM manual.

6) The authors are still being deliberately misleading about the issue of “missing components.” The authors say that “no imputation or compensation mechanisms are necessary in the case of undefined component statistics” – this just isn’t true. First the authors lose a bunch of power if iHS can’t be calculated, and to lesser degrees they lose power if the other single stats can’t be calculated. Despite this fact the authors make the misleading claim in lines 484-486 that “SWIFr loses little power...” Moreover, without any component or group of components the model probabilities need to be recalibrated. Further any other method could handle missing components similarly by training classifiers with subsets of stats—this is not unique to your method.

7) The claim in line 32 that SWIFr is unique in learning joint distributions of the summary stats is also not true. Thus in lines 29-34 the only true claim to uniqueness in my opinion is that SWIFr calculates calibrated probabilities—you should be acknowledging the field here more than you are.

8) My point 32 in my original review still stands. 100 simulations is way too small—you are claiming that 100 realizations of the ARG gives you 400,000 sites for training, however those sites have correlated histories. My point is that 100 realizations from the ARG is far too little. This number need to be pushed up considerably.

Reviewer #3:

Remarks to the Author:

Alpert Sugden et al. have made a tremendous effort in revising their manuscript. It is clear that the manuscript has greatly improved with the aid of reviewer comments, and I am satisfied with the way that my prior comments were addressed. I have no further concerns.

Reviewers' comments:

Reviewer #1 (Remarks to the Author):

I have read your answers to the reviewers and I am glad that you have carefully addressed my previous comments. You have included isotonic regression to calibrate probabilities and you are now using FDR/power curves for method comparisons. I have no further comments to provide.

We thank reviewer 1 for his previous critiques; in particular, reporting calibrated sweep probabilities is now a unique contribution SWIF(r) makes to the sweep-detection literature.

Reviewer #2 (Remarks to the Author):

Review of Sugden et al revision

The authors have improved the scholarship and quality of the analyses to a degree in this revision but they still have not adequately addressed many of my major concerns. Moreover they continue to make some misleading claims--why do this guys? My critiques fall along the following lines:

1) Proper comparison to other tools for finding sweeps- the authors have added comparisons to SweepFinder, evolBoosting, and a modified evoNet. First the comparison to SweepFinder is inappropriate—that is specifically for finding completed sweeps and is known to lack power in comparison to ML methods. If the authors want to use a univariate summary they should compare to iSAFE (by the way the inclusion of iSAFE as a component as is done in the supplement is misleading—I'll come back to this point). Secondly modifying the summary stats used in evoNet is unhelpful—this does not present a fair comparison to the way the method was designed or known in the community—you need to do the actual comparison. Lastly I realize upon reading the revision that nowhere through this do the authors acknowledge that they are using data from multiple populations in comparison to methods that were designed for single populations (save for CMS). This should be brought to the forefront of the paper and is not meant as a criticism.

We thank the reviewer for pointing out the need for a consistent approach for choosing methods against which SWIF(r) is benchmarked. Our goal is to benchmark against methods that were specifically designed to scan genome-wide datasets and classify genomic sites or regions as neutrally evolving or containing adaptive mutations (see line 48-50). Following this approach, we believe there is merit in benchmarking against SweepFinder: SweepFinder is computed site-by-site, and SFSelect and S/HIC also benchmark against it. We do acknowledge on lines 134-145 that SweepFinder was designed to identify complete sweeps in a population of interest; we have left comparisons against SweepFinder in this version of the manuscript.

iSAFE is explicitly designed for fine-mapping beneficial alleles in a region previously determined to be undergoing positive selection; the authors state in their introduction that

“iSAFE requires that the broad region under selection is identified using existing methods”. We think it inappropriate to alter iSAFE from its original intended application, and maintain that SWIF(r) and other composite machine learning frameworks can leverage powerful statistics like iSAFE to increase power when detecting adaptive mutations (Supplementary Figure 29). We have altered the caption of this figure to note iSAFE’s intended application, excerpted below:

One such statistic, iSAFE (preprint available at <https://www.biorxiv.org/content/early/2017/10/01/139055>), is an approach that ranks mutations within a genomic region that has been predetermined to contain a selective sweep. As shown below (light blue ROC curve), iSAFE is much more powerful than the component statistics used in this implementation of SWIF(r) at localizing beneficial mutations...

We agree with the reviewer that benchmarking against evoNet with the 345 statistics used in the original publication is most appropriate. We have now done this (Supplementary Figure 10); computing the 345 component statistics in Sheehan and Song (2016, PLoS Computational Biology) required help from Sara Sheehan Mathieson (email correspondence pasted at the end of our response to this point). We find that SWIF(r) outperforms evoNet across all sets of parameters, despite the fact that evoNet uses a much larger set of component statistics. We believe that this is because the vast majority of the 345 statistics are not informative for distinguishing between selective sweeps and neutral evolution (we suspect that these statistics are instead informative for demography, which evoNet also infers). We illustrate this in panel F of Supplementary Figure 10, where we calculate the Kullback-Leibler (K-L) divergence from the sweep distributions to neutral distributions for each component statistic (with K-L divergence values near zero indicating majority overlap of the two distributions). The outliers are statistics known to be informative for selection, such as Tajima’s D, and Garud et al.’s (2015, PLoS Genetics) H statistics, meaning these are the most discriminative statistics for these two evolutionary scenarios. We excerpt the caption for Supplementary Figure 10F below:

Panel F: Given that evoNet uses 345 statistics while SWIF(r) uses only four, the superior performance of SWIF(r) is surprising at a glance; we believe that this is likely due to the fact that the majority of the 345 statistics are not informative for selection (but are likely informative for demographic history, which evoNet also infers). To demonstrate this, for each statistic, we have computed the Kullback-Leibler divergence from the sweep distribution to the neutral distribution of the statistic calculated in simulations. A K-L divergence near zero represents near-total overlap of the two distributions, which indicates that the statistic is not informative for separating neutral and sweep windows. We find that for the vast majority of the 345 component statistics, the K-L divergence is near zero; a few exceptions are the H statistics developed by Garud et al. to detect hard and soft sweeps, a few measures of linkage disequilibrium, number of singletons, and Tajima’s D.

As the reviewer suggested, we now clarify that SWIF(r) uses comparisons across populations to calculate per-site sweep probabilities (lines 113-114 and 152-155).

From: **Sara Mathieson** <smathiel@swarthmore.edu>
Date: Mon, Oct 16, 2017 at 2:51 PM
Subject: Re: question about evoNet
To: Sara Mathieson <sara.k.mathieson@gmail.com>
Cc: Lauren Sugden <lauren.v.sugden@gmail.com>

Hi Lauren,

Okay, I'm attaching a zip file with some updated source to compute the statistics (instructions in the README). The code changes are only for newer versions of java, they don't change how the statistics are computed. The setup is slightly clunky, but it should work for most situations. The idea is that you could have multiple demographies (i.e. population size change histories) *and* multiple regions with different types of selection for each demography. Each demography has a separate folder ("demo0", "demo1", etc) and within these folders the regions should be called "data0.msms", "data1.msms", etc. Here is an example command line and then what each parameter means:

```
java -jar -Xmx5G statsZI.jar --beginDemo=0 --endDemo=1 --numPerDemo=1  
--msmsFolder=example/data/ --statsFolder=example/stats/
```

-Xmx5G: this is probably overkill, adjust for your own situation and how much data you have

--beginDemo: usually 0 (start with the first demography)

--endDemo: excludes end point, so if you only have one demography, use 1 for this parameter (in the paper: 2400)

--numPerDemo: number of regions (i.e. datasets) within each demography (in the paper: 160, 40 for each type of selection)

--msmsFolder: path to the data

--statsFolder: folder where the statistics should be output

There will be one stats file for each demography, and each dataset will be on a separate line. So if you run the example in the README, you should get one output stats file with one line of stats (and a header for the stats).

If you want to modify the statistics, you can do so from source and recompile the jar file. Or if you just want to remove some statistics, you can post-process the output files.

Let me know if that makes sense or if you have any questions!!

-Sara

2) The authors make the claim in the responses and in lines 327-329 that somehow SWIFr is unique in being able to include new summary stats- this is true for all supervised ML methods

including evolBoosting (which is just using boosted regression on a collection of summary stats...). The authors need to delete this false claim.

We have now altered *lines 332-334* to emphasize that this feature is not unique to SWIF(r). The text now reads: *“Future applications of SWIF(r) and other composite sweep-detection frameworks can easily incorporate new summary statistics such as iSAFE, which ranks candidate adaptive mutations in a predefined region under selection (Supplementary Figure 29).”*

3) The authors have not followed my very important suggestion of redoing their entire analysis using a modern demographic model with strong growth such as Auton et al.. This is still a deal breaker and what they present is inappropriate. What the authors instead do is show that model-misspecification between the Schaffner and (now dated) Gravel models do lead to changes in performance, albeit small. This is problematic and does not satisfy this reviewer.

As we understand it, the reviewer would like to see SWIF(r) trained using a demographic model with strong recent growth. We believe we have addressed this, as we detail below, and we are unsure what exact model the reviewer means when the reviewer refers to “Auton et al.” The paper we think the reviewer is referring to is “A global reference for human genetic variation” by Auton et al. (2015, Nature), in which the authors apply the pairwise sequentially Markovian coalescent (PSMC) method (Li and Durbin 2011, Nature) to infer population size changes over time for 26 populations within the 1000 Genomes phase 3 dataset. This analysis infers effective population size as a piecewise constant function over time for each population. We cannot glean exact population size estimates from the supplementary files released by Auton et al. (2015, Nature); while the authors release files they state are output from PSMC, the PSMC utilities for processing output files do not execute on the supplementary files (Auton et al. Supplementary Information section 9.4; files at <ftp://ftp.1000genomes.ebi.ac.uk/vol1/ftp/release/20130502/supporting/psmc/>).

It is important to note that PSMC infers a subset of the parameters required for a full demographic history of populations: in particular, PSMC does not infer divergence times or migration rates between populations. PSMC has further been demonstrated to have a large degree of uncertainty for recent time inferences (less than 20,000-30,000 years ago; see Schiffels and Durbin 2014, Nature Genetics, and Sheehan, Harris and Song 2013, Genetics). For these reasons, we do not believe it is possible to extract a usable demographic model from Auton et al. (2015, Nature) for simulating training data for SWIF(r).

Since the reviewer’s primary concern centers on the strength of recent population growth in our training simulations, we would offer that the demographic model we use from Gravel et al. (2012, PNAS) does indeed include rates of population growth that are comparable in strength with those inferred by Auton et al. (2015, Nature) with PSMC. We have attached a table at the end of this response in which we scale the Gravel et al. (2012, PNAS) population size parameters before and after recent population expansion to obtain values that we can compare to the scaled populations sizes inferred by Auton et al. in Extended Data Figure 7 (also attached

at the end of this response). Since Auton et al. use a range of mutation rates, we have included scaled sizes using three different mutation rates (the extremes used by Auton et al., and the value used by Gravel et al.) Although we acknowledge that the expansion inferred by Auton et al. has a shorter duration (15kya-present versus 23kya-present for Gravel et al.), for all mutation rates, the population size change inferred by Gravel et al. for non-African populations is more extreme than that inferred by Auton et al. (we have highlighted one of these comparisons in bold for purposes of clarity). Neither model predicts a strong recent expansion for the African YRI population.

We have additionally updated **Supplementary Figure 13B** to include ROC curves comparing SWIF(r)'s performance when trained on the Schaffner model (with no population growth) and tested on the Gravel model with doubled rates of recent exponential expansion over the last 23,000 years. The doubled rates are 0.76% and 0.96% for Europe and East Asia respectively, which corresponds to present-day population sizes that are 15-fold (Europe) and 30-fold (East Asia) larger than the original sizes inferred by Gravel et al. (2012, PNAS). We find that SWIF(r) is robust, even to this dramatic misspecification. We have updated **lines 188-196** in the main text, and the caption of **Supplementary Figure 13**, to reflect this analysis.

4) The authors have not provided the misspecification results that I asked for—they were asked to perform gross model misspecification (equilibrium population size training set vs Auton et al testing set) so that a proper comparison could be done among modern ML methods. This needs to be done.

In addition to our response point 3 above regarding the robustness of SWIF(r) to misspecification of recent growth, we note that this request appears quite different from what the reviewer requested in the first round of review. This reviewer's previous critique regarding demographic misspecification is excerpted below (More specific issues, #12):

Reviewer 2: "Robustness to misspecification section—the authors should present a case of really bad misspecification – no population growth vs the Auton model or similar. Other methods like SHIC and Evolboosting have been shown to be robust to misspecification – how does SWIFr compare to those methods in this regard?"

As we noted in our response to the first round of reviews:

*"In response to Major Issues #3 above, we simulated a new dataset using a demographic model from Gravel et al. (2011) that includes exponential population growth, where our simulations using the Schaffner mode have no population growth. As mentioned above, a new panel in **Supplementary Figure 13**, and text in **lines [188-196 in the newly revised manuscript]** shows that SWIF(r) is robust to this kind of misspecification."*

Based on the critiques received during the first round of review, we did what this reviewer requested. We were not asked to train using equilibrium population size (no population size changes). S/HIC and Evolboosting do generate training simulations for single populations

*without population size changes; however, as this reviewer noted earlier in this set of critiques (point 1), SWIF(r) uses cross-population comparisons in order to compute per-site sweep probabilities. This reviewer also noted that this is a feature of SWIF(r) that should be acknowledged. We see no obvious way to compare the robustness of SWIF(r) to demographic misspecification using equilibrium population sizes against methods that focus on single populations, given that SWIF(r) uses comparisons between multiple populations with a shared divergence history to localize adaptive mutations. Simulating individual populations with differing effective population sizes but no shared divergence or migration renders cross-population comparisons impossible. **Supplementary Figure 13B** shows robustness of SWIF(r) to misspecification of recent population growth, as this reviewer initially requested (see also our response to point 3 above), and **Supplementary Figure 13A** shows robustness of SWIF(r) to misspecification of divergence times, post-divergence bottlenecks and different migration events.*

5) The details surrounding the simulation of background selection are unclear and sometimes inappropriate. First lines 457-458 “Note that for testing the robustness of SWIFr we only considered sites from these simulations that landed in exons or UTRs”—what does this mean? Further the rescaling of factors that you are doing (lines 461-464) is inappropriate for models with selection. This is well known (Uricchio and Hernandez 2014) and is described for background selection specifically in the SFS_CODE manual. The authors need to redo these simulations without this parameter rescaling for it to be valid. As an aside Philip Messer incorrectly points to parameter rescaling as acceptable in the SLiM manual.

*We thank the reviewer for bringing to our attention the issues with parameter rescaling in the presence of background selection. We have regenerated these simulations without parameter rescaling, and have updated **Supplementary Figure 14** accordingly, showing that the robustness of SWIF(r) to background selection still holds. We also thank the reviewer for pointing out the need to clarify how we tested for robustness to background selection. We have altered **lines 467-471** (excerpted below), and have also updated the caption of **Supplementary Figure 14**.*

“For testing the robustness of SWIF(r) to background selection, we trained SWIF(r) to distinguish between neutral mutations (from simulated neutral regions), and adaptive mutations (from simulated sweep regions), and then applied this classifier to variants in simulated genic regions undergoing background selection (mutations in UTR, exonic, and intronic sites).”

6) The authors are still being deliberately misleading about the issue of “missing components.” The authors say that “no imputation or compensation mechanisms are necessary in the case of undefined component statistics” – this just isn’t true. First the authors lose a bunch of power if iHS can’t be calculated, and to lesser degrees they lose power if the other single stats can’t be calculated. Despite this fact the authors make the misleading claim in lines 484-486 that “SWIFr loses little power...” Moreover, without any component or group of

components the model probabilities need to be recalibrated. Further any other method could handle missing components similarly by training classifiers with subsets of stats—this is not unique to your method.

It is not our goal to be misleading, and we maintain that the ability to run SWIF(r) and interpret its output without imputing undefined statistics makes SWIF(r) easier to use than other composite sweep-detection methods. We have altered lines 31-32 to read “SWIF(r) can be run without imputing undefined statistics.” We have also altered lines 491-493 to read “We find that SWIF(r) loses power to identify sites with adaptive mutations when iHS is undefined, likely because iHS is far more powerful for detecting incomplete sweeps than the other four statistics. SWIF(r) loses very little power when other component statistics are undefined (Supplementary Figure 33).” To further acknowledge the reviewer’s point about calibration, we have altered lines 345-350 to read: “For calibrating SWIF(r), we chose training training sets made up overwhelmingly of neutral variants; while our calibration of SWIF(r) always preserves the rank order of posterior probabilities (Supplementary Figure 3), the specific choice of training set makeup can have a dramatic effect on the calibration, and future applications of SWIF(r) can use different criteria for calibration. For example, one could calibrate SWIF(r) based on specific strengths of selection, or perform different calibrations for scenarios in which certain component statistics are undefined.”

7) The claim in line 32 that SWIFr is unique in learning joint distributions of the summary stats is also not true. Thus in lines 29-34 the only true claim to uniqueness in my opinion is that SWIFr calculates calibrated probabilities—you should be acknowledging the field here more than you are.

We agree with the reviewer that learning joint distributions of summary statistics is not unique to SWIF(r): as stated in lines 34-35 of our previous submission, “Existing composite methods for selection scans have subsets of these features, but SWIF(r) combines all three in a unified statistical framework.” In this revision, we further underscore that these individual features are available in other methods with the following revised text (lines 34-36): “Existing composite methods for selection scans have subsets of these features (e.g. CMS returns site-based scores, and evoBoosting, evoNet, and other machine learning approaches leverage correlations among component statistics), but SWIF(r) combines all three in a unified statistical framework.”

8) My point 32 in my original review still stands. 100 simulations is way too small—you are claiming that 100 realizations of the ARG gives you 400,000 sites for training, however those sites have correlated histories. My point is that 100 realizations from the ARG is far too little. This number need to be pushed up considerably.

We appreciate the reviewer's concern that our neutral simulations are not sufficient for training SWIF(r). To test whether this was the case, we trained a second version of SWIF(r) that was identical except that it used 1,000 1MB neutral simulations, and then we compared the two versions of SWIF(r) on both simulated data and on data from the 1000 Genomes Project (attached at the end of this response). In both cases, we find that the increase in training simulations does not alter our downstream results: first, ROC curves on simulated data show identical performance for the two versions of SWIF(r) for the task of distinguishing neutrally evolving mutations from adaptive mutations, and second, the probabilities returned by the second version of SWIF(r) (based on 1,000 simulations) for sites in the 1000 Genomes dataset correlate nearly perfectly ($r^2=0.999$) with the probabilities returned by the first version of SWIF(r) (based on 100 simulations). We attribute these results to the Gaussian Mixture Model smoothing that we apply to the learned joint distributions, which is intended to help SWIF(r) avoid overfitting. Based on these results, we have respectfully chosen not to regenerate all of our downstream analyses and figures.

Reviewer #3 (Remarks to the Author):

Alpert Sugden et al. have made a tremendous effort in revising their manuscript. It is clear that the manuscript has greatly improved with the aid of reviewer comments, and I am satisfied with the way that my prior comments were addressed. I have no further concerns.

Sincerely,
Michael DeGiorgio

We thank the reviewer for signing his review and acknowledge him by name in our Acknowledgments section; his comments improved the manuscript and method.

References Cited:

- 1000 Genomes Project Consortium. "A global reference for human genetic variation." *Nature* 526, no. 7571 (2015): 68-74.
- Gravel, Simon, Brenna M. Henn, Ryan N. Gutenkunst, Amit R. Indap, Gabor T. Marth, Andrew G. Clark, Fuli Yu et al. "Demographic history and rare allele sharing among human populations." *Proceedings of the National Academy of Sciences* 108, no. 29 (2011): 11983-11988.
- Li, Heng, and Richard Durbin. "Inference of human population history from individual whole-genome sequences." *Nature* 475, no. 7357 (2011): 493-496.
- Schaffner, Stephen F., Catherine Foo, Stacey Gabriel, David Reich, Mark J. Daly, and David Altshuler. "Calibrating a coalescent simulation of human genome sequence variation." *Genome research* 15, no. 11 (2005): 1576-1583.
- Schiffels, Stephan, and Richard Durbin. "Inferring human population size and separation history from multiple genome sequences." *Nature genetics* 46, no. 8 (2014): 919-925.
- Sheehan, Sara, Kelley Harris, and Yun S. Song. "Estimating variable effective population sizes from multiple genomes: a sequentially Markov conditional sampling distribution approach." *Genetics* 194, no. 3 (2013): 647-662.

Sheehan, Sara, and Yun S. Song. "Deep learning for population genetic inference." *PLoS Computational Biology* 12, no. 3 (2016): e1004845.

	Auton PSMC population sizes scaled ($4N\mu*10^3$)		Gravel inferred population sizes							
	A=pre-expansion	B=present-day	inferred Ne		scaled ($\mu=1.25e-8$)		scaled ($\mu=1.5e-8$)		scaled ($\mu=2.36e-8$)	
			A	B	A	B	A	B	A	B
YRI	~0.5	~0.55	15500	15500	0.775	0.775	0.93	0.93	1.4632	1.4632
CEU	~0.1	~0.85	1032	35900	0.0516	1.795	0.06192	2.154	0.097421	3.38896
CHB	~0.1	~1.35	550	49000	0.0275	2.45	0.033	2.94	0.05192	4.6256

Table 1. Recent population expansion inferred by Auton *et al.* (2015, Nature) and Gravel *et al.* (2012, PNAS) for YRI (Yoruba from Ibadan, Nigeria), CEU (northern and western European from Utah), and CHB (Han Chinese from Beijing). Both Auton and Gravel infer strong recent population growth in non-African populations. Values from Auton *et al.* are scaled in units of $4N\mu*10^3$, and were estimated from their PSMC plot in Extended Data Figure 7 (here, columns 2 and 3). Population sizes are reported in Gravel *et al.* in terms of Ne (columns 4 and 5). In columns 6-11, we use three different mutation rates to generate scaled population sizes that can be compared with the values from Auton *et al.* Columns 6-7 use mutation rate $\mu=1.25e-8$, the lower end of the interval used by Auton *et al.*, and columns 8-9 use $\mu=1.5e-8$, the upper end of that interval. Columns 10-11 use $\mu=2.36e-8$, the value used by Gravel *et al.* for demographic inference. Scaled population sizes are calculated before expansion (labeled A; 15kya for Auton, 23kya for Gravel) and at present-day (labeled B). Bold values are highlighted to illustrate the comparisons of pre-expansion and present-day scaled population sizes inferred by the two methods for one of the mutation rates; in both cases, and with all mutation rates, the population size change inferred by Gravel *et al.* is more extreme than that inferred by Auton *et al.*

Extended Data Figure 7 | Unsmoothed PSMC curves. a, The median PSMC curve for each population. b, PSMC curves estimated separately for all individuals within the 1000 Genomes sample. c, Unsmoothed PSMC curves comparing estimates from the low coverage data (dashed lines) to those

obtained from high coverage PCR-free data (solid lines). Notable differences are confined to very recent time intervals, where the additional rare variants identified by deep sequencing suggest larger population sizes.

A**B**
Increase in the number of neutral simulations does not change downstream results of SWIF(r). We trained a second version of SWIF(r) that was identical to the one in our original submission, except that it used 1,000 1MB neutral simulations. We then compared the two versions of SWIF(r) on both simulated data and on data from the 1000 Genomes Project. In both cases, we find that the increase in training simulations does not alter our downstream results: **A)** ROC curves on simulated data show identical performance for the two versions of SWIF(r) for the task of distinguishing neutrally evolving mutations from adaptive mutations. **B)** The probabilities returned by the second version of SWIF(r) (based on 1,000 simulations) for sites in the 1000 Genomes dataset correlate nearly perfectly ($r^2=0.999$) with the probabilities returned by the first version of SWIF(r) (based on 100 simulations).

Reviewer #2 (Remarks to the Author):

The revised manuscript meets my expectations at this point .

REVIEWERS' COMMENTS:

Reviewer #2 (Remarks to the Author):

The revised manuscript meets my expectations at this point

We appreciate this reviewer's input, and believe that their comments in previous reviews have greatly improved the manuscript. We are very happy that the reviewer is satisfied with our revisions.